# Utility Boundary of Dataset Distillation: Scaling and Configuration-Coverage Laws

## Abstract

Dataset distillation (DD) aims to construct compact synthetic datasets that allow models to achieve comparable performance to full-data training while substantially reducing storage and computation. Despite rapid empirical progress, its theoretical foundations remain limited: existing methods (gradient, distribution, trajectory matching) are built on heterogeneous surrogate objectives and optimization assumptions, which makes it difficult to analyze their common principles or provide general guarantees. Moreover, it is still unclear under what conditions distilled data can retain the effectiveness of full datasets when the training configuration, such as optimizer, architecture, or augmentation, changes. To answer these questions, we propose a unified theoretical framework, termed configuration–dynamics–error analysis, which reformulates major DD approaches under a common generalization-error perspective and provides two main results: (i) a scaling law that provides a single-configuration upper bound, characterizing how the error decreases as the distilled sample size increases and explaining the commonly observed performance saturation effect; and (ii) a coverage law showing that the required distilled sample size scales linearly with configuration diversity, with provably matching upper and lower bounds. In addition, our unified analysis reveals that various matching methods are interchangeable surrogates, reducing the same generalization error, clarifying why they can all achieve dataset distillation and providing guidance on how surrogate choices affect sample efficiency and robustness. Experiments across diverse methods and configurations empirically confirm the derived laws, advancing a theoretical foundation for DD and enabling theory-driven design of compact, configuration-robust dataset distillation.

## 1 Introduction

*Dataset distillation (DD)* (Wang et al., 2018; Sucholutsky & Schonlau, 2021), also known as dataset condensation (DC) (Zhao et al., 2020; Wang et al., 2022), seeks to synthesize a compact dataset that enables models to approach the accuracy of full-data training while greatly reducing storage and compute costs. Over the past few years, three main categories of matching-based methods have emerged. *Gradient matching (GM)* aligns gradients between real and synthetic data through bilevel optimization, extended by augmentation consistency (Zhao & Bilen, 2021), diversity regularization (Cazenavette et al., 2023), and reverse matching (Ye et al., 2024). *Distribution matching (DM)* matches feature statistics, from early MMD-based formulations (Li et al., 2017) to higher-order or quantile-based variants (Wang et al., 2022; Zhang et al., 2024; Wei et al., 2024). *Trajectory matching (TM)* aligns full optimization dynamics, introduced in MTT (Cazenavette et al., 2022) and later extended to self-supervised and detection tasks (Lee et al., 2023; Qi et al., 2024) (a comprehensive review of related work is provided in Appendix B).

Despite empirical advances, the theoretical foundation of DD remains fragmented. Existing analyses are confined to paradigm-specific assumptions: GM theory is largely restricted to first-order gradient matching (Zhao & Bilen, 2021; Deng & Russakovsky, 2022); DM relies on kernel- or moment-based statistics (Li et al., 2017), thereby neglecting optimization dynamics; and TM, while empirically strong (Cazenavette et al., 2022), lack rigorous convergence guarantees beyond heuristic approximations. These paradigm-specific limitations highlight the absence of a unified theoretical view to relate different DD approaches, making it unclear why all three categories can yield near full-data utility or how to explain recurring empirical patterns such as the saturation of accuracy

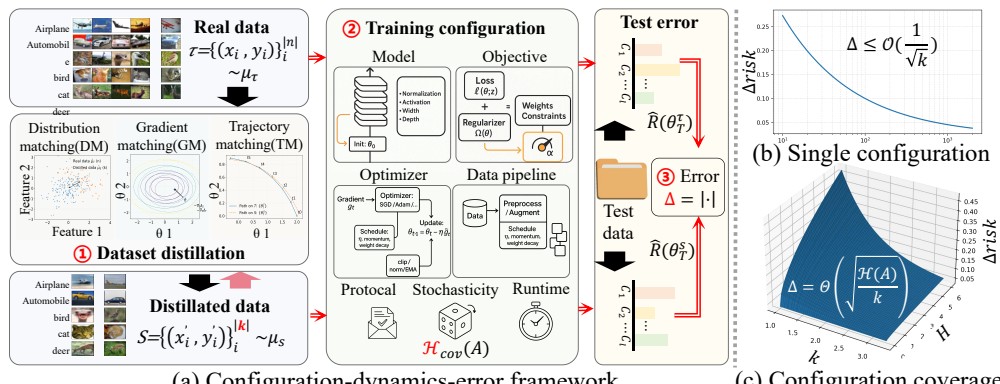

(a) Configuration-dynamics-error framework  (c) Configuration coverage

Figure 1: Configuration-dynamics-error framework: a configuration (optimizer, architecture, augmentation, etc.) together with the training distribution (either the real dataset or its distilled dataset) induces optimization dynamics, whose risk is evaluated through generalization error bounds; this yields the scaling law for a single configuration and the coverage law across configuration families.

gains with larger distilled sample sizes (Cazenavette et al., 2022). A further challenge is robustness to *training-configuration* shifts, e.g., changes in optimizer, architecture, or augmentation between distillation and downstream training. Since DD is expected to substitute for the full dataset in practice, distilled data must remain effective under such shifts. Yet current evaluations often restrict to fixed setups or mild parameter perturbations (Nguyen et al., 2021; Zhao & Bilen, 2023), and reported results reveal instability: sensitivity to random seeds (Wang et al., 2018), reliance on augmentation (Zhao & Bilen, 2021), and weak cross-architecture transfer (Liu et al., 2022). To this, we introduce the notion of a *utility boundary*: the relationship between the distilled sample size and the diversity of configurations within which the distilled dataset can still match the performance of full-data training.

We address these challenges by proposing a unified *configuration–dynamics–error* framework. A configuration specifies the update operator (e.g., optimizer, architecture, augmentation) that governs parameter changes; together with the training distribution this induces the optimization dynamics, and the resulting risk is evaluated through generalization error bounds on a common test distribution. Building on this framework, we first analyze the single-configuration case, yielding (1) *scaling law:* As the distilled sample size $k$ increases, the generalization error decreases until it reaches the irreducible error bound $\epsilon_{\text{bound}}$, determined by the configuration, following the statistical rate $\Delta \leq \mathcal{O}(1/\sqrt{k}) + \epsilon_{\text{bound}}$. This explains the commonly observed images per class (IPC) saturation: when $k$ is sufficiently large, the error is dominated by the irreducible floor and accuracy can no longer improve by enlarging $k$. In this regime, reducing $\epsilon_{\text{bound}}$ is more valuable than blindly increasing $k$. (2) Then, to account for generalization across configurations, we extend the analysis from a single configuration to a family of configurations. In this setting, $\mathcal{H}(\mathcal{A}, r)$ measures configuration diversity (e.g., via a covering number), yielding the *configuration-coverage law:* $\Delta = \Theta(\sqrt{\mathcal{H}(\mathcal{A}, r)/k})$. This characterizes the utility boundary of dataset distillation for the first time, showing how the required sample size must grow with configuration diversity to maintain generalization.

Together, these laws clarify why different DD methods can behave consistently. Within this framework, GM, DM, and TM are not independent heuristics but instances of a common objective: minimizing the *matching discrepancy* that measures how differently real and distilled data drive training dynamics under a configuration. While DD methods align different objects, gradients, optimization trajectories, or feature-level statistics, our theoretical analysis, corroborated by experiments across diverse methods and configurations, shows that these distinctions are merely surrogate choices within a common outer–inner (bi-level) mechanism. Their generalization is therefore governed by the same scaling law in single configurations and the same coverage law across configuration diversity. Thus, our framework unifies diverse DD methods under a single generalization error bound analysis and provides guidance for designing distilled datasets that achieve both sample efficiency and robustness. To summarize, we make the following contributions in this work:

- *Unified framework.* We introduce a configuration–dynamics–error framework that places GM, DM, and TM within a single generalization error analysis.

- *Scaling law.* We provide the first generalization bound that relates the distilled sample size to test error in a fixed configuration, explaining the IPC saturation phenomenon.

- *Coverage law.* We derive the first formal utility boundary showing how distilled sample size should scale with configuration diversity.

- *Unified DD.* We unify the three categories of DD methods under the proposed framework and empirically confirm the theoretical findings across representative methods and datasets.

## 2 PRELIMINARIES AND PROBLEM SETUP

**Dataset distillation (DD).** Given a real dataset $\mathcal{D}_\tau = \{(x_i, y_i)\}_{i=1}^n$ with empirical distribution $\hat{\mu}_\tau = \frac{1}{n}\sum_{i=1}^n \iota_{(x_i,y_i)}$[1], the goal of DD is to construct a compact synthetic dataset $\mathcal{D}_s = \{(x'_j, y'_j)\}_{j=1}^k$ with empirical distribution $\hat{\mu}_s = \sum_{j=1}^k \iota_{(x'_j, y'_j)}$ for $k \ll n$, such that training on $\hat{\mu}_s$ matches training on $\hat{\mu}_\tau$ in test performance (Wang et al., 2018; Sucholutsky & Schonlau, 2021; Zhao et al., 2020). Let $\nu$ denote a test distribution with empirical counterpart $\hat{\nu} = \frac{1}{m}\sum_{l=1}^m \iota_{(x_l^{\text{te}}, y_l^{\text{te}})}$. We evaluate empirical and population risks:

$$\hat{R}(\theta) = \mathbb{E}_{\hat{\nu}}[\ell(\theta; z)], \qquad R_\nu(\theta) = \mathbb{E}_\nu[\ell(\theta; z)], \tag{1}$$

where $z$ presents a data point, and compare models trained on $\hat{\mu}_s$ versus $\hat{\mu}_\tau$.

**Single training configuration.** Most theories are established under a single fixed training configuration (e.g., optimizer, hyperparameters, augmentation, architecture) and show that particular matching strategies (gradient, distribution, or trajectory) improve accuracy in the single setting (Zhao et al., 2020; Cazenavette et al., 2022; Zhao & Bilen, 2023). Formally, one training step under a (fixed) configuration can be written as:

$$\theta_{t+1} = \Phi(\theta_t; \mu) = \theta_t - \eta\, P(\theta_t)\, \mathbb{E}_\mu[g(\theta_t; z)], \tag{2}$$

where $P(\theta)$ denotes a (possibly adaptive) preconditioner and $g$ the per-sample update. We refer to the configuration used to generate the distilled dataset as the *source configuration*. The configuration under which we evaluate and compare the utility of real versus distilled data is called the *target configuration*.

## 3 A UNIFIED CONFIGURATION-DYNAMICS-ERROR FRAMEWORK

In practice, the ultimate goal of a distilled dataset is to replace the real dataset across diverse applications. Since the source configurations used for deployment may differ from those assumed during generation, it is necessary to extend the analysis from a single setup to a *family of configurations*. Doing so highlights three key aspects that jointly determine the effectiveness of distilled data: the *diversity* of configurations it must cover, the *alignment* it maintains with real data, and the *stability* with which training dynamics transfer across configurations.

**Space of configurations.** An configuration $a$ specifies optimizer, hyperparameters, augmentation, and architecture. The training under configuration $a$ on distribution $\mu$ induces parameter iterate:

$$\theta_{t+1} = \Phi_a(\theta_t; \mu) = \theta_t - \eta\, P_a(\theta_t)\, \mathbb{E}_\mu\, g_a(\theta_t; z) \in \Gamma_a, \tag{3}$$

with feasible set $\Gamma_a$ for parameters. The way we update parameters here generalizes classical stochastic approximation and adaptive methods (Robbins & Monro, 1951; Bottou et al., 2018). We consider a target configuration family $\mathcal{A} \subseteq \mathcal{C}$ of training configurations that reflects the intended deployment setting (Shalev-Shwartz & Ben-David, 2014).

**Diversity.** Each configuration $a \in \mathcal{A}$ induces its own feasible parameter set $\Gamma_a \subseteq \mathbb{R}^d$. To compare two configurations $a, a' \in \mathcal{A}$ on the same real data $\hat{\mu}_\tau$, we define the configuration-distance

$$d_{\mathcal{A}}(a, a') = \sup_{\theta \in \Gamma_a \cap \Gamma_{a'}} \left\| P_a(\theta)\, \mathbb{E}_{\hat{\mu}_\tau} g_a(\theta; z) - P_{a'}(\theta)\, \mathbb{E}_{\hat{\mu}_\tau} g_{a'}(\theta; z) \right\|_2. \tag{4}$$

This metric is inspired by the stability and uniform convergence analyses of algorithmic dynamics (Hardt et al., 2016; Raginsky et al., 2017). The *coverage complexity* of $\mathcal{A}$ at radius $r > 0$ under

---

[1] $\iota$ denotes the Dirac measure that assigns unit mass to the sample point.

$d_{\mathcal{A}}$ is $\mathcal{H}_{\mathrm{cov}}(\mathcal{A}, r) = \log N(\mathcal{A}, d_{\mathcal{A}}, r)$, where $N(\mathcal{A}, d_{\mathcal{A}}, r)$ is the minimal number of $d_{\mathcal{A}}$-balls of radius $r$ needed to cover $\mathcal{A}$. Here $r$ is mathematically the covering *radius*, which determines the *resolution* of ecological distinctions: smaller $r$ resolves finer differences and thus increases $\mathcal{H}_{\mathrm{cov}}(\mathcal{A}, r)$.

**Alignment.** For measures $\mu, \nu$ and configuration $a$, we define the *matching discrepancy*

$$\Delta_a(\mu, \nu) := \sup_{\theta \in \Gamma_a} \big\| P_a(\theta)\big(\mathbb{E}_\mu g_a(\theta; z) - \mathbb{E}_\nu g_a(\theta; z)\big)\big\|_2. \tag{5}$$

This notion unifies classical discrepancy measures used in dataset distillation and domain adaptation (Zhao et al., 2020; Cazenavette et al., 2022; Zhao & Bilen, 2023; Ben-David et al., 2006). Specializing to empirical real and empirical synthetic distributions gives $\Delta_a(\hat{\mu}_\tau, \hat{\mu}_s)$.

**Stability.** The final stage of our configuration–dynamics–error framework concerns how discrepancies in dynamics translate into generalization error. Throughout the paper, our analysis builds on stability- and information-theoretic approaches to generalization (Bousquet & Elisseeff, 2002; Russo & Zou, 2016; Xu & Raginsky, 2017), and decomposes the error into three components: (i) an optimization residual, determined by the optimization steps $T$; (ii) statistical fluctuations, arising from finite sample sizes $n, m, k$; and (iii) a matching term, governed by the alignment discrepancy $\Delta_a$. This unified form can be summarized as

$$|R_\nu(\theta_T^{(\hat{\mu}_s; a)}) - R_\nu(\theta_T^{(\hat{\mu}_\tau; a)})| \lesssim \underbrace{\text{opt. residual}}_{T} + \underbrace{\text{stat. fluctuations}}_{n, m, k} + \underbrace{\text{matching term}}_{\Delta_a}. \tag{6}$$

At this point we have a complete *configuration–dynamics–error* framework as Figure 1. Configurations specify the update operators that drive parameter changes; their diversity is captured through covering complexity under a configuration distance (*Diversity*); the gap between synthetic and real data within each configuration is measured by the matching discrepancy (*Alignment*); and the transfer from dynamics to generalization error is governed by generalization error bounds decomposition (*Stability*). Together, these elements form a coherent chain from configuration to dynamics to error, closing the framework and setting the stage for Sections 4-5 on theoretical analysis next.

# 4 SINGLE–CONFIGURATION GENERALIZATION BOUND

We instantiate the Configuration-dynamics-error framework with a fixed configuration $a$ and derive a finite–sample bound that reveals the scaling law of dataset distillation.

**Assumption 4.1** (Regularity). On the feasible parameter domain $\Gamma_a$, we assume: (i) bounded per–sample update and preconditioner, i.e., $\|g_a(\theta; z)\| \leq B_g$, $\|P_a(\theta)\| \leq \kappa_a$; (ii) bounded loss $|\ell(\theta; z)| \leq B_\ell$ and $L_R$–Lipschitz test risk $\hat{R}$; (iii) contractive dynamics under a PL–type condition, with contraction rate $\rho_a \in (0, 1)$ and constant $C_{2,a} = (1 - \rho_a)/L_R$. These assumptions are standard in stability/generalization analyses and PL-based convergence (see Hardt et al. (2016); Bousquet & Elisseeff (2002); Karimi et al. (2016)).

**Definition 4.1** (Intrinsic generalization error). For configuration $a$, the $k$–prototype class of distributions, $\mathcal{P}_k = \{\sum_{j=1}^k \iota_{z_j} : \iota\}$, induces an irreducible generalization error $\Delta_a^\star := \inf_{\mu \in \mathcal{P}_k} \Delta_a(\mu_\tau, \mu)$. It measures the best possible matching between $k$ prototypes and the real dataset under configuration $a$, and determines this error in our bound.

**Theorem 4.2** (Single–configuration risk bound). *Let $\theta_T^{(s)}$ and $\theta_T^{(\tau)}$ denote the parameters after $T$ steps trained on synthetic and real data, respectively, with initialization gap $\delta_0 = \theta_0^{(s)} - \theta_0^{(\tau)}$. Then with probability at least $1 - \varepsilon$,*

$$|R_\nu(\theta_T^{(s)}) - R_\nu(\theta_T^{(\tau)})| \leq L_R \rho_a^T \|\delta_0\| + \frac{\eta \kappa_a}{C_{2,a}}(\Delta_a^\star + e_g) + e_{\mathrm{te}}, \tag{7}$$

*where $e_g = \mathcal{O}(1/\sqrt{k} + 1/\sqrt{n})$ is the fluctuation from distillation and training samples (see, e.g., Bartlett & Mendelson (2002), for Rademacher-based rates), and $e_{\mathrm{te}} = \mathcal{O}(1/\sqrt{m})$ is the test concentration error (see, e.g., Vershynin (2018), for standard sub-Gaussian bounds). If distilled dataset $\mathcal{D}_s$ generate depending on real dataset $\mathcal{D}_\tau$, then $e_g$ further incurs an information–theoretic penalty $\mathcal{O}(\sqrt{I(\mathcal{D}_s; \mathcal{D}_\tau)/k})$ (Russo & Zou, 2016) (see Appendix C for proof details).*

*Remark* 4.3. As $T, n, m$ are sufficiently large, optimization and statistical terms vanish. We then arrive at the single–configuration *scaling law*:

$$\left| R_\nu(\theta_T^{(s)}) - R_\nu(\theta_T^{(\tau)}) \right| \approx \frac{\eta \kappa_a}{C_{2,a}} \Delta_a^\star + \mathcal{O}(1/\sqrt{k}). \tag{8}$$

**Corollary 4.4.** *For a target error $\epsilon_0$, distilled sample size $k$ must satisfy*

$$k = \Omega\Big((\epsilon_0 - \Delta_a^\star \eta \kappa_a / C_{2,a})^{-2}\Big). \tag{9}$$

*Remark* 4.5. i) The generalization error could decrease with $k$ until saturation at the irreducible error bound $\Delta_a^\star \eta \kappa_a / C_{2,a}$, which accounts for the commonly observed IPC saturation; ii) With finite training and test samples $n, m$, their sizes may limit accuracy, but the distilled sample size $k$ remains the fundamental bottleneck of distillation since $k \ll n, m$.

## 5 COVERAGE-AWARE BOUNDS: RISK ACROSS TRAINING CONFIGURATIONS

The preceding remark establishes the local scaling behavior under a fixed configuration, highlighting the role of $k$ as the fundamental bottleneck. In practice, however, distilled data are expected to remain effective across not just one but a family of configurations $\mathcal{A} \subseteq \mathcal{C}$ (optimizers, architectures, augmentations). In what follows, we analyze how the generalization error scales with the configuration diversity of $\mathcal{A}$, derive the corresponding upper and lower bounds, and ultimately arrive at a tight *coverage law*.

With concepts introduced in Sections 2–4, ranging from configuration–distance $d_\mathcal{A}$, coverage diversity $\mathcal{H}_{\text{cov}}(\mathcal{A}, r) = \log N(\mathcal{A}, d_\mathcal{A}, r)$, matching discrepancy $\Delta_a$, irreducible generalization error $\Delta_a^\star$, to dynamics constants $\rho_a$ and $C_{2,a}$, we further introduce Rademacher constants $C_G^+, \tilde{C}_G^{+2}$, and extend the Lipschitz assumption:

**Assumption 5.1** (Lipschitz transfer across configurations and parameters). *There exist $L_{\text{conf}}, L_\theta > 0$ such that for all $a, a' \in \mathcal{C}$, all $\theta, \theta' \in \Gamma_a \cap \Gamma_{a'}$, and $\mu \in \{\hat{\mu}_\tau, \hat{\mu}_s\}$,*

$$\big\| P_a(\theta)\mathbb{E}_\mu g_a(\theta; z) - P_{a'}(\theta)\mathbb{E}_\mu g_{a'}(\theta; z) \big\|_2 \le L_{\text{conf}}\, d_\mathcal{A}(a, a'),$$

$$\big\| P_a(\theta)\mathbb{E}_\mu g_a(\theta; z) - P_a(\theta')\mathbb{E}_\mu g_a(\theta'; z) \big\|_2 \le L_\theta \, \|\theta - \theta'\|_2,$$

which requires smooth variation across configurations and parameter–Lipschitz continuity; it is mild, as common optimizers such as SGD (with learning rates in $[10^{-3}, 10^{-1}]$) and Adam (with $\beta \in [0.8, 0.999]$) satisfy it in practice.

Based on the mild extension of $d_\mathcal{A}$, which only requires the uniform Lipschitz continuity over $\{\hat{\mu}_\tau, \hat{\mu}_s\}$ and all $\theta \in \Gamma$, the $d_\mathcal{A}$–based coverage argument extends consistently from cover centers to configuration family $\mathcal{A}$, and the configuration-dynamic-risk framework can extend from single configuration points to all configurations.

**Theorem 5.1** (Uniform cross–configuration bound). *For any $\varepsilon \in (0, 1)$, with probability at least $1 - \varepsilon$ over the draws of $\hat{\mu}_\tau, \hat{\mu}_s, \hat{\nu}$, it holds for any configuration prior $\Pi$ supported on $\mathcal{A}$ that*

$$\mathbb{E}_{a \sim \Pi}\big| R_\nu(\theta_T^{(s,a)}) - R_\nu(\theta_T^{(\tau,a)}) \big| \ \le \ \epsilon_{\text{bound}}^{\text{upper}} + A_1 \tfrac{\mathcal{H}_{\text{cov}}(\mathcal{A}, r)}{k} + A_2 \sqrt{\tfrac{\mathcal{H}_{\text{cov}}(\mathcal{A}, r)}{k}}, \tag{10}$$

$$\sup_{a \in \mathcal{A}} \big| R_\nu(\theta_T^{(s,a)}) - R_\nu(\theta_T^{(\tau,a)}) \big| \ \le \ \epsilon_{\text{bound}}^{\text{upper}} + \tfrac{C_{\text{cov}}(\mathcal{A})}{\sqrt{k}}, \tag{11}$$

$$\epsilon_{\text{bound}}^{\text{upper}} = \mathcal{O}\Big( \rho_{\max}^T \|\delta_0\| + \sup_{a \in \mathcal{A}} \Delta_a^\star + \tfrac{1}{\sqrt{n}} + \tfrac{\sqrt{\mathcal{H}_{\text{cov}}(\mathcal{A}, r)}}{\sqrt{m}} \Big), \quad C_{\text{cov}}(\mathcal{A}) = \mathcal{O}\big(\sqrt{\mathcal{H}_{\text{cov}}(\mathcal{A}, r)}\big).$$

*If $\mathcal{D}_s$ depends on $\mathcal{D}_\tau$, an additional correction $\mathcal{O}\Big(\sqrt{I(\mathcal{D}_s; \mathcal{D}_\tau)/k}\Big)$ is added to both bounds (see Appendix D.1 and D.2 for proof details).*

The above upper bound indicates that the required distilled size increases linearly with *the configuration diversity* $\mathcal{H}_{\text{cov}}(\mathcal{A})$. A natural further question then is: *is this dependence optimal?* The next theorem answers it in the affirmative by giving a matching lower bound and thus justifying the optimal dependence of the distilled sample size on the configuration diversity.

---

[2] $C_G^+$ denotes the supremum Rademacher complexity constant across configurations. When finite-sample or information-theoretic corrections are present, we denote the corrected version by $\tilde{C}_G^+$; both share the same order.

**Theorem 5.2** (Coverage lower bound). *Suppose Assumption 4.1 holds (single–configuration regularity). Assume further an* identifiability condition*: there exists $\lambda > 0$ such that, for all $\theta \in \Gamma$ and any two distinct configurations $a, a' \in \mathcal{A}$, $\left\| P_a(\theta) \mathbb{E}_{\hat{\mu}_\tau} g_a(\theta; z) - P_{a'}(\theta) \mathbb{E}_{\hat{\mu}_\tau} g_{a'}(\theta; z) \right\|_2 \geq \lambda\, d_{\mathcal{A}}(a, a')$. That is, update dynamics corresponding to different configurations are uniformly separated. Let $\mathcal{A}$ admit a $\rho$–packing with $M$ elements (so that $\mathcal{H}_{\mathrm{cov}}(\mathcal{A}, r) = \log M$). Then, for any distillation algorithm producing $k$ synthetic samples, there exists a distribution over this packed family such that*

$$\mathbb{E}_a \left| R_\nu(\theta_T^{(s,a)}) - R_\nu(\theta_T^{(\tau,a)}) \right| \geq \epsilon_{\mathrm{bound}}^{\mathrm{lower}} + c_{\mathrm{lb}}\, \rho \lambda\, \sqrt{\tfrac{\mathcal{H}_{\mathrm{cov}}(\mathcal{A},r)}{k}},$$

*where $c_{\mathrm{lb}} \in (0, 1)$ is a universal constant (see Appendix D.4 for proof details).*

**Corollary 5.3** (Coverage law). *For any generalization error $\epsilon_0 > \epsilon_{\mathrm{bound}}$, if the distilled sample size satisfies $k \geq K_{\min}(\epsilon_0, \mathcal{A}) = \left( \frac{C_{\mathrm{cov}}(\mathcal{A})}{\epsilon_0 - \epsilon_{\mathrm{bound}}} \right)^2 = \Theta\big(\mathcal{H}_{\mathrm{cov}}(\mathcal{A}, r)\big)$, then it holds that*

$$\sup_{a \in \mathcal{A}} \left| R_\nu(\theta_T^{(s,a)}) - R_\nu(\theta_T^{(\tau,a)}) \right| \leq \epsilon_0. \tag{12}$$

*Remark* 5.4 (Coverage law). (i) The upper bound separates two errors: an approximation error $\mathcal{H}_{\mathrm{cov}}/k$, since a size-$k$ set cannot fully cover $\mathcal{A}$, and a concentration error $\sqrt{\mathcal{H}_{\mathrm{cov}}/k}$ from uniform guarantees. In typical regimes, the latter dominates, as it decays more slowly and thus sets the critical rate. (ii) The residual $\epsilon_{\mathrm{bound}}$ collects optimization error, irreducible matching discrepancy, and sampling noise, yielding a non-vanishing floor. (iii) Eq. (10) gives an *average-case* guarantee under a prior $\Pi$, while Eq. (11) strengthens it to a *worst-case* guarantee across all configurations, reducing to the $\sqrt{\mathcal{H}_{\mathrm{cov}}/k}$ rate. (iv) The lower bound shows that any target error $\epsilon_0 > \epsilon_{\mathrm{bound}}$ requires $k = \Omega\big(\mathcal{H}_{\mathrm{cov}}(\mathcal{A}, r)\big)$. This matches the upper bound $k = \mathcal{O}\big(\mathcal{H}_{\mathrm{cov}}(\mathcal{A}, r)\big)$ up to constants. Taken together, these results establish the tightness of the coverage law: as $\mathcal{H}_{\mathrm{cov}}$ grows, $k$ must scale proportionally to maintain accuracy, and no distillation algorithm can avoid the $\sqrt{\mathcal{H}_{\mathrm{cov}}/k}$ barrier.

*Remark* 5.5 (Practical estimation of coverage entropy). Since the coverage law depends on $\mathcal{H}_{\mathrm{cov}}(\mathcal{A}, r)$, an important practical question is how to estimate it. In principle, computing $H_{\mathrm{cov}}(\mathcal{A}, r)$ requires solving a minimal covering problem over $(\mathcal{A}, d_{\mathcal{A}})$, which is NP–hard. As an approximation, one may define $d_{\mathcal{A}}(a_i, a_j)$ via the averaged $\ell_2$-distance between their normalized one-step updates (e.g., preconditioned gradients at the same initialization and mini-batches), and apply a greedy $r$–cover to obtain an empirical covering number $N_r$ with $\widehat{H}_{\mathrm{cov}} = \log N_r$. For tractability, however, our experiments use $\log M$, the logarithm of the number of candidate configurations, as a proxy. Since $1 \leq N_r \leq M$ and, under mild Lipschitz assumptions on $d_{\mathcal{A}}$, $N_r$ is typically of the same order as $M$, $\log M$ preserves the dominant scaling with configuration diversity while avoiding costly pairwise distance computations.

*Remark* 5.6. New insights into dataset distillation emerge from the coverage law and its practical estimation: (i) the number of distilled samples $k$ must grow in proportion to the coverage complexity $\mathcal{H}_{\mathrm{cov}}(\mathcal{A}, r)$ in order to maintain a fixed generalization error; and (ii) Estimating $\mathcal{H}_{\mathrm{cov}}(\mathcal{A}, r)$, or using a proxy such as $\log M$, provides a principled way to determine how many distilled samples are needed to ensure that the synthetic dataset preserves the utility of the real dataset across a given configuration coverage. These insights connect the theoretical limits with practical guidelines for designing robust dataset distillation methods.

# 6 UNIFYING VARIOUS CATEGORIES OF DATASET DISTILLATION

From Sections 4-5, the matching discrepancy $\Delta_a(\hat{\mu}_\tau, \hat{\mu}_s)$ turns out to be the key to the generalization errors. We are now in a position to scrutinize why the three major distillation methods, i.e., DM, GM, and TM, albeit in different forms, all reduce the same $\Delta_a$ via the bi-level optimization mechanism.

**Unified bi-level optimization.** Let the distilled dataset be parameterized by $\xi$ with distribution $\mu(\xi)$, and $\Theta_j$ the inner states queried at outer iteration $j$. Each method minimizes a surrogate $\mathcal{M}_\phi(\mu(\xi); \hat{\mu}_\tau, b, \Theta_j)$ with $\phi \in \{\mathrm{DM}, \mathrm{GM}, \mathrm{TM}\}$, where $b$ denotes the *source configuration* under which distillation is performed (to distinguish it from the target configuration $a$ used for evaluation):

Inner loop (training under $b$): $\quad \theta_{t+1} = \theta_t - \eta\, P_b(\theta_t)\, \mathbb{E}_{z \sim \mu(\xi)} g_b(\theta_t; z)$,

Outer loop (updating $\xi$): $\quad \xi^{(j+1)} = \xi^{(j)} - \eta_j\, \nabla_\xi\, \mathcal{M}_\phi(\mu(\xi^{(j)}); \hat{\mu}_\tau, b, \Theta_j)$.

Table 1: Unified practical comparison and surrogate-to-alignment bridge. Left: how each branch is optimized; Middle: how its surrogate controls $\Delta_a$; Right: what drives the outer rate.

| | Outer obj. | Inner $\Theta_j$ | Robust | Compute | Bridge to $\Delta_a$ | Outer-rate driver |
|---|---|---|---|---|---|---|
| DM | $W_1$ MMD | none | **High** | Low | $\mathfrak{B}_{\mathrm{DM}}^{W_1} = \kappa_a L_{z,a} W_1$ $\mathfrak{B}_{\mathrm{DM}}^{MMD} = \kappa_a C_k \mathrm{MMD}_k$ | critic smoothness aug. strength |
| GM | Grad gap | 1-few $\theta$ | Mid | Mid | $\mathfrak{B}_{\mathrm{GM}} = \kappa_a |\Theta_j| \mathcal{M}_{\mathrm{GM}}$ | anchors/short path |
| TM | Path gap | unroll $L_b$ | Low | **High** | $\mathfrak{B}_{\mathrm{TM}} \simeq \kappa_a \frac{L_\theta + 2/\eta}{\omega_{\min}} \mathcal{M}_{\mathrm{TM}}$ | $L_b$/implicit grads |

**Method-specific surrogates.** The surrogate $\mathcal{M}_\phi$ takes different forms but all fit into the same bi-level template: DM compares empirical distributions using maximum mean discrepancy, e.g., $\mathcal{M}_{\mathrm{DM}} = \mathrm{MMD}_k(\hat{\mu}_s, \hat{\mu}_\tau)$, where the kernel embeds input-space geometry into an RKHS for measuring distributional differences. GM aligns average gradients at anchor states $\Theta_j$, e.g., $\mathcal{M}_{\mathrm{GM}} = \frac{1}{|\Theta_j|} \sum_{\theta \in \Theta_j} \left\| \mathbb{E}_{z \sim \hat{\mu}_s} g_b(\theta; z) - \mathbb{E}_{z \sim \hat{\mu}_\tau} g_b(\theta; z) \right\|_2$, forcing the synthetic set to match optimization directions on the real data. TM compares short optimization trajectories, e.g., $\mathcal{M}_{\mathrm{TM}} = \sum_{t=0}^{L_b} \omega_t \|\theta_t^{(s,b)} - \theta_t^{(\tau,b)}\|_2$, with weights $\omega_t$ along $L_b$ steps unrolled from a shared initialization.

Despite their differences, these surrogates admit the same contraction property:

$$\mathbb{E}[\mathcal{M}_\phi(\xi^{(j+1)})] \leq (1 - \alpha_\phi) \mathbb{E}[\mathcal{M}_\phi(\xi^{(j)})] + \epsilon_{\mathrm{est}}^{(\phi)}, \tag{13}$$

where the contraction rate is $\alpha_\phi = \eta_j \mu_\phi$ (under $L_\phi$–smoothness and a PL condition), and $\epsilon_{\mathrm{est}}^{(\phi)}$ denotes the estimation error arising from finite-sample approximations of the surrogate. This shows that all three objectives progressively contract their surrogate mismatches (see Appendix E.2 for proof details).

**Lemma 6.1** (Exchangeability of surrogates). *Fix configuration $a = b$. Under the smoothness, Lipschitz, and contraction conditions in Assumption 4.1, the matching discrepancy admits the bounds*

$$\Delta_a(\hat{\mu}_\tau, \hat{\mu}_s) \leq \underbrace{\kappa_a L_{z,a} W_1 \text{ or } \kappa_a C_k \mathrm{MMD}_k}_{\mathfrak{B}_{\mathrm{DM}}}, \quad \underbrace{\kappa_a |\Theta_j| \mathcal{M}_{\mathrm{GM}}}_{\mathfrak{B}_{\mathrm{GM}}}, \quad \underbrace{\kappa_a \frac{L_\theta + 2/\eta}{\omega_{\min}} \mathcal{M}_{\mathrm{TM}} + \kappa_a L_\theta \varepsilon_{\mathrm{path}}}_{\mathfrak{B}_{\mathrm{TM}}}, \tag{14}$$

*where $\mathfrak{B}_{\mathrm{DM}}$, $\mathfrak{B}_{\mathrm{GM}}$, and $\mathfrak{B}_{\mathrm{TM}}$ are the distribution-, gradient-, and trajectory-based surrogate bounds on $\Delta_a$. Moreover, these bounds are equivalent up to constant factors, i.e.,*

$$\mathfrak{B}_{\mathrm{TM}} = \mathcal{O}(\mathfrak{B}_{\mathrm{GM}}), \quad \mathfrak{B}_{\mathrm{GM}} = \mathcal{O}(\mathfrak{B}_{\mathrm{DM}}).$$

*Remark* 6.2 (Exchangeability of surrogates). The three surrogate bounds $\mathfrak{B}_*$ emphasize different controlling factors of the discrepancy: (i) $\mathfrak{B}_{\mathrm{DM}}$ is governed by feature-level divergence ($W_1$ or $\mathrm{MMD}_k$) scaled by data smoothness $L_{z,a}$; (ii) $\mathfrak{B}_{\mathrm{GM}}$ by gradient mismatch $\mathcal{M}_{\mathrm{GM}}$ amplified by the effective parameter size $|\Theta_j|$; (iii) $\mathfrak{B}_{\mathrm{TM}}$ by trajectory deviation $\mathcal{M}_{\mathrm{TM}}$ and truncation error $\varepsilon_{\mathrm{path}}$, weighted by step-size constants. Although these sources differ, distribution spread, gradient variability, and path stability, the resulting bounds remain equivalent up to constants, showing that DM, GM, and TM are interchangeable surrogates contracting the same discrepancy $\Delta_a$ (see notation details in Table 6 and Appendix A, see proof details in Appendix E.3 and E.4 ).

**Theorem 6.3** (Dynamic single–configuration bound for unifying DD methods). *Fix configuration $a = b$ and run $J$ outer steps with surrogate $\mathcal{M}_\phi$. Under Assumption 4.1 and the contraction property Eq. (13), with probability at least $1 - \varepsilon$,*

$$\left| R_\nu(\theta_T^{(s,a)}) - R_\nu(\theta_T^{(\tau,a)}) \right| \leq \epsilon_{\mathrm{bound}}^{\mathrm{distillation}} + \epsilon_{\mathrm{method}}^{(\phi)} + \epsilon_k^{(\phi)},$$

*where $\epsilon_{\mathrm{bound}}^{\mathrm{distillation}} = L_R \rho_a^T \|\delta_0\| + \mathcal{O}(1/\sqrt{m})$, $\epsilon_{\mathrm{method}}^{(\phi)} = \mathcal{O}(\frac{C_{\phi,a}}{C_{2,a}}[(1 - \alpha_\phi)^J \mathcal{M}_\phi(\xi^{(0)}) + \epsilon_{\mathrm{est}}^{(\phi)}])$, and $\epsilon_k^{(\phi)} = \tilde{\mathcal{O}}(1/\sqrt{k})$.*

*Remark* 6.4. $\epsilon_{\mathrm{bound}}^{\mathrm{distillation}}$ represents the irreducible generalization error, collecting the method–independent terms that do not vanish with larger $k$, including optimization residuals due to finite $T$ and finite-sample fluctuations from the test set of size $m$; $\epsilon_{\mathrm{method}}^{(\phi)}$ captures the method–specific contributions that depend on the surrogate choice $\phi \in \{\mathrm{DM}, \mathrm{GM}, \mathrm{TM}\}$. Here $C_{\phi,a}$

is a distillation method–dependent factor (possibly varying with configuration $a$) and $\mathcal{M}_\phi(\xi^{(0)})$ is the initial surrogate mismatch; $\epsilon_k^{(\phi)}$ represents the sampling error that decreases with the number of distilled samples $k$, uniformly across all three surrogates (The explicit form of $C_{\phi,a}$ and the detailed proof are deferred to the Appendix E.5).

**Theorem 6.5** (Coverage–aware bound with dynamic outer progress). *Let $\mathcal{A}$ denote the family of configurations, with prior $\Pi$ supported on a $\rho$–packing, and $a, b \in \mathcal{A}$. Under Assumption 5.1, with probability at least $1 - \varepsilon$,*

$$\sup_{a \in \mathcal{A}} \left| R_\nu(\theta_T^{(s,a)}) - R_\nu(\theta_T^{(\tau,a)}) \right| \leq \epsilon_{\text{bound}}^{\text{distillation}} + \epsilon_{\text{method}}^{(\phi)} + \epsilon_{k,\text{cov}}^{(\phi)}.$$

*Relative to Theorem 6.3, only two changes. First, the method term is controlled by the worst–case constant $C_{\phi,\max} = \sup_{a \in \mathcal{A}} C_{\phi,a}$ instead of the per–configuration factor $C_{\phi,a}$. Second, the sampling term acquires an explicit dependence on the coverage complexity, $\epsilon_{k,\text{cov}}^{(\phi)} = \tilde{\mathcal{O}}(\sqrt{\mathcal{H}_{\text{cov}}(\mathcal{A}, r)/k})$, in place of the $\tilde{\mathcal{O}}(1/\sqrt{k})$ rate for a single configuration (see Appendix E.6 for proof details).*

*Remark* 6.6 (Practical guidance). Theorems 6.3 and 6.5 decompose the generalization error into three parts: a method–independent irreducible bound, a method–dependent contraction term, and a sampling term that depends jointly on the configuration coverage $\mathcal{H}_{\text{cov}}(\mathcal{A}, r)$ and the distilled sample size $k$. Crucially, this analysis shows that despite their surface differences, GM, DM, and TM are governed by the same scaling law in single configurations and the same coverage law across configuration families, providing a unified set of design principles for practical dataset distillation. The generalization error can be reduced by tuning the surrogate and its hyperparameters to shrink the contraction term (e.g., GM with carefully chosen anchors, TM under stable dynamics, DM with an appropriate kernel), and by scaling $k$ in proportion to $\mathcal{H}_{\text{cov}}(\mathcal{A}, r)$ to control the sampling error.

# 7 EXPERIMENTS

We verify our theoretical results on MNIST (LeCun et al., 2002), CIFAR-10/100 (Krizhevsky et al., 2009), and ImageNette (Deng et al., 2009). We evaluate four canonical distillation families: *Gradient Matching* (DC (Zhao et al., 2020), DSA (Zhao & Bilen, 2021)), *Distribution Matching* (DM (Zhao & Bilen, 2023)), *Trajectory Matching* (MTT (Cazenavette et al., 2022)), and a recent *diffusion-based* method (MGD[3] (Chan-Santiago et al., 2025)). Each target configuration consists of an architecture (ConvNet, LeNet (LeCun et al., 2002), ResNet-18 (He et al., 2016), AlexNet (Krizhevsky et al., 2012)), optimizer (SGD/Adam), and augmentation (DSA on/off). The distilled dataset is always generated in the source configuration ConvNet+SGD, using the official open-source implementations released by the corresponding papers to ensure consistency and reproducibility across methods. We vary the number of distilled samples $k \in \{2, 4, 6, 8, 12, 18, 28, 51, 100, 200\}$, and report the generalization error $\Delta_a(\hat{\mu}_\tau, \hat{\mu}_s) = |\hat{R}(\theta_T^{(s,a)}) - \hat{R}(\theta_T^{(\tau,a)})|$, where $a$ indexes the target configuration. In practice, we approximate the coverage complexity $\mathcal{H}_{\text{cov}}(\mathcal{A}, r)$ by $\log M$, where $M$ is the number of sampled configurations.

**Single-configuration regime.** We first test the $1/\sqrt{k}$ scaling law (Theorem 6.3) on fixed $a$ is ConvNet+SGD. Figures 2 show that $\Delta_a$ decreases nearly linearly with $1/\sqrt{k}$, with coefficient of determination $R^2$ values mostly above 0.85, confirming the established scaling law. On *MNIST*, DC, DSA, and DM achieve almost perfect fits ($R^2 = 0.90$–$0.99$), while MTT exhibits a much steeper slope ($\beta_1 = -0.33$) with high variance, reflecting instability from trajectory unrolling. On *CIFAR-10*, all methods follow the law with strong fits ($R^2 = 0.86$–$0.96$); DC and DSA give the most stable slopes, showing that although the scaling law is universal, different methods vary in their sample efficiency, i.e., how effectively additional distilled samples reduce error, and in the stability of this improvement across configurations. On *CIFAR-100*, DC and DSA again maintain high linearity ($R^2 = 0.92$–$0.97$), but DM (0.84) and especially MTT (0.73) deviate more, highlighting how trajectory mismatch is amplified by dataset complexity. On *ImageNette*, the diffusion-based method also exhibits a clear $1/\sqrt{k}$ scaling ($R^2 = 0.87$–$0.92$), showing that the law extends beyond matching-based approaches. Together, these results verify that the $1/\sqrt{k}$ law holds universally (high $R^2$), while the intercept corresponds to dataset-specific irreducible generalization error bounds.

**Cross-configuration coverage.** We next test the coverage law $\Delta \propto \sqrt{\mathcal{H}_{\text{cov}}(\mathcal{A})/k}$ (Theorems 5.1–5.2). On *MNIST*, DC and DSA show clean linear fits with slopes $\beta_1 \approx 0.20$ ($R^2 = 0.88$–$0.92$), DM

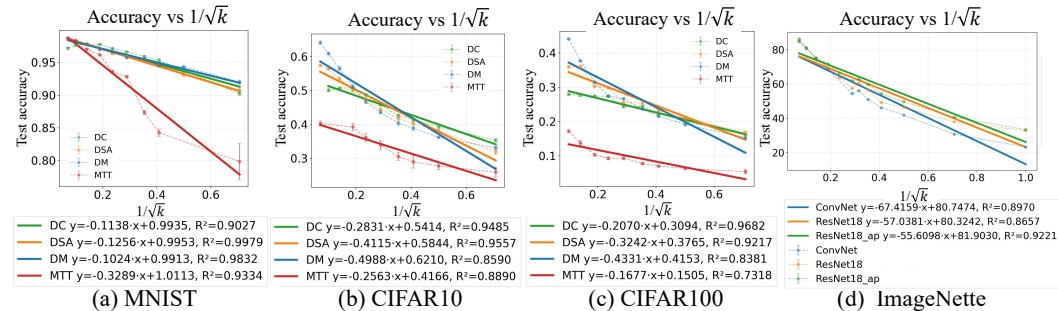

(a) MNIST  (b) CIFAR10  (c) CIFAR100  (d) ImageNette

Figure 2: Single-configuration scaling law. On MNIST, CIFAR-10/100, and ImageNette, the curves of generalization error $\Delta$ against $1/\sqrt{k}$ for GM, DM, and MTT shows linear decay at small $k$ followed by saturation at a positive generalization error bound. Regression intercepts give $\epsilon_{\text{bound}}(a)$, consistent with Theorems 4.2 and 6.3.

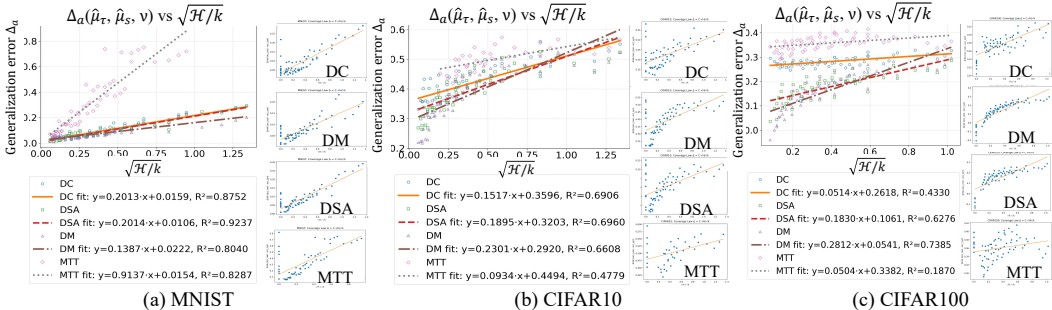

(a) MNIST  (b) CIFAR10  (c) CIFAR100

Figure 3: Configuration coverage law. For subsets of $m$ configurations, plotting $Y = \Delta(k, M)$ against $X = \sqrt{\log M}/\sqrt{k}$ yields near-linear trends under random sampling. Results remain consistent with Theorems 5.1 and 6.5.

follows with a smaller slope ($\beta_1 = 0.14$, $R^2 = 0.80$), while MTT rises to $\beta_1 = 0.91$ ($R^2 = 0.83$), indicating much higher sensitivity to configuration shifts. On *CIFAR-10*, the trend persists with moderate fits ($R^2 \approx 0.66$–$0.70$): slopes increase from MTT ($\beta_1 = 0.09$) $\rightarrow$ DC ($\beta_1 = 0.15$) $\rightarrow$ DSA ($\beta_1 = 0.19$) $\rightarrow$ DM ($\beta_1 = 0.23$). This ordering reflects the relative robustness of different objectives to configuration diversity. On *CIFAR-100*, the law remains visible but with weaker fits (DM: slope $\beta_1 = 0.28$, $R^2 = 0.74$; DSA: $\beta_1 = 0.18$, $R^2 = 0.63$; DC/MTT: $\beta_1 = 0.05$, $R^2 < 0.45$). The lower $R^2$ is expected: with more classes and higher intra-class variability, empirical estimates of cross-configuration errors become noisier, so points scatter more around the predicted line even though the linear scaling still holds. Across datasets, the slope acts as a method-dependent sensitivity to coverage complexity: GM methods achieve lower error within a configuration but degrade faster across configurations, DM is comparatively robust to diversity but depends more on $k$, and TM suffers the largest instability.

## 8 CONCLUSION AND LIMITATIONS

We propose the first configuration-dynamics–error framework that unifies gradient-, distribution-, and trajectory-matching distillation. Our analysis reveals two fundamental dataset distillation laws: 1) the scaling law, which explains that the generalization error reduces at a rate of $1/\sqrt{k}$ with the distilled sample size enlarge, and explain the saturation effect in IPC scaling; 2) the coverage law, showing that the distilled sample size must scale linearly with configuration diversity. These two laws characterize the utility boundary of data distillation, and offer a recipe for the theory-guided and configuration-robust methods design.

**Limitations.** Despite this advance, our theory relies on PL-type dynamics, which may break down in highly non-convex or adversarial training settings. The coverage measure of configuration diversity via the distance of update parameters may neglect semantic/domain shifts.

ETHICS STATEMENT

This work follows the ICLR Code of Ethics and the broader principles of responsible research. It does not involve human subjects, personal data, or other sensitive information that would require ethics approval. All datasets used are publicly available and properly licensed, and we cite their original sources. To support transparency and reproducibility, we release our implementation and describe our experimental settings in detail. The research was conducted without conflicts of interest or external sponsorships that could have influenced its design, execution, or reporting.

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

APPENDIX CONTENTS

## A  NOTATIONS

## B  RELATED WORK

### B.1  METHODOLOGICAL ADVANCES IN DATASET DISTILLATION

**Gradient Matching (GM).** Dataset distillation was first formulated via gradient matching, where synthetic data are optimized to align the training gradients of real data (Zhao et al., 2020). Differentiable Siamese augmentation improved stability and generalization across pipelines (Zhao & Bilen, 2021). Later works explored diversity regularization (Cazenavette et al., 2022), representative matching (DREAM) (Liu et al., 2023), and feature regression (FRePo) (Zhou et al., 2022). Engineering efforts such as DC-BENCH (Cui et al., 2022) provided standardized evaluation. Despite progress, GM often struggles with cross-architecture transfer.

**Distribution Matching (DM).** Distribution-based methods align feature or embedding distributions rather than raw gradients. Zhao and Bilen (Zhao & Bilen, 2023) proposed matching distributions through MMD in intermediate feature space. CAFe introduced hierarchical alignment (Wang et al.,

Table 2: Notations for preliminaries 2, configuration-dynamic-risk framework 3, single–configuration 4, and configurations coverage 5.

| Symbol | Meaning |
|---|---|
| **Data & Distributions** | |
| $D_\tau, q_\tau$ | Real training dataset and its population distribution. |
| $\hat{\mu}_\tau$ | Empirical measure of $D_\tau$. |
| $D_s, \hat{\mu}_s$ | Distilled dataset and its empirical/prototype measure (support size $k$). |
| $\mathcal{P}_k$ | Class of $k$-prototype measures (support on at most $k$ atoms). |
| $\nu, \hat{\nu}$ | Test distribution and its empirical counterpart of size $m$. |
| $z = (x, y)$ | A data point used in the loss $\ell(\theta; z)$. |
| **configurations & Geometry** | |
| $\mathcal{A}, \mathcal{C} \subseteq \mathcal{A}$ | Universe of training configurations; target subfamily for generalization. |
| $a \in \mathcal{A}$ | A concrete configuration (optimizer, hyperparameters, augmentation, architecture). |
| $\Gamma, \Gamma_a$ | Common feasible set; feasible set associated with configuration $a$. |
| $d_\mathcal{A}(a, a')$ | configuration-distance (update-field discrepancy between $a$ and $a'$). |
| $H_{\text{cov}}(r)$ | Coverage complexity: $\log \mathcal{N}(\mathcal{A}, d_\mathcal{A}, r)$. |
| $\Pi$ | Prior over configurations (e.g., uniform over a $\rho$-cover/packing). |
| **Losses & Risks** | |
| $\ell(\theta; z)$ | Per-sample loss. |
| $R_\nu(\theta)$ | Population risk under distribution $q$: $R_\nu = \mathbb{E}_\nu \, \ell(\theta; z)$. |
| $\hat{R}(\theta)$ | Empirical risk under $\hat{\nu}$. |
| **Dynamics & Regularity** | |
| $\theta_t, \theta_T$ | Parameters at step $t$ and after $T$ steps. |
| $\Phi_a(\theta; \mu)$ | Update map under configuration $a$ and data $\mu$: $\theta_{t+1} = \Phi_a(\theta_t; \mu)$. |
| $P_a(\theta)$ | Preconditioner/metric of configuration $a$ (e.g., adaptive gradient metric). |
| $g_a(\theta; z)$ | Per-sample update field used by configuration $a$. |
| $\eta$ | Inner-loop step size (learning rate). |
| $\rho_a \in (0, 1)$ | Contraction factor of $\Phi_a$ on $\Gamma$ (PL/smooth regime). |
| $C_{2,a} = \frac{1-\rho_a}{L_R}$ | Risk-to-alignment transfer constant ($L_R$: risk Lipschitz in $\theta$). |
| $\kappa_a$ | Uniform bound on $\|P_a(\theta)\|$ over $\Gamma$. |
| $B_g, B_\ell$ | Bounds on $\|g_a(\theta; z)\|$ and $|\ell(\theta; z)|$, respectively. |
| $\delta_0$ | Initialization mismatch: $\delta_0 = \theta_0^{(s)} - \theta_0^{(\tau)}$. |
| **Alignment & Aggregation** | |
| $\Delta_a(\mu, \nu)$ | Preconditioned alignment discrepancy between data $\mu$ and $\nu$ under $a$. |
| $\Delta_a^\star$ | Irreducible alignment error at budget $k$: $\inf_{\mu \in \mathcal{P}_k} \Delta_a(\hat{\mu}_\tau, \mu)$. |
| $\Delta_\sharp^\star$ | Aggregated irreducible error: $\inf_{\mu \in \mathcal{P}_k} \mathbb{E}_{a \sim \Pi} \Delta_a(\hat{\mu}_\tau, \mu)$. |
| **Complexities & Rates** | |
| $R_k(\mathcal{G}_a), R_n(\mathcal{G}_a)$ | Rademacher complexities of gradient classes at sizes $k$ and $n$. |
| $R_m(\mathcal{L}_a)$ | Rademacher complexity of the test-loss class at size $m$. |
| $e_g(n, k, \varepsilon), e_{\text{te}}(m, \varepsilon)$ | Estimation terms on training/test sides (confidence $\varepsilon$). |
| $\epsilon_{\text{bound}}$ | $k$-independent error floor (optimization + irreducible alignment + statistics). |
| $C_{\text{cov}}(\mathcal{A})$ | Coverage slope controlling the $\sqrt{H_{\text{cov}}/k}$ term. |
| **Information Terms** | |
| $I(D_s; D_\tau)$ | Mutual information between distilled and real training data. |
| $C_I, C_I'$ | Problem-dependent constants in MI-based corrections (avg./uniform forms). |

2022), while M3D (Zhang et al., 2024) and latent quantile matching (LQM) (Wei et al., 2024) further improved statistical robustness. DM methods emphasize scalability and robustness, with strong performance under augmentation shifts.

**Trajectory Matching (TM).** Trajectory matching aligns the full optimization dynamics rather than single-step gradients. The MTT framework (Cazenavette et al., 2022) established this principle, later

Table 3: Notations for unifying various branches of dataset distillation 6.

| Symbol | Meaning |
|---|---|
| **Distribution-matching (DM) branch** | |
| $L_{z,a}$ | Lipschitz constant of $g_a(\theta; z)$ with respect to the data metric $d_Z$. |
| $W_1$ | 1-Wasserstein distance $W_1(\hat{\mu}_s, \hat{\mu}_\tau)$. |
| $C_k$ | RKHS aggregate norm bound, $\sup_\theta \left( \sum_j \|g_{a,j}(\theta; \cdot)\|_{\mathcal{H}_k}^2 \right)^{1/2}$. |
| $\mathrm{MMD}_k$ | Maximum mean discrepancy with kernel $k$, measuring distributional distance in RKHS. |
| **Gradient-matching (GM) branch** | |
| $\|\Theta_j\|$ | Cardinality of the GM anchor set $\Theta_j$. |
| $\mathcal{M}_{\mathrm{GM}}$ | GM surrogate: anchor-averaged field gap $\frac{1}{\|\Theta_j\|} \sum_{\theta \in \Theta_j} \|\mathbb{E}_{\hat{\mu}_s} g_a - \mathbb{E}_{\hat{\mu}_\tau} g_a\|_2$. |
| **Trajectory-matching (TM) branch** | |
| $\omega_{\min}$ | Minimum TM weight, $\omega_{\min} := \min_t \omega_t > 0$. |
| $M_{\mathrm{TM}}$ | TM surrogate, $M_{\mathrm{TM}} = \sum_{t=0}^{L_b} \omega_t \|\theta_t^{(s,a)} - \theta_t^{(\tau,a)}\|_2$. |
| $\varepsilon_{\mathrm{path}}$ | Path coverage radius: maximum distance from an optimal point to the unrolled path set. |
| **Outer-loop / Branch-agnostic** | |
| $\varphi \in \{\mathrm{DM, GM, TM}\}$ | Distillation surrogate branch (distribution-, gradient-, or trajectory-matching). |
| $C_{\varphi,a}$ | Bridge constant for branch $\varphi$ under configuration $a$. |
| $\alpha_\varphi$ | Outer-loop contraction rate in the surrogate recursion Eq. (13). |
| $J$ | Number of outer-loop (bilevel) iterations. |
| $M_\varphi(\xi^{(0)})$ | Initial surrogate misfit at outer-loop initialization $\xi^{(0)}$. |
| $\epsilon_{\mathrm{est}}^{(\varphi)}$ | Estimation/statistical error floor for branch $\varphi$. |
| $\xi$ | Outer-loop optimization variables (e.g., prototypes, labels, or weights). |
| **Abbreviations** | |
| "DM/GM/TM" | Three surrogate types: distribution-matching, gradient-matching, and trajectory-matching. |

improved by truncated backpropagation (Kim et al., 2022) and automated trajectory design (Liu et al., 2024a). Extensions include self-supervised distillation (Zhou et al., 2024). TM captures optimization pathways faithfully but is computationally demanding.

**Beyond GM/DM/TM.** Generative priors constrain synthetic samples in pretrained generative latent spaces, e.g., GLaD (Cazenavette et al., 2023) and diffusion-based methods (Su et al., 2024). Large-scale efforts like SRe$^2$L (Yin et al., 2023) and TESLA (Cui et al., 2023) scale distillation to ImageNet-1K. Beyond images, condensation has been extended to graphs (Liu et al., 2024b; Zheng & Li, 2023) and adversarially robust regimes (Sun et al., 2024). Label-centric approaches argue that soft labels often dominate performance gains (Sucholutsky & Schonlau, 2021; Chen et al., 2023). These directions illustrate the versatility of dataset distillation across domains and modalities. *We do not directly evaluate generative-based methods in our experiments, since our focus is on canonical optimization-driven procedures (GM/DM/TM). Nevertheless, our theoretical framework is general and applies equally to settings where synthetic data are produced by generative priors or large-scale latent models, as the alignment–risk perspective only depends on the induced empirical distributions rather than the mechanism of synthesis.*

### B.2 THEORETICAL EXPLORATIONS IN DATASET DISTILLATION

**Kernel and NTK perspectives.** Several works analyze distillation under kernel ridge regression or NTK approximations. KIP (Nguyen et al., 2020) and its infinite-width extension (Nguyen et al., 2021) provided conditions for exact recovery. Random feature approximation (RFAD) improved scalability (Loo et al., 2022). Recent works extend these guarantees to provable bounds (Chen et al., 2024).

**Generalization and stability.** Classical results on uniform stability (Hardt et al., 2016), information-theoretic generalization (Xu & Raginsky, 2017; Bu et al., 2020), and Rademacher complexity (Bartlett & Mendelson, 2002) have been adapted to study synthetic datasets. Convexified implicit gradient methods (Wai et al., 2020) provide insights into bilevel optimization stability, directly relevant to GM and TM.

**Scaling laws and soft labels.** Empirical analyses consistently show saturation as the number of distilled samples (IPC) increases (Cazenavette et al., 2022; Zhao & Bilen, 2023). Soft-label studies (Sucholutsky & Schonlau, 2021; Chen et al., 2023) demonstrate Pareto frontiers relating IPC and generalization accuracy, suggesting the existence of an irreducible *alignment floor*.

**configuration and transfer.** Cross-architecture transferability remains a challenge: distilled datasets often fail under mismatched architectures or augmentations (Liu et al., 2023; Cazenavette et al., 2023). Large-scale efforts like SRe$^2$L (Yin et al., 2023) and TESLA (Cui et al., 2023), as well as generative priors (Cazenavette et al., 2023; Su et al., 2024), can be interpreted as attempts to reduce alignment floors or expand coverage. Design space analyses (Shao et al., 2024) and diversity–realism tradeoff studies (Sun et al., 2024) further quantify ecological robustness.

**Connection to our work.** Our configuration-dynamics–risk framework unifies GM, DM, and TM as alignment instances, and establishes both the *single-configuration scaling law* and the *coverage law*. This explains the observed saturation in IPC scaling and the degradation under configuration shifts, bridging empirical phenomena with provable guarantees.

## C PROOFS FOR THE SINGLE–CONFIGURATION BOUND

Recall the update rule for fixed training configuration $a$ with dataset distribution $\mu$ Eq. (3):

$$\theta_{t+1} = \Phi_a(\theta_t; \mu) = \theta_t - \eta\, P_a(\theta_t)\, \mathbb{E}_\mu\, g_a(\theta_t; z),$$

the empirical risks Eq. (1), and Assumption 4.1. Throughout, $\|\cdot\|$ denotes the Euclidean norm and the associated operator norm.

The distribution-aware discrepancy is defined are Eq. (5) as:

$$\Delta_a(\mu, \nu) := \sup_{\theta \in \Gamma_a} \left\| P_a(\theta)\big(\mathbb{E}_\mu g_a(\theta; z) - \mathbb{E}_\nu g_a(\theta; z)\big) \right\|_2.$$

Given a $k$-prototype class $\mathcal{P}_k$ and the real empirical distribution $\hat{\mu}_\tau$, the *intrinsic alignment error* is

$$\Delta_a^\star = \inf_{\mu \in \mathcal{P}_k} \Delta_a(\hat{\mu}_\tau, \mu).$$

### C.1 TECHNICAL LEMMAS USED IN THE PROOFS

**Lemma C.1** (Risk Lipschitz reduction to parameter mismatch). *Under Assumption 4.1, for any* $\theta, \theta' \in \Gamma_a$ *and any empirical test* $\hat{\nu}$,

$$\left| \hat{R}(\theta) - \hat{R}(\theta') \right| = \left| \mathbb{E}_{\hat{\nu}}\, \ell(\theta; z) - \mathbb{E}_{\hat{\nu}}\, \ell(\theta'; z) \right| \leq L_R\, \|\theta - \theta'\|.$$

*Proof.* Assumption 4.1 states that the test risk $R_\nu(\theta) = \mathbb{E}_\nu \ell(\theta; z)$ is $L_R$–Lipschitz on $\Gamma_a$. To obtain the same Lipschitz constant for *any* empirical measure $\hat{\nu}$ (including finite-support measures), we rely on the standard pointwise Lipschitz condition on the loss:

$$\forall z, \ \forall \theta, \theta' \in \Gamma_a: \quad |\ell(\theta; z) - \ell(\theta'; z)| \leq L_R\, \|\theta - \theta'\|. \tag{C.1.1}$$

Condition Eq. (C.1.1) is implied, e.g., by a uniform gradient bound $\sup_{\xi \in \Gamma_a} \|\nabla_\theta \ell(\xi; z)\| \leq L_R$ for all $z$ via the mean value theorem (e.g., Bubeck et al. 2015, Prop. B.10; Nesterov 2013, Chap. 2). It also immediately implies that every risk functional $R_\nu(\theta) = \mathbb{E}_\nu \ell(\theta; z)$—for any probability measure $\nu$ on $z$—is $L_R$–Lipschitz, since expectations preserve Lipschitz moduli (see Step 2 below). Thus, Assumption 4.1 plus Eq. (C.1.1) yields Lipschitzness uniformly over $\nu$, including $\nu = \hat{\nu}$.

By the definition $\hat{R}(\theta) = \mathbb{E}_{\hat{\nu}} \ell(\theta; z) = \frac{1}{m} \sum_{\ell=1}^m \ell(\theta; z_\ell^{\text{te}})$,

$$\left| \hat{R}(\theta) - \hat{R}(\theta') \right| = \left| \frac{1}{m} \sum_{\ell=1}^m \left( \ell(\theta; z_\ell^{\text{te}}) - \ell(\theta'; z_\ell^{\text{te}}) \right) \right|.$$

Apply triangle inequality to the finite sum (equivalently, linearity of expectation and Jensen's inequality for $|\cdot|$):

$$\left| \frac{1}{m} \sum_{\ell=1}^{m} \left( \ell(\theta; z_\ell^{\mathrm{te}}) - \ell(\theta'; z_\ell^{\mathrm{te}}) \right) \right| \leq \frac{1}{m} \sum_{\ell=1}^{m} \left| \ell(\theta; z_\ell^{\mathrm{te}}) - \ell(\theta'; z_\ell^{\mathrm{te}}) \right| = \mathbb{E}_{\hat{\nu}} \left| \ell(\theta; z) - \ell(\theta'; z) \right|.$$

Using Eq. (C.1.1) inside the expectation,

$$\mathbb{E}_{\hat{\nu}} \left| \ell(\theta; z) - \ell(\theta'; z) \right| \leq \mathbb{E}_{\hat{\nu}} \left( L_R \|\theta - \theta'\| \right) = L_R \|\theta - \theta'\|,$$

since $\|\theta - \theta'\|$ does not depend on $z$.

Combining all above equtions,

$$\left| \hat{R}(\theta) - \hat{R}(\theta') \right| \leq L_R \|\theta - \theta'\|,$$

which is the desired inequality. □

**Lemma C.2** (One-step decomposition: contraction + two-sample drift). *Fix an configuration $a$ and a step size $\eta \in (0, 2/L)$. Let $\delta_t := \theta_t^{(s)} - \theta_t^{(\tau)}$ denote the parameter gap after $t$ iterations when training respectively on $\hat{\mu}_s$ and $\hat{\mu}_\tau$. Under Assumption 4.1, we have*

$$\|\delta_{t+1}\| = \left\| \Phi_a(\theta_t^{(s)}; \hat{\mu}_s) - \Phi_a(\theta_t^{(\tau)}; \hat{\mu}_\tau) \right\| \leq \rho_a \|\delta_t\| + \eta\, \kappa_a \sup_{\theta \in \Gamma_a} \left\| \mathbb{E}_{\hat{\mu}_s} g_a(\theta; z) - \mathbb{E}_{\hat{\mu}_\tau} g_a(\theta; z) \right\|.$$

*Proof.* By definition of the update map Eq. (3),

$$\theta_{t+1}^{(s)} = \Phi_a(\theta_t^{(s)}; \hat{\mu}_s), \qquad \theta_{t+1}^{(\tau)} = \Phi_a(\theta_t^{(\tau)}; \hat{\mu}_\tau).$$

Hence

$$\delta_{t+1} = \Phi_a(\theta_t^{(s)}; \hat{\mu}_s) - \Phi_a(\theta_t^{(\tau)}; \hat{\mu}_\tau).$$

Insert and subtract $\Phi_a(\theta_t^{(\tau)}; \hat{\mu}_s)$, then apply the triangle inequality:

$$\|\delta_{t+1}\| \leq \underbrace{\left\| \Phi_a(\theta_t^{(s)}; \hat{\mu}_s) - \Phi_a(\theta_t^{(\tau)}; \hat{\mu}_s) \right\|}_{\text{same data distribution}} + \underbrace{\left\| \Phi_a(\theta_t^{(\tau)}; \hat{\mu}_s) - \Phi_a(\theta_t^{(\tau)}; \hat{\mu}_\tau) \right\|}_{\text{same parameter}}. \tag{C.2.1}$$

Assumption 4.1 states that for any fixed $\mu$ the update map is contractive with factor $\rho_a \in (0, 1)$:

$$\|\Phi_a(\theta; \mu) - \Phi_a(\theta'; \mu)\| \leq \rho_a \|\theta - \theta'\|.$$

This is the standard contraction property of gradient descent under smoothness and strong convexity (see Nesterov 2013, theorem 2.1.15; Bubeck et al. 2015, Prop. 3.5) or under the PL condition (see Karimi et al. 2016, theorem 1). Applying it with $\mu = \hat{\mu}_s$ and $(\theta, \theta') = (\theta_t^{(s)}, \theta_t^{(\tau)})$ gives

$$\|\Phi_a(\theta_t^{(s)}; \hat{\mu}_s) - \Phi_a(\theta_t^{(\tau)}; \hat{\mu}_s)\| \leq \rho_a \|\theta_t^{(s)} - \theta_t^{(\tau)}\| = \rho_a \|\delta_t\|.$$

For the second term,

$$\Phi_a(\theta_t^{(\tau)}; \hat{\mu}_s) - \Phi_a(\theta_t^{(\tau)}; \hat{\mu}_\tau) = -\eta\, P_a(\theta_t^{(\tau)}) \Big( \mathbb{E}_{\hat{\mu}_s} g_a(\theta_t^{(\tau)}; z) - \mathbb{E}_{\hat{\mu}_\tau} g_a(\theta_t^{(\tau)}; z) \Big).$$

Taking norms and using submultiplicativity, together with the bound $\|P_a(\theta)\| \leq \kappa_a$, yields

$$\|\Phi_a(\theta_t^{(\tau)}; \hat{\mu}_s) - \Phi_a(\theta_t^{(\tau)}; \hat{\mu}_\tau)\| \leq \eta\, \kappa_a \sup_{\theta \in \Gamma_a} \left\| \mathbb{E}_{\hat{\mu}_s} g_a(\theta; z) - \mathbb{E}_{\hat{\mu}_\tau} g_a(\theta; z) \right\|.$$

Substituting the two estimates back into the decomposition from Eq (C.2.1) gives

$$\|\delta_{t+1}\| \leq \rho_a \|\delta_t\| + \eta\, \kappa_a \sup_{\theta \in \Gamma_a} \left\| \mathbb{E}_{\hat{\mu}_s} g_a(\theta; z) - \mathbb{E}_{\hat{\mu}_\tau} g_a(\theta; z) \right\|,$$

which is exactly the claimed bound. □

**Lemma C.3** (Unrolling the recursion). *Let $u_{t+1} \leq \rho \, u_t + c$ with $\rho \in (0,1)$, $u_0 \geq 0$, $c \geq 0$. Then $u_T \leq \rho^T u_0 + \frac{c}{1-\rho}$.*

*Proof.* Standard geometric series: $u_1 \leq \rho u_0 + c$, $u_2 \leq \rho^2 u_0 + \rho c + c$, ... , $u_T \leq \rho^T u_0 + c(1 + \rho + \cdots + \rho^{T-1}) \leq \rho^T u_0 + \frac{c}{1-\rho}$. $\qquad\square$

**Lemma C.4** (Two-sample uniform deviation via Rademacher complexity). *Let $\mathcal{G}_a := \{ z \mapsto g_a(\theta; z) : \theta \in \Gamma_a \}$ be a vector-valued class with $\|g_a(\theta; z)\| \leq B_g$ for all $z$ and $\theta$. For independent samples $\mathcal{D}_s = \{z_i\}_{i=1}^k \sim \hat{\mu}_s$ and $\mathcal{D}_\tau = \{z_j'\}_{j=1}^n \sim \hat{\mu}_\tau$, with probability at least $1 - \varepsilon$,*

$$\sup_{\theta \in \Gamma_a} \left\| \mathbb{E}_{\hat{\mu}_s} g_a(\theta; z) - \mathbb{E}_{\hat{\mu}_\tau} g_a(\theta; z) \right\| \leq \Delta_a^\star + 2(\mathfrak{R}_k(\mathcal{G}_a) + \mathfrak{R}_n(\mathcal{G}_a)) + B_g \sqrt{2 \log \tfrac{4}{\varepsilon}} \left( \tfrac{1}{\sqrt{k}} + \tfrac{1}{\sqrt{n}} \right).$$

*Proof.*

**Step C.4.0 Scalarization and Rademacher conventions.** Let $\| \cdot \|_*$ be the dual norm of $\| \cdot \|$. For any signed measure $\nu$ and any $\theta$,

$$\left\| \mathbb{E}_\nu g_a(\theta; z) \right\| = \sup_{\|v\|_* \leq 1} \left\langle v, \mathbb{E}_\nu g_a(\theta; z) \right\rangle = \sup_{\|v\|_* \leq 1} \mathbb{E}_\nu \left\langle v, g_a(\theta; z) \right\rangle.$$

Hence all vector deviations can be reduced to a scalar class

$$\mathcal{F}_a := \left\{ f_{\theta,v}(z) := \langle v, g_a(\theta; z) \rangle \,\middle|\, \theta \in \Gamma_a,\, \|v\|_* \leq 1 \right\}, \qquad |f_{\theta,v}(z)| \leq B_g.$$

We use the (empirical) Rademacher complexity with the factor "2" built in (Mohri et al. 2018, Def. 3.1): for a sample $U = (z_1, \ldots, z_m)$,

$$\widehat{\mathfrak{R}}_U(\mathcal{F}_a) := \mathbb{E}_\sigma \left[ \frac{2}{m} \sup_{f \in \mathcal{F}_a} \sum_{i=1}^m \sigma_i f(z_i) \right], \qquad \mathfrak{R}_m(\mathcal{F}_a) := \mathbb{E}_U \widehat{\mathfrak{R}}_U(\mathcal{F}_a).$$

Under this normalization, the standard uniform deviation bound (Mohri et al. 2018, theorem 3.3; see also Bartlett & Mendelson 2002; Xu et al. 2016) reads: for any $\varepsilon \in (0, 1)$, with probability at least $1 - \varepsilon$,

$$\sup_{f \in \mathcal{F}_a} \left| \mathbb{E}f - \mathbb{E}_{\hat{U}} f \right| \leq \widehat{\mathfrak{R}}_U(\mathcal{F}_a) + B_g \sqrt{\frac{\log(2/\varepsilon)}{2m}}. \tag{15}$$

(The $\widehat{\mathfrak{R}}_U$ term can be further upper bounded by $\mathfrak{R}_m(\mathcal{F}_a)$ in expectation; we will retain $\mathfrak{R}_m(\cdot)$ notation below. A direct treatment of the vector norm via the vector-contraction inequality Maurer, 2016, theorem 3 yields the same dependence on the scalarized class.)

**Step C.4.1 Four-point decomposition with a free $k$-prototype.** Fix any $\mu \in \mathcal{P}_k$. For every $\theta$,

$$\mathbb{E}_{\hat{\mu}_s} g_a(\theta) - \mathbb{E}_{\hat{\mu}_\tau} g_a(\theta) = \underbrace{(\mathbb{E}_{\hat{\mu}_s} - \mathbb{E}_{\mu_s}) g_a(\theta)}_{\text{empirical vs. population on } S} + \underbrace{(\mathbb{E}_{\mu_s} - \mathbb{E}_\mu) g_a(\theta)}_{\text{bridge within } \mathcal{P}_k}$$
$$+ \underbrace{(\mathbb{E}_\mu - \mathbb{E}_{\mu_\tau}) g_a(\theta)}_{\text{alignment to population } T} + \underbrace{(\mathbb{E}_{\mu_\tau} - \mathbb{E}_{\hat{\mu}_\tau}) g_a(\theta)}_{\text{empirical vs. population on } T}.$$

Taking $\| \cdot \|$ and the supremum over $\theta \in \Gamma_a$, then applying the triangle inequality,

$$\sup_\theta \left\| \mathbb{E}_{\hat{\mu}_s} g_a(\theta) - \mathbb{E}_{\hat{\mu}_\tau} g_a(\theta) \right\| \leq \underbrace{\sup_\theta \left\| \mathbb{E}_{\hat{\mu}_s} g_a(\theta) - \mathbb{E}_{\mu_s} g_a(\theta) \right\|}_{=: \Delta_S} + \underbrace{\sup_\theta \left\| \mathbb{E}_{\mu_s} g_a(\theta) - \mathbb{E}_\mu g_a(\theta) \right\|}_{=: B(\mu)}$$
$$+ \underbrace{\sup_\theta \left\| \mathbb{E}_\mu g_a(\theta) - \mathbb{E}_{\mu_\tau} g_a(\theta) \right\|}_{=: A_{\text{pop}}(\mu)} + \underbrace{\sup_\theta \left\| \mathbb{E}_{\hat{\mu}_\tau} g_a(\theta) - \mathbb{E}_{\mu_\tau} g_a(\theta) \right\|}_{=: \Delta_T}.$$
$$\tag{16}$$

Choose $\mu = \mu_s \in \mathcal{P}_k$ (the population behind the $k$-prototype), so that $B(\mu_s) = 0$. Define the intrinsic alignment floor against the *population* target as

$$\Delta_a^\star := \inf_{\mu \in \mathcal{P}_k} \Delta_a(\mu, \mu_\tau) = \inf_{\mu \in \mathcal{P}_k} \sup_{\theta \in \Gamma_a} \left\| \mathbb{E}_\mu g_a(\theta) - \mathbb{E}_{\mu_\tau} g_a(\theta) \right\|.$$

Then from Eq. (16),

$$\sup_\theta \left\| \mathbb{E}_{\hat{\mu}_s} g_a(\theta) - \mathbb{E}_{\hat{\mu}_\tau} g_a(\theta) \right\| \leq \Delta_S + \Delta_a^\star + \Delta_T. \tag{17}$$

**Step C.4.2 Bound $\Delta_S$ by Rademacher complexity.** By duality and scalarization (the class is symmetric so "$\sup(\cdot)$" equals "$\sup | \cdot |$"),

$$\Delta_S = \sup_\theta \sup_{\|v\|_* \leq 1} \left| \mathbb{E}_{\hat{\mu}_s} \langle v, g_a(\theta; z) \rangle - \mathbb{E}_{\mu_s} \langle v, g_a(\theta; z) \rangle \right| = \sup_{f \in \mathcal{F}_a} \left| \mathbb{E}_{\hat{\mu}_s} f - \mathbb{E}_{\mu_s} f \right|.$$

Applying Eq. (15) with $m = k$ and failure probability $\varepsilon/2$ and using $|f| \leq B_g$,

$$\Delta_S \leq \widehat{\mathfrak{R}}_S(\mathcal{F}_a) + B_g \sqrt{\frac{\log(4/\varepsilon)}{2k}} \leq \mathfrak{R}_k(\mathcal{F}_a) + B_g \sqrt{\frac{\log(4/\varepsilon)}{2k}}. \tag{18}$$

**Step C.4.3 Bound $\Delta_T$ analogously.** A symmetric argument for $T$ with size $n$ yields, with probability $\geq 1 - \varepsilon/2$,

$$\Delta_T = \sup_{f \in \mathcal{F}_a} \left| \mathbb{E}_{\hat{\mu}_\tau} f - \mathbb{E}_{\mu_\tau} f \right| \leq \widehat{\mathfrak{R}}_T(\mathcal{F}_a) + B_g \sqrt{\frac{\log(4/\varepsilon)}{2n}} \leq \mathfrak{R}_n(\mathcal{F}_a) + B_g \sqrt{\frac{\log(4/\varepsilon)}{2n}}. \tag{19}$$

**Step C.4.4 Combining all and applying union bound.** Combining Eq. (17), Eq. (18), and Eq. (19), and taking a union bound over the two events (each failing with prob. $\leq \varepsilon/2$), we obtain that with probability at least $1 - \varepsilon$,

$$\sup_\theta \left\| \mathbb{E}_{\hat{\mu}_s} g_a(\theta) - \mathbb{E}_{\hat{\mu}_\tau} g_a(\theta) \right\| \leq \Delta_a^\star + \widehat{\mathfrak{R}}_S(\mathcal{F}_a) + \widehat{\mathfrak{R}}_T(\mathcal{F}_a) + B_g \sqrt{\frac{\log(4/\varepsilon)}{2}} \left( \frac{1}{\sqrt{k}} + \frac{1}{\sqrt{n}} \right).$$

Replacing the empirical complexities by their expectations gives a sample-independent version with $\mathfrak{R}_k(\mathcal{F}_a) + \mathfrak{R}_n(\mathcal{F}_a)$. If one uses the "no-2" normalization for Rademacher complexity (Mohri et al., 2018), the bound incurs the standard extra factor 2 in front of $\mathfrak{R}_m^{\text{std}}(\mathcal{F}_a)$. $\qquad \square$

**Lemma C.5** (Test generalization bound). *Let $\mathcal{L}_a := \{z \mapsto \ell(\theta; z) : \theta \in \Gamma_a\}$ with $|\ell(\theta; z)| \leq B_\ell$. For i.i.d. test sample $\hat{\nu}$ of size $m$ from $q$, with probability at least $1 - \varepsilon$,*

$$\sup_{\theta \in \Gamma_a} \left| R_\nu(\theta) - \hat{R}(\theta) \right| \leq 2\, \mathfrak{R}_m(\mathcal{L}_a) + B_\ell \sqrt{\frac{2\log(4/\varepsilon)}{m}}.$$

*Proof.* Standard empirical process bound via symmetrization and concentration (Mohri et al. 2018, theorem 3.3; see also Bartlett & Mendelson 2002).

**Step C.5.1 Setup.** Define

$$\Psi(U) := \sup_{f \in \mathcal{L}_a} \left| \mathbb{E}_\nu f - \mathbb{E}_{\hat{U}} f \right|, \quad \mathbb{E}_{\hat{U}} f = \frac{1}{m} \sum_{i=1}^m f(Z_i).$$

By symmetry,

$$\Psi(U) \leq \sup_f (\mathbb{E}_\nu f - \mathbb{E}_{\hat{U}} f) + \sup_f (\mathbb{E}_{\hat{U}} f - \mathbb{E}_\nu f),$$

so it suffices to control $\Phi(U) := \sup_{f \in \mathcal{L}_a} (\mathbb{E}_\nu f - \mathbb{E}_{\hat{U}} f)$.

**Step C.5.2 Symmetrization.** Introduce an independent "ghost sample" $U' = (Z_1', \ldots, Z_m') \sim q^m$. Since $\mathbb{E}_\nu f = \mathbb{E}_{U'} \mathbb{E}_{\hat{U}'} f$, by Jensen

$$\mathbb{E}_U[\Phi(U)] \leq \mathbb{E}_{U,U'} \left[ \sup_{f \in \mathcal{L}_a} \frac{1}{m} \sum_{i=1}^m \left( f(Z_i') - f(Z_i) \right) \right].$$

Adding Rademacher variables $\sigma_i \in \{\pm 1\}$ and applying the triangle inequality yields

$$\mathbb{E}_U[\Phi(U)] \leq 2\, \mathbb{E}_U \mathbb{E}_\sigma \left[ \frac{1}{m} \sup_{f \in \mathcal{L}_a} \sum_{i=1}^m \sigma_i f(Z_i) \right] = 2\, \mathfrak{R}_m(\mathcal{L}_a).$$

**Step C.5.3 Concentration.** Replacing a single sample point $Z_i$ by $\tilde{Z}_i$ changes $\Phi(U)$ by at most $2B_\ell/m$, since $|f(z)| \leq B_\ell$. Hence $\Phi(U)$ satisfies the bounded-differences condition, and McDiarmid's inequality gives

$$\Pr\{\Phi(U) - \mathbb{E}\Phi(U) \geq t\} \leq \exp\left(-\frac{mt^2}{2B_\ell^2}\right).$$

Choosing $t = B_\ell\sqrt{\frac{2\log(2/\delta)}{m}}$ yields, with probability $\geq 1 - \delta$,

$$\Phi(U) \leq \mathbb{E}\Phi(U) + B_\ell\sqrt{\frac{2\log(2/\delta)}{m}}.$$

**Step C.5.4 Combining all.** Substituting $\mathbb{E}\Phi(U) \leq 2\mathfrak{R}_m(\mathcal{L}_a)$ from Step 2, and repeating the argument for $\sup_f(\mathbb{E}_{\hat{U}}f - \mathbb{E}_\nu f)$, a union bound with $\delta = \varepsilon/2$ gives

$$\sup_{f\in\mathcal{L}_a}\left|\mathbb{E}_\nu f - \mathbb{E}_{\hat{U}}f\right| \leq 2\mathfrak{R}_m(\mathcal{L}_a) + B_\ell\sqrt{\frac{2\log(4/\varepsilon)}{m}}$$

with probability at least $1 - \varepsilon$. Replacing $f$ by $\ell(\theta;\cdot)$ completes the proof. $\square$

**Lemma C.6** (Information-corrected two-sample deviation). *If the distilled dataset $\mathcal{D}_s$ depends on $\mathcal{D}_\tau$, then with probability at least $1 - \varepsilon$,*

$$\sup_{\theta\in\Gamma_a}\left\|\mathbb{E}_{\hat{\mu}_s}g_a(\theta;z) - \mathbb{E}_{\hat{\mu}_\tau}g_a(\theta;z)\right\| \leq \Delta_a^\star + 2(\mathfrak{R}_k(\mathcal{G}_a) + \mathfrak{R}_n(\mathcal{G}_a)) + B_g\sqrt{2\log\frac{8}{\varepsilon}}\left(\frac{1}{\sqrt{k}} + \frac{1}{\sqrt{n}}\right)$$

$$+ C_I\sqrt{\frac{I(\mathcal{D}_s;\mathcal{D}_\tau) + \log\frac{8}{\varepsilon}}{k}}.$$

*Proof.* **Mutual-information (MI) tail inequality.** The only difference of this proof compared with lemma C.4 is that we assumed istilled dataset $\mathcal{D}_s$ depends on $\mathcal{D}_\tau$. Thus, we apply a high-probability MI generalization bound (Bu et al. 2020, theorem 7; cf. Xu & Raginsky 2017, theorem 1, Steinke & Zakynthinou 2020, theorem 1): for any $\varepsilon_{\mathrm{mi}} \in (0,1)$, with probability at least $1 - \varepsilon_{\mathrm{mi}}$,

$$\phi(S,\mathcal{D}_\tau) \leq \mathbb{E}\left[\phi(S,\mathcal{D}_\tau)\mid\mathcal{D}_\tau\right] + \sqrt{2\sigma_k^2\left(I(S;\mathcal{D}_\tau) + \log\frac{1}{\varepsilon_{\mathrm{mi}}}\right)}. \tag{20}$$

Because $S$ is (possibly randomized) post-processing of $\mathcal{D}_s$, data processing yields $I(S;\mathcal{D}_\tau) \leq I(\mathcal{D}_s;\mathcal{D}_\tau)$. Combining Eq. (18) Eq. (20), and the assumption $k << n$ gives

$$\Delta_S \leq \mathfrak{R}_k(\mathcal{G}_a) + \sqrt{\frac{2B_g^2}{k}\left(I(\mathcal{D}_s;\mathcal{D}_\tau) + \log\frac{1}{\varepsilon_{\mathrm{mi}}}\right)} \leq \mathfrak{R}_k(\mathcal{G}_a) + C_I\sqrt{\frac{I(\mathcal{D}_s;\mathcal{D}_\tau) + \log\frac{1}{\varepsilon_{\mathrm{mi}}}}{k}}, \tag{21}$$

where $C_I := \sqrt{2}\,B_g$ (or a slightly larger universal constant to absorb scalarization).

**Adding the empirical fluctuation term.** Independently, the usual (conditional) concentration around the conditional mean yields, for any $\varepsilon_S \in (0,1)$, with probability at least $1 - \varepsilon_S$,

$$\Delta_S \leq \mathfrak{R}_k(\mathcal{G}_a) + B_g\sqrt{\frac{\log(2/\varepsilon_S)}{2k}}. \tag{22}$$

A union bound over Eq. (21) and Eq. (22) will then provide both terms simultaneously.

Choose

$$\varepsilon_T = \varepsilon_S = \varepsilon_{\mathrm{mi}} = \varepsilon/4,$$

and apply a union bound (note independence is not required for a union bound). We obtain, with probability at least $1 - \varepsilon$,

$$\sup_\theta\left\|\mathbb{E}_{\hat{\mu}_s}g_a(\theta) - \mathbb{E}_{\hat{\mu}_\tau}g_a(\theta)\right\| \leq \Delta_a^\star + \underbrace{(\mathfrak{R}_k(\mathcal{G}_a) + \mathfrak{R}_n(\mathcal{G}_a))}_{\text{rad. complexities}}$$

$$+ \underbrace{B_g\sqrt{\frac{\log(8/\varepsilon)}{2}}\left(\frac{1}{\sqrt{k}} + \frac{1}{\sqrt{n}}\right)}_{\text{empirical concentration}} + \underbrace{C_I\sqrt{\frac{I(\mathcal{D}_s;\mathcal{D}_\tau) + \log(4/\varepsilon)}{k}}}_{\text{MI correction}}.$$

Switching to the common Rademacher normalization with the factor 2 (as in the lemma statement) gives $2\big(\mathfrak{R}_k(\mathcal{G}_a) + \mathfrak{R}_n(\mathcal{G}_a)\big)$, and tightening constants in the Hoeffding terms leads to $B_g\sqrt{2\log(8/\varepsilon)}\big(\frac{1}{\sqrt{k}} + \frac{1}{\sqrt{n}}\big)$. Finally, we relax the MI denominator to $\min\{n,k\}$ using $\frac{1}{\sqrt{k}} \leq \frac{\sqrt{2}}{\sqrt{\min\{n,k\}}}$ and absorb $\sqrt{2}$ into $C_I$, which yields exactly

$$\sup_{\theta \in \Gamma_a} \big\|\mathbb{E}_{\hat{\mu}_s} g_a(\theta; z) - \mathbb{E}_{\hat{\mu}_\tau} g_a(\theta; z)\big\| \leq \Delta_a^\star + 2(\mathfrak{R}_k(\mathcal{G}_a) + \mathfrak{R}_n(\mathcal{G}_a)) + B_g\sqrt{2\log\tfrac{8}{\varepsilon}}\Big(\tfrac{1}{\sqrt{k}} + \tfrac{1}{\sqrt{n}}\Big)$$

$$+ C_I\sqrt{\frac{I(\mathcal{D}_s; \mathcal{D}_\tau) + \log\frac{8}{\varepsilon}}{k}}.$$

$\square$

## C.2 PROOF OF THEOREM 4.2

*Proof of Theorem 4.2.*

**Step C.2.1 Parameter gap.** Apply Lemma C.2 with $c := \eta\,\kappa_a\,\sup_\theta \|\mathbb{E}_{\hat{\mu}_s} g_a - \mathbb{E}_{\hat{\mu}_\tau} g_a\|$ and $q := \rho_a \in (0,1)$, then Lemma C.3 gives

$$\|\delta_T\| \leq \rho_a^T \|\delta_0\| + \frac{\eta\,\kappa_a}{1-\rho_a} \sup_{\theta \in \Gamma_a} \big\|\mathbb{E}_{\hat{\mu}_s} g_a(\theta; z) - \mathbb{E}_{\hat{\mu}_\tau} g_a(\theta; z)\big\|. \tag{23}$$

**Step C.2.2 Risk gap.** By Lemma C.1,

$$\big|\hat{R}(\theta_T^{(s)}) - \hat{R}(\theta_T^{(\tau)})\big| \leq L_R \|\delta_T\|. \tag{24}$$

Combine Eq. (23)–(24) and recall $C_{2,a} = \frac{1-\rho_a}{L_R}$:

$$\big|\hat{R}(\theta_T^{(s)}) - \hat{R}(\theta_T^{(\tau)})\big| \leq L_R\rho_a^T\|\delta_0\| + \frac{\eta\,\kappa_a L_R}{1-\rho_a} \sup_\theta \big\|\mathbb{E}_{\hat{\mu}_s} g_a - \mathbb{E}_{\hat{\mu}_\tau} g_a\big\| \tag{25}$$

**Step C.2.3 Two-sample discrepancy bound.** Apply Lemma C.4 with probability $\geq 1 - \varepsilon/2$:

$$\sup_\theta \big\|\mathbb{E}_{\hat{\mu}_s} g_a - \mathbb{E}_{\hat{\mu}_\tau} g_a\big\| \leq \Delta_a^\star + 2(\mathfrak{R}_k(\mathcal{G}_a) + \mathfrak{R}_n(\mathcal{G}_a)) + B_g\sqrt{2\log\tfrac{4}{\varepsilon}}\Big(\tfrac{1}{\sqrt{k}} + \tfrac{1}{\sqrt{n}}\Big).$$

Plug this into Eq. (25) and regroup the terms as

$$\frac{\eta\,\kappa_a}{C_{2,a}}\left[\Delta_a^\star + 2(\mathfrak{R}_k(\mathcal{G}_a) + \mathfrak{R}_n(\mathcal{G}_a)) + B_g\sqrt{2\log\tfrac{4}{\varepsilon}}\Big(\tfrac{1}{\sqrt{k}} + \tfrac{1}{\sqrt{n}}\Big)\right].$$

Absorb $\eta$ into the (fixed) constants inside the complexity term by defining

$$e_g(n, k, \varepsilon) := 2\big(\mathfrak{R}_k(\mathcal{G}_a) + \mathfrak{R}_n(\mathcal{G}_a)\big) + B_g\sqrt{2\log\tfrac{4}{\varepsilon}}\Big(\tfrac{1}{\sqrt{k}} + \tfrac{1}{\sqrt{n}}\Big),$$

Thus,

$$\big|\hat{R}(\theta_T^{(s)}) - \hat{R}(\theta_T^{(\tau)})\big| \leq L_R\rho_a^T\|\delta_0\| + \frac{\eta\kappa_a}{C_{2,a}}\Big(\Delta_a^\star + e_g(n, k, \varepsilon)\Big).$$

**Step C.2.4 Test generalization.** Using Lemma C.5,

$$\big|R_\nu(\theta) - \hat{R}(\theta)\big| \leq e_{\text{te}}(m, \varepsilon) := 2\,\mathfrak{R}_m(\mathcal{L}_a) + B_\ell\sqrt{\tfrac{2\log(4/\varepsilon)}{m}}.$$

A union bound over the training part and the test part yields the bound stated in Theorem 4.2 with probability at least $1 - \varepsilon$.

**Information-corrected variant.** Replace Lemma C.4 by Lemma C.6 and repeat the steps above; this introduces the additional $C_I\sqrt{\tfrac{I(\mathcal{D}_s; \mathcal{D}_\tau) + \log(8/\varepsilon)}{k}}$ term, as claimed. $\square$

### C.3 PROOF OF COROLLARY 4.4 (COMPLEXITY CONSEQUENCES)

*Proof of Corollary 4.4.* Start from Theorem 4.2:

$$\left|\hat{R}(\theta_T^{(s)}) - \hat{R}(\theta_T^{(\tau)})\right| \leq L_R \rho_a^T \|\delta_0\| + \frac{\eta \kappa_a}{C_{2,a}}(\Delta_a^\star + e_g(n,k,\varepsilon)) + e_{\text{te}}(m,\varepsilon).$$

Fix a target $\epsilon_0 > 0$ and an error split $(\beta_0, \beta_1, \beta_{\text{te}})$ with $\sum \beta = 1$. It suffices that each term is $\leq$ its budget:

$$L_R \rho_a^T \|\delta_0\| \leq \beta_0 \epsilon_0, \qquad \frac{\eta \kappa_a}{C_{2,a}}(\Delta_a^\star + e_g(n,k,\varepsilon)) \leq \beta_1 \epsilon_0, \qquad e_{\text{te}}(m,\varepsilon) \leq \beta_{\text{te}} \epsilon_0.$$

**Iterations** $T$. Assume the PL-type decay $\rho_a^T \leq C_{\text{opt}}/T^{\beta_{\text{opt}}}$ (Karimi et al. 2016, Theorem 1). To enforce $L_R \rho_a^T \|\delta_0\| \leq \beta_0 \epsilon_0$, it suffices that

$$\frac{C_{\text{opt}}}{T^{\beta_{\text{opt}}}} \leq \frac{\beta_0 \epsilon_0}{L_R \|\delta_0\|}, \quad \Longleftrightarrow \quad T \geq \left(\frac{L_R \|\delta_0\| C_{\text{opt}}}{\beta_0 \epsilon_0}\right)^{1/\beta_{\text{opt}}}.$$

**Distilled samples** $k$. Under standard rates for Rademacher complexity (Bartlett & Mendelson 2002; Xu et al. 2016; Mohri et al. 2018), assume there exist $C_g, C_{\text{te}}$ such that $\mathfrak{R}_k(\mathcal{G}_a) \leq C_g/\sqrt{k}$, $\mathfrak{R}_n(\mathcal{G}_a) \leq C_g/\sqrt{n}$. Then

$$e_g(n,k,\varepsilon) \leq 2\left(\frac{C_g}{\sqrt{k}} + \frac{C_g}{\sqrt{n}}\right) + B_g\sqrt{2\log\frac{4}{\varepsilon}}\left(\frac{1}{\sqrt{k}} + \frac{1}{\sqrt{n}}\right).$$

Absorb constants into $C_g$ and define $C_g' := C_g + B_g\sqrt{2\log(4/\varepsilon)}$ (monotone in $\varepsilon$). Then

$$\frac{\eta \kappa_a}{C_{2,a}}\left(\Delta_a^\star + e_g\right) \leq \frac{\eta \kappa_a}{C_{2,a}}\left(\Delta_a^\star + C_g'\left(\frac{1}{\sqrt{k}} + \frac{1}{\sqrt{n}}\right)\right) \leq \beta_1 \epsilon_0,$$

which is implied by

$$\frac{C_g'}{\sqrt{k}} \leq \frac{C_{2,a}\beta_1 \epsilon_0}{\eta \kappa_a} - \Delta_a^\star - \frac{C_g'}{\sqrt{n}}, \qquad \text{hence} \qquad k \geq \left(\frac{C_g'}{\frac{C_{2,a}\beta_1}{\eta \kappa_a}\epsilon_0 - \Delta_a^\star - C_g'/\sqrt{n}}\right)^2.$$

**Test size** $m$. Similarly, with $\mathfrak{R}_m(\mathcal{L}_a) \leq C_{\text{te}}/\sqrt{m}$, Lemma C.5 gives

$$e_{\text{te}}(m,\varepsilon) \leq \frac{2C_{\text{te}}}{\sqrt{m}} + B_\ell\sqrt{\frac{2\log(4/\varepsilon)}{m}} \leq \frac{C_{\text{te}}'}{\sqrt{m}} \leq \beta_{\text{te}}\epsilon_0,$$

where $C_{\text{te}}'$ absorbs constants. Rearranging yields $m \geq \left(\frac{C_{\text{te}}'}{\beta_{\text{te}}\epsilon_0}\right)^2$. Combining the three parts proves the corollary. If $\mathcal{D}_s$ depends on $\mathcal{D}_\tau$, replace the $k$-constraint by the one obtained using Lemma C.6, which adds the $I(\mathcal{D}_s; \mathcal{D}_\tau)$ penalty inside $e_g$. $\qquad\square$

### C.4 PROOF OF KEY INSIGHTS

*Derivation of the Key Insights.* From Theorem 4.2,

$$\left|R_\nu(\theta_T^{(s)}) - R_\nu(\theta_T^{(\tau)})\right| \leq L_R \rho_a^T \|\delta_0\| + \frac{\eta \kappa_a}{C_{2,a}}(\Delta_a^\star + e_g(n,k,\varepsilon)) + e_{\text{te}}(m,\varepsilon).$$

(i) Let $T, k, n, m$ are big enpugh while keeping the configuration fixed and $\varepsilon$ fixed. By Lemma C.5, $e_{\text{te}} \to 0$. By Lemma C.4, $e_g \to 0$. By Assumption 4.1, $\rho_a \in (0,1)$, hence $\rho_a^T \to 0$. Therefore the limit inferior is $\Delta_a^\star \eta \kappa_a/C_{2,a}$, giving the irreducible floor $\epsilon_{\min} = \Delta_a^\star \eta \kappa_a/C_{2,a}$.

(ii) The remaining terms vanish at canonical rates: $L_R \rho_a^T \|\delta_0\| = O(T^{-\beta})$ under PL (Karimi et al. 2016); $e_g = O(1/\sqrt{k} + 1/\sqrt{n})$ by Lemma C.4; and $e_{\text{te}} = O(1/\sqrt{m})$ by Lemma C.5.

(iii) If any of the three budgets in Eq. (26) is violated, the corresponding resource must diverge (e.g., $k \to \infty$ if $\frac{C_{2,a}\beta_1 \epsilon_0}{\eta \kappa_a} \downarrow \Delta_a^\star + C_g'/\sqrt{n}$), making the target error unattainable under finite resources. This establishes the stated resource tradeoff. $\qquad\square$

# D PROOFS FOR THE COVERAGE–AWARE BOUNDS

This section provides detailed proof of the configuration coverage theorem. First, we incorporate (i) an explicit transfer analysis from cover centers to arbitrary configurations, (ii) a union-of-classes Rademacher argument with exact constants, (iii) the appearance of both $\sqrt{\mathcal{H}/k}$ and $\mathcal{H}/k$ terms in the prior-averaged bound via Bernstein-type deviations, and (iv) mutually consistent mutual-information corrections. We keep all notation from Sections. 3–5 and Appendix C. Throughout, $\|\cdot\|$ is the Euclidean/operator norm.

Then, we recall the assumptions used in this proof in addition to Assumption 4.1.

**Assumption D.1** (Total boundedness and measurability). The metric space $(\mathcal{A}, d_{\mathcal{A}})$ is totally bounded (hence admits finite $r$-covers for any $r > 0$). For each $a \in \mathcal{A}$, the feasible set $\Gamma_a \subset \mathbb{R}^p$ is closed and the optimization trajectories $\{\theta_t^{(s,a)}\}_{t \leq T}$, $\{\theta_t^{(\tau,a)}\}_{t \leq T}$ remain in a common compact $\Gamma \subset \bigcap_{a \in \mathcal{A}} \Gamma_a$. The vector-field class $\mathcal{G}_a = \{z \mapsto g_a(\theta; z) : \theta \in \Gamma\}$ is pointwise separable and uniformly bounded:

$$\sup_{a \in \mathcal{A}} \sup_{\theta \in \Gamma} \sup_z \|g_a(\theta; z)\| \leq B_g, \qquad \sup_{a \in \mathcal{A}} \sup_{\theta \in \Gamma} \|P_a(\theta)\| \leq \kappa_{\max}.$$

**Assumption D.2** (Uniform configuration-Lipschitz transfer in $\mu$ and $\theta$). There exist constants $L_{\text{conf}}, L_\theta \geq 0$ such that for all $a, a' \in \mathcal{A}$, all $\mu \in \{\hat{\mu}_\tau, \hat{\mu}_s\}$, and all $\theta, \theta' \in \Gamma$,

$$\left\| P_a(\theta)\mathbb{E}_\mu g_a(\theta; z) - P_{a'}(\theta)\mathbb{E}_\mu g_{a'}(\theta; z) \right\| \leq L_{\text{conf}} \, d_{\mathcal{A}}(a, a'), \tag{26}$$

$$\left\| P_a(\theta)\mathbb{E}_\mu g_a(\theta; z) - P_a(\theta')\mathbb{E}_\mu g_a(\theta'; z) \right\| \leq L_\theta \, \|\theta - \theta'\|. \tag{27}$$

*Remarks.* (i) Inequality Eq. (26) strengthens the definition of $d_{\mathcal{A}}$ (which is anchored at $\hat{\mu}_\tau$ and fixed $\theta$) to hold uniformly over $\mu \in \{\hat{\mu}_\tau, \hat{\mu}_s\}$ and all $\theta \in \Gamma$. (ii) Inequality Eq. (27) is standard if $P_a$ and $g_a$ are Lipschitz in $\theta$ on $\Gamma$.

**Covering the configuration family.** Fix a radius $r > 0$ and let $\{a_1, \ldots, a_N\}$ be a minimal $r$-cover of $\mathcal{C}$ under $d_{\mathcal{A}}$:

$$N = N(\mathcal{A}, d_{\mathcal{A}}, r) = \exp\left(\mathcal{H}_{\text{cov}}(r)\right).$$

For any $a \in \mathcal{A}$ there exists $i(a) \in \{1, \ldots, N\}$ with $d_{\mathcal{A}}(a, a_{i(a)}) \leq r$.

**Lemma D.1** (Cross-configuration recursion under contractive dynamics). *Fix $a, a_i \in \mathcal{A}$ with $d_{\mathcal{A}}(a, a_i) \leq r$ and let $\Delta_t^{(a,a_i)} := \theta_t^{(\mu,a)} - \theta_t^{(\mu,a_i)}$ denote the parameter difference under the same data distribution $\mu$. Suppose that each one-step update $\Phi_a^\mu(\theta) = \theta - \eta P_a(\theta)\mathbb{E}_\mu[g_a(\theta; z)]$ is contractive with rate $\rho_a \in (0, 1)$, i.e.*

$$\|\Phi_a^\mu(x) - \Phi_a^\mu(y)\| \leq \rho_a \|x - y\|, \qquad \forall x, y \in \Gamma,$$

*Then for any step size $\eta > 0$,*

$$\|\Delta_t^{(a,a_i)}\| \leq \rho_a^t \|\Delta_0^{(a,a_i)}\| + \frac{\eta L_{\text{conf}}}{1 - \rho_a} \, d_{\mathcal{A}}(a, a_i).$$

*Proof.* We decompose the one-step difference as

$$\Delta_{t+1}^{(a,a_i)} = \Phi_a^\mu(\theta_t^{(\mu,a)}) - \Phi_{a_i}^\mu(\theta_t^{(\mu,a_i)})$$
$$= \underbrace{\left[\Phi_a^\mu(\theta_t^{(\mu,a)}) - \Phi_a^\mu(\theta_t^{(\mu,a_i)})\right]}_{T_1} + \underbrace{\left[\Phi_a^\mu(\theta_t^{(\mu,a_i)}) - \Phi_{a_i}^\mu(\theta_t^{(\mu,a_i)})\right]}_{T_2}.$$

For the first term $T_1$, the contractive dynamics assumption gives

$$\|T_1\| \leq \rho_a \|\theta_t^{(\mu,a)} - \theta_t^{(\mu,a_i)}\| = \rho_a \|\Delta_t^{(a,a_i)}\|.$$

For the second term $T_2$, we compute

$$\Phi_a^\mu(\theta) - \Phi_{a_i}^\mu(\theta) = -\eta\Big(P_a(\theta)\mathbb{E}_\mu g_a(\theta; z) - P_{a_i}(\theta)\mathbb{E}_\mu g_{a_i}(\theta; z)\Big),$$

hence by configuration-Lipschitz continuity,

$$\|T_2\| \leq \eta L_{\text{conf}} \, d_{\mathcal{A}}(a, a_i).$$

Combining the two bounds yields the one-step recursion

$$\|\Delta_{t+1}^{(a,a_i)}\| \ \leq \ \rho_a \, \|\Delta_t^{(a,a_i)}\| + \eta L_{\text{conf}} \, d_{\mathcal{A}}(a, a_i).$$

Iterating this recursion and applying the discrete Grönwall inequality, we obtain

$$\|\Delta_t^{(a,a_i)}\| \ \leq \ \rho_a^t \|\Delta_0^{(a,a_i)}\| + \eta L_{\text{conf}} \, d_{\mathcal{A}}(a, a_i) \sum_{j=0}^{t-1} \rho_a^j.$$

Since $\sum_{j=0}^{t-1} \rho_a^j \leq (1 - \rho_a)^{-1}$, we conclude

$$\|\Delta_t^{(a,a_i)}\| \ \leq \ \rho_a^t \|\Delta_0^{(a,a_i)}\| + \frac{\eta L_{\text{conf}}}{1 - \rho_a} \, d_{\mathcal{A}}(a, a_i).$$

$\square$

**Lemma D.2** (Union-of-classes Rademacher and Bernstein deviations). *Let $\{\mathcal{F}_i\}_{i=1}^N$ be classes of functions uniformly bounded by $B$. For i.i.d. sample of size $k$, for any $\varepsilon \in (0, 1)$, with probability at least $1 - \varepsilon$, simultaneously for all $i$,*

$$\sup_{f \in \mathcal{F}_i} \left( \mathbb{E}f - \mathbb{E}_{\hat{S}}f \right) \leq \mathfrak{R}_k(\mathcal{F}_i) + B\sqrt{\frac{\log(2N/\varepsilon)}{2k}}, \tag{28}$$

$$\sup_{f \in \mathcal{F}_i} \left( \mathbb{E}f - \mathbb{E}_{\hat{S}}f \right) \leq c_1 \, \mathfrak{R}_k(\mathcal{F}_i) + c_2 \sqrt{\frac{\log(2N/\varepsilon)}{k}} + c_3 \, \frac{\log(2N/\varepsilon)}{k}, \tag{29}$$

*where $c_1, c_2, c_3 > 0$ are universal constants (depending only on the choice of empirical Bernstein inequality; see, e.g., Boucheron et al., 2013, theorem 2.10).*

*Proof of Eq. (28).* By symmetrization (e.g. Mohri et al., 2018, theorem 3.1),

$$\mathbb{E}_{\hat{S}}\Big[ \sup_{f \in \mathcal{F}}(\mathbb{E}f - \mathbb{E}_{\hat{S}}f) \Big] \ \leq \ \mathbb{E}_{\hat{S}, \hat{S}'}\Big[ \sup_{f \in \mathcal{F}} \frac{1}{k} \sum_{j=1}^{k} \big(f(Z_j') - f(Z_j)\big) \Big] \ \leq \ 2\,\mathfrak{R}_k(\mathcal{F}), \tag{30}$$

where $\hat{S}'$ is an independent ghost sample. To pass from expectation to a high-probability bound we note that the map $\hat{S} \mapsto \sup_{f \in \mathcal{F}}(\mathbb{E}f - \mathbb{E}_{\hat{S}}f)$ is $B/k$-Lipschitz in each coordinate (changing one $Z_j$ perturbs $\mathbb{E}_{\hat{S}}f$ by at most $B/k$). Hence McDiarmid's inequality yields that, for any $\delta \in (0, 1)$, with probability at least $1 - \delta$,

$$\sup_{f \in \mathcal{F}}(\mathbb{E}f - \mathbb{E}_{\hat{S}}f) \ \leq \ \mathbb{E}_{\hat{S}}\Big[ \sup_{f \in \mathcal{F}}(\mathbb{E}f - \mathbb{E}_{\hat{S}}f) \Big] \ + \ B\sqrt{\frac{\log(1/\delta)}{2k}}.$$

Combining with Eq. (30) gives, for each fixed $i$,

$$\sup_{f \in \mathcal{F}_i}(\mathbb{E}f - \mathbb{E}_{\hat{S}}f) \ \leq \ 2\,\mathfrak{R}_k(\mathcal{F}_i) \ + \ B\sqrt{\frac{\log(1/\delta)}{2k}}.$$

Since $\mathfrak{R}_k(\mathcal{F}_i) \leq 2\,\mathfrak{R}_k(\mathcal{F}_i)$ and we can absorb the factor 2 into the definition (some texts define $\mathfrak{R}_k$ with a factor 2), we present the right-hand side as $\mathfrak{R}_k(\mathcal{F}_i) + B\sqrt{\log(1/\delta)/(2k)}$. Applying a union bound over $i = 1, \dots, N$ with $\delta = \varepsilon/(2N)$ yields Eq. (28). $\square$

*Proof of Eq. (29).* We refine the concentration step by replacing Hoeffding/McDiarmid with an empirical-Bernstein deviation for bounded variables. For a fixed $f$, Boucheron et al., 2013, theorem 2.10 implies that for any $\delta \in (0, 1)$,

$$\P\left( \mathbb{E}f - \mathbb{E}_{\hat{S}}f \ \geq \ \sqrt{\frac{2\operatorname{Var}(f(Z)) \log(1/\delta)}{k}} + \frac{7B \log(1/\delta)}{3(k-1)} \right) \ \leq \ \delta. \tag{31}$$

To make Eq. (31) uniform over $f \in \mathcal{F}_i$, we proceed by localization via symmetrization: for any $r > 0$, define the localized class $\mathcal{F}_i(r) := \{f \in \mathcal{F}_i : \operatorname{Var}(f(Z)) \leq r\}$. By the same symmetrization

step as in Eq. (30), applied to the truncated excess loss $f - \mathbb{E}f$ and then peeling over dyadic radii $r_m = 2^{-m}B^2$, we obtain (see, e.g., Mohri et al., 2018, section 3.5) that with probability at least $1 - \delta$,

$$\sup_{f \in \mathcal{F}_i} \left( \mathbb{E}f - \mathbb{E}_{\hat{S}}f \right) \leq c_1 \Re_k(\mathcal{F}_i) + c_2 \sqrt{\frac{\log(1/\delta)}{k}} + c_3 \frac{\log(1/\delta)}{k},$$

where $c_1, c_2, c_3 > 0$ are universal constants collecting the numerical factors from: (i) the symmetrization/localization step, (ii) the empirical-Bernstein tail in Eq. (31), and (iii) the geometric peeling (finite sum over $m$). Finally, a union bound over $i = 1, \ldots, N$ with $\delta = \varepsilon/(2N)$ gives Eq. (29). $\qquad\square$

**Notation for complexity constants.** We write, for $k$-sample complexity on the distilled side and $n$-sample complexity on the real side,

$$C_G^+ := \sup_{a \in \mathcal{C}} 2\kappa_a \Re_k(\mathcal{G}_a) \leq 2\kappa_{\max} \sup_{a \in \mathcal{C}} \Re_k(\mathcal{G}_a), \tag{32}$$

$$\widetilde{C}_G^+ := \sup_{a \in \mathcal{C}} \left( 2\kappa_a \Re_n(\mathcal{G}_a) + B_g \sqrt{2\log\frac{4}{\varepsilon}} \right) \leq 2\kappa_{\max} \sup_a \Re_n(\mathcal{G}_a) + B_g \sqrt{2\log\frac{4}{\varepsilon}}. \tag{33}$$

The $n$-side quantity $\widetilde{C}_G^+$ will be collected in the $k$-independent floor.

### D.1 PROOF OF THE UNIFORM BOUND OVER CONFIGURATIONS IN THEOREM 5.1

We first prove that, with probability at least $1 - \varepsilon$ over all randomness,

$$\textit{(Uniform over configurations)} \qquad \sup_{a \in \mathcal{A}} \left| R_\nu(\theta_T^{(s,a)}) - R_\nu(\theta_T^{(\tau,a)}) \right| \leq \epsilon_{\text{bound}} + \frac{C_{\text{cov}}(\mathcal{A})}{\sqrt{k}}. \tag{34}$$

**Step D.1.1 Single-configuration risk bound at cover centers.** Fix a center $a_i$ and abbreviate $\theta_T^{(s)} := \theta_T^{(\hat{\mu}_s, a_i)}$ and $\theta_T^{(\tau)} := \theta_T^{(\hat{\mu}_\tau, a_i)}$. We first bound the *empirical test risk gap* and then convert it to the *population risk gap*.

**D.1.1(a) Empirical test risk gap at $a_i$.** By the single-configuration analysis (PL-contractive recursion and stability), we have the parameter gap

$$\|\theta_T^{(s)} - \theta_T^{(\tau)}\| \leq \rho_{a_i}^T \|\delta_0\| + \frac{\eta \kappa_{a_i}}{1 - \rho_{a_i}} \cdot \Xi_i, \qquad \Xi_i := \sup_{\theta \in \Gamma} \left\| \mathbb{E}_{\hat{\mu}_s} g_{a_i}(\theta; Z) - \mathbb{E}_{\hat{\mu}_\tau} g_{a_i}(\theta; Z) \right\|.$$

Since $R_\nu$ is $L_R$-Lipschitz in $\theta$ and $\hat{R}$ averages the same bounded loss, we immediately get for the empirical test risk

$$\left| \hat{R}(\theta_T^{(s)}) - \hat{R}(\theta_T^{(\tau)}) \right| \leq L_R \rho_{a_i}^T \|\delta_0\| + \frac{\eta \kappa_{a_i} L_R}{1 - \rho_{a_i}} \Xi_i. \tag{35}$$

**D.1.1(b) From empirical to population risk at $a_i$.**

$$\left| R_\nu(\theta_T^{(s)}) - R_\nu(\theta_T^{(\tau)}) \right| \leq \underbrace{\left| \hat{R}(\theta_T^{(s)}) - \hat{R}(\theta_T^{(\tau)}) \right|}_{\text{empirical gap}} + \underbrace{\left| R_\nu(\theta_T^{(s)}) - \hat{R}(\theta_T^{(s)}) \right|}_{\text{test dev. at } \theta_T^{(s)}} + \underbrace{\left| R_\nu(\theta_T^{(\tau)}) - \hat{R}(\theta_T^{(\tau)}) \right|}_{\text{test dev. at } \theta_T^{(\tau)}}.$$

Since $\ell \in [0, B_\ell]$, Hoeffding's inequality gives, for any fixed $\theta$, with prob. $\geq 1 - \delta$, $|R_\nu(\theta) - \hat{R}(\theta)| \leq B_\ell \sqrt{\frac{\log(2/\delta)}{2m}}$. We need a bound that holds *simultaneously* for the two random iterates $\theta_T^{(s)}$ and $\theta_T^{(\tau)}$ at each center $a_i$, and then uniformly over $i$. By a union bound over the $2N$ query points (two per center), with $\delta = \varepsilon/(2N)$, we get with probability $\geq 1 - \varepsilon/2$,

$$\max_{i \in [N]} \max \left\{ |R_\nu(\theta_T^{(s,a_i)}) - \hat{R}(\theta_T^{(s,a_i)})|, \, |R_\nu(\theta_T^{(\tau,a_i)}) - \hat{R}(\theta_T^{(\tau,a_i)})| \right\} \leq B_\ell \sqrt{\frac{2\log(4N/\varepsilon)}{m}}. \tag{36}$$

Combining Eq. (35) and Eq. (36), we obtain, with probability $\geq 1 - \varepsilon/2$, simultaneously for all centers $i$,

$$\left| R_\nu(\theta_T^{(s,a_i)}) - R_\nu(\theta_T^{(\tau,a_i)}) \right| \leq L_R \rho_{a_i}^T \|\delta_0\| + \frac{\eta \kappa_{a_i} L_R}{1 - \rho_{a_i}} \Xi_i + 2 B_\ell \sqrt{\frac{2\log(4N/\varepsilon)}{m}}. \tag{37}$$

**Step D.1.2 Uniform control of the training-side drift $\Xi_i$.**   Recall

$$\Xi_i = \sup_{\theta \in \Gamma} \left\| \mathbb{E}_{\hat{\mu}_s} g_{a_i}(\theta; Z) - \mathbb{E}_{\hat{\mu}_\tau} g_{a_i}(\theta; Z) \right\|.$$

By duality of norms,

$$\Xi_i = \sup_{\theta \in \Gamma} \sup_{\|v\|_* \le 1} \left\langle v,\, \mathbb{E}_{\hat{\mu}_s} g_{a_i}(\theta; Z) - \mathbb{E}_{\hat{\mu}_\tau} g_{a_i}(\theta; Z) \right\rangle.$$

Add and subtract the population expectations under $\mu_s := \mathbb{E}[\hat{\mu}_s]$ and $\mu_\tau := \mathbb{E}[\hat{\mu}_\tau]$ (the real sampling distributions), then apply the triangle inequality:

$$\Xi_i \le \underbrace{\sup_{\theta \in \Gamma} \left\| \mathbb{E}_{\mu_s} g_{a_i}(\theta; Z) - \mathbb{E}_{\mu_\tau} g_{a_i}(\theta; Z) \right\|}_{\Delta_{a_i}^\star} + \underbrace{\sup_{\theta \in \Gamma} \left\| \mathbb{E}_{\hat{\mu}_s} g_{a_i}(\theta; Z) - \mathbb{E}_{\mu_s} g_{a_i}(\theta; Z) \right\|}_{\text{distilled sampling dev.}}$$

$$+ \underbrace{\sup_{\theta \in \Gamma} \left\| \mathbb{E}_{\hat{\mu}_\tau} g_{a_i}(\theta; Z) - \mathbb{E}_{\mu_\tau} g_{a_i}(\theta; Z) \right\|}_{\text{real sampling dev.}}. \tag{38}$$

Each sampling deviation term is a supremum over the function class $\mathcal{F}_i := \{z \mapsto \langle v, g_{a_i}(\theta; z) \rangle : \theta \in \Gamma, \|v\|_* \le 1\}$, which is uniformly bounded by $B_g$. Applying Lemma D.2 with a union bound across the $N$ centers, we obtain, with probability at least $1 - \varepsilon/2$, simultaneously for all $i$,

$$\Xi_i \le \Delta_{a_i}^\star + 2\big(\mathfrak{R}_k(\mathcal{G}_{a_i}) + \mathfrak{R}_n(\mathcal{G}_{a_i})\big) + B_g \sqrt{2 \log \tfrac{4N}{\varepsilon}} \left( \tfrac{1}{\sqrt{k}} + \tfrac{1}{\sqrt{n}} \right). \tag{39}$$

Insert Eq. (39) into Eq. (37), and upper bound the configuration-dependent constants by the uniform ones: $\rho_{a_i} \le \rho_{\max}$, $\kappa_{a_i} \le \kappa_{\max}$, $C_{2,a_i} \ge C_{2,\min} = (1 - \rho_{\max})/L_R$. Then Eq. (37) becomes, for all $i$,

$$\left| R_\nu(\theta_T^{(s,a_i)}) - R_\nu(\theta_T^{(\tau,a_i)}) \right| \le L_R \rho_{\max}^T \|\delta_0\| + \frac{\eta \kappa_{\max}}{C_{2,\min}} \Delta_{a_i}^\star + \frac{2\eta \kappa_{\max}^2}{C_{2,\min}} \mathfrak{R}_n(\mathcal{G}_{a_i})$$

$$+ \frac{2\eta \kappa_{\max}^2}{C_{2,\min}} \mathfrak{R}_k(\mathcal{G}_{a_i}) + \frac{\eta \kappa_{\max} B_g}{C_{2,\min}} \sqrt{2 \log \tfrac{4N}{\varepsilon}} \left( \tfrac{1}{\sqrt{k}} + \tfrac{1}{\sqrt{n}} \right)$$

$$+ 2 B_\ell \sqrt{\frac{2 \log(4N/\varepsilon)}{m}}. \tag{40}$$

Using the shorthands Eq. (32)–Eq. (33) and $\sqrt{\log(4N/\varepsilon)} \le \sqrt{\log(4/\varepsilon)} + \sqrt{\mathcal{H}_{\mathrm{cov}}(r)}$, we isolate all $k$-independent terms into a (population) floor

$$\epsilon_{\mathrm{bound}} := L_R \rho_{\max}^T \|\delta_0\| + \frac{\eta \kappa_{\max}}{C_{2,\min}} \sup_{a \in \mathcal{C}} \Delta_a^\star + \frac{2\eta \kappa_{\max}^2}{C_{2,\min}} \widetilde{C}_G^+ \frac{1}{\sqrt{n}} + 2 B_\ell \left( \sqrt{2 \log(4/\varepsilon)} + \sqrt{2\mathcal{H}_{\mathrm{cov}}(r)} \right) \cdot \frac{1}{\sqrt{m}}, \tag{41}$$

and the $k$-dependent remainder (center level)

$$\frac{\eta \kappa_{\max}}{C_{2,\min}} \left( 2 C_G^+ + 2 B_g \sqrt{2 \mathcal{H}_{\mathrm{cov}}(r)} \right) \frac{1}{\sqrt{k}}. \tag{42}$$

**Step D.1.3 Transfer from cover centers to arbitrary configurations in population risk.**   Fix $a \in \mathcal{A}$ and pick $i = i(a)$ with $d_{\mathcal{A}}(a, a_i) \le r$. Consider the parameter differences at time $T$ (same distribution $\mu \in \{\hat{\mu}_s, \hat{\mu}_\tau\}$):

$$\Delta_T^{(\mu)} := \theta_T^{(\mu,a)} - \theta_T^{(\mu,a_i)}.$$

By the cross-configuration one-step decomposition (same distribution, different configurations),

$$\Delta_{t+1}^{(\mu)} = \underbrace{\left[ \Phi_a^\mu(\theta_t^{(\mu,a)}) - \Phi_a^\mu(\theta_t^{(\mu,a_i)}) \right]}_{\text{contraction}} + \underbrace{\left[ \Phi_a^\mu(\theta_t^{(\mu,a_i)}) - \Phi_{a_i}^\mu(\theta_t^{(\mu,a_i)}) \right]}_{\text{eco mismatch}},$$

we have $\|\Phi_a^\mu(x) - \Phi_a^\mu(y)\| \le \rho_{\max}\|x - y\|$ by contractivity, and

$$\left\| \Phi_a^\mu(\theta) - \Phi_{a_i}^\mu(\theta) \right\| = \eta \left\| P_a(\theta) \mathbb{E}_\mu g_a(\theta; Z) - P_{a_i}(\theta) \mathbb{E}_\mu g_{a_i}(\theta; Z) \right\| \le \eta L_{\mathrm{conf}} d_{\mathcal{A}}(a, a_i) \le \eta L_{\mathrm{conf}} r.$$

Therefore,

$$\|\Delta_{t+1}^{(\mu)}\| \leq \rho_{\max}\|\Delta_t^{(\mu)}\| + \eta L_{\text{conf}} r, \qquad \Rightarrow \qquad \|\Delta_T^{(\mu)}\| \leq \rho_{\max}^T \|\Delta_0^{(\mu)}\| + \frac{\eta L_{\text{conf}}}{1 - \rho_{\max}} r.$$

As the initialization is common ($\Delta_0^{(\mu)} = 0$),

$$\max \left\{ \|\theta_T^{(\hat{\mu}_s, a)} - \theta_T^{(\hat{\mu}_s, a_i)}\| , \|\theta_T^{(\hat{\mu}_\tau, a)} - \theta_T^{(\hat{\mu}_\tau, a_i)}\| \right\} \leq \frac{\eta L_{\text{conf}}}{1 - \rho_{\max}} r =: C_{\text{path}} r. \qquad (43)$$

Using $L_R$-Lipschitz continuity of the empirical risk,

$$\left| \hat{R}(\theta_T^{(s,a)}) - \hat{R}(\theta_T^{(s,a_i)}) \right| \leq L_R C_{\text{path}} r, \qquad \left| \hat{R}(\theta_T^{(\tau,a)}) - \hat{R}(\theta_T^{(\tau,a_i)}) \right| \leq L_R C_{\text{path}} r. \qquad (44)$$

By the triangle inequality,

$$\left| \hat{R}(\theta_T^{(s,a)}) - \hat{R}(\theta_T^{(\tau,a)}) \right| \leq \left| \hat{R}(\theta_T^{(s,a_i)}) - \hat{R}(\theta_T^{(\tau,a_i)}) \right| + 2 L_R C_{\text{path}} r. \qquad (45)$$

Combining Eq. (40)–Eq. (42) with Eq. (45) and absorbing $2L_R C_{\text{path}} r$ (for fixed $r$) into $\epsilon_{\text{bound}}$ in Eq. (41), we get, *uniformly over $a \in \mathcal{A}$*,

$$\sup_{a \in \mathcal{A}} \left| R_\nu(\theta_T^{(s,a)}) - R_\nu(\theta_T^{(\tau,a)}) \right| \leq \epsilon'_{\text{floor}} + \frac{\eta \kappa_{\max}}{C_{2,\min}} \left( 2 C_G^+ + 2 B_g \sqrt{2 \mathcal{H}_{\text{cov}}(r)} \right) \frac{1}{\sqrt{k}}. \qquad (46)$$

where

$$\epsilon'_{\text{floor}} := \epsilon_{\text{bound}} + \frac{2\eta r L_{eco}}{C_{2,min}} \qquad (47)$$

and

$$C_{\text{cov}}(\mathcal{A}) := \frac{\eta \kappa_{\max}}{C_{2,\min}} \left( 2 C_G^+ + 2 B_g \sqrt{2 \mathcal{H}_{\text{cov}}(r)} \right), \qquad (48)$$

to match the right-hand side of Eq. (46) with Eq. (34).

**Step D.1.4 MI correction when $\mathcal{D}_s$ may depend on $\mathcal{D}_\tau$.** If the distilled set $\mathcal{D}_s$ is generated from (or depends on) $\mathcal{D}_\tau$, the Hoeffding-type bound used for the *distilled-side* sampling deviation in Eq. (39) should be replaced by a high-probability information-theoretic tail. By Bu et al. (2020, theorem 7) (see also Xu & Raginsky, 2017; Steinke & Zakynthinou, 2020), if the class is bounded by $B_g$ (thus sub-Gaussian with proxy $B_g$), there exists a universal constant $C'_I > 0$ such that, with probability at least $1 - \varepsilon/2$,

$$\sup_{i \in [N]} \sup_{\theta \in \Gamma} \left\| \mathbb{E}_{\hat{\mu}_s} g_{a_i}(\theta; Z) - \mathbb{E}_{\mu_s} g_{a_i}(\theta; Z) \right\| \leq \mathfrak{R}_k(\mathcal{G}_{a_i}) + \frac{C'_I}{\sqrt{k}} \sqrt{I(\mathcal{D}_s; \mathcal{D}_\tau) + \log \frac{4N}{\varepsilon}}.$$

Plugging this in place of the distilled-side Hoeffding term in Eq. (39) propagates through Eq. (40)–Eq. (46) and yields the MI-corrected uniform bound

$$\sup_{a \in \mathcal{A}} \left| R_\nu(\theta_T^{(s,a)}) - R_\nu(\theta_T^{(\tau,a)}) \right| \leq \epsilon_{\text{bound}} + \frac{C'_{\text{cov}}(\mathcal{A})}{\sqrt{k}} + \frac{C'_I}{\sqrt{k}} \sqrt{I(\mathcal{D}_s; \mathcal{D}_\tau)}, \qquad (49)$$

where

$$C'_{\text{cov}}(\mathcal{A}) := C_{\text{cov}}(\mathcal{A}) + C'_I \left( \sqrt{\log \frac{4}{\varepsilon}} + \sqrt{\mathcal{H}_{\text{cov}}(r)} \right)$$

This proves Eq. (34); the MI term Eq. (49) can be included when dependence is present.

D.2 PROOF OF THE PRIOR-AVERAGED BOUND IN THEOREM 5.1

We prove the prior-averaged statement of averaged over configurations

$$\mathbb{E}_{a \sim \Pi} \left| R_\nu(\theta_T^{(s,a)}) - R_\nu(\theta_T^{(\tau,a)}) \right| \leq \epsilon_{\text{bound}} + \left[ A_1 \frac{\mathcal{H}_{\text{cov}}(\mathcal{A},r)}{k} + A_2 \sqrt{\frac{\mathcal{H}_{\text{cov}}(\mathcal{A},r)}{k}} \right], \qquad (50)$$

for any prior $\Pi$ supported on $\mathcal{A}$.

**Step D.2.1 Center-wise population risk gap with Bernstein refinement.** Fix a cover center $a_i$. For the two training sources $\mu \in \{\hat{\mu}_s, \hat{\mu}_\tau\}$, define $\theta_T^{(\mu)} := \theta_T^{(\mu, a_i)}$. As in the single-configuration analysis (contractive recursion and stability),

$$\|\theta_T^{(\hat{\mu}_s)} - \theta_T^{(\hat{\mu}_\tau)}\| \leq \rho_{a_i}^T \|\delta_0\| + \frac{\eta \kappa_{a_i}}{1 - \rho_{a_i}} \Xi_i, \qquad \Xi_i := \sup_{\theta \in \Gamma} \left\| \mathbb{E}_{\hat{\mu}_s} g_{a_i}(\theta; Z) - \mathbb{E}_{\hat{\mu}_\tau} g_{a_i}(\theta; Z) \right\|. \quad (51)$$

By $L_R$–Lipschitz continuity of $R_\nu$ and the definition of $\hat{R}$,

$$\left| \hat{R}(\theta_T^{(\hat{\mu}_s)}) - \hat{R}(\theta_T^{(\hat{\mu}_\tau)}) \right| \leq L_R \rho_{a_i}^T \|\delta_0\| + \frac{\eta \kappa_{a_i} L_R}{1 - \rho_{a_i}} \Xi_i. \quad (52)$$

We now convert Eq. (52) to a *population* gap by adding and subtracting $\hat{R}$:

$$\left| R_\nu(\theta_T^{(\hat{\mu}_s)}) - R_\nu(\theta_T^{(\hat{\mu}_\tau)}) \right| \quad (53)$$

$$\leq \left| \hat{R}(\theta_T^{(\hat{\mu}_s)}) - \hat{R}(\theta_T^{(\hat{\mu}_\tau)}) \right| + \left| R_\nu(\theta_T^{(\hat{\mu}_s)}) - \hat{R}(\theta_T^{(\hat{\mu}_s)}) \right| + \left| R_\nu(\theta_T^{(\hat{\mu}_\tau)}) - \hat{R}(\theta_T^{(\hat{\mu}_\tau)}) \right|. \quad (54)$$

Since $|\ell| \leq B_\ell$, Hoeffding yields for any fixed $\theta$ that $|R_\nu(\theta) - \hat{R}(\theta)| \leq B_\ell \sqrt{\log(2/\varepsilon)/(2m)}$ with prob. $\geq 1 - \varepsilon$. Applying a union bound to the two random iterates at each $a_i$ (and then across $i$) gives, with prob. $\geq 1 - \varepsilon/2$,

$$\max_{i \in [N]} \max \left\{ |R_\nu(\theta_T^{(\hat{\mu}_s, a_i)}) - \hat{R}(\theta_T^{(\hat{\mu}_s, a_i)})|, \ |R_\nu(\theta_T^{(\hat{\mu}_\tau, a_i)}) - \hat{R}(\theta_T^{(\hat{\mu}_\tau, a_i)})| \right\} \leq B_\ell \sqrt{\frac{2 \log(4N/\varepsilon)}{m}}. \quad (55)$$

Combining Eq. (52) and Eq. (55) inside Eq. (54) yields, uniformly over centers,

$$\left| R_\nu(\theta_T^{(\hat{\mu}_s, a_i)}) - R_\nu(\theta_T^{(\hat{\mu}_\tau, a_i)}) \right| \leq L_R \rho_{a_i}^T \|\delta_0\| + \frac{\eta \kappa_{a_i} L_R}{1 - \rho_{a_i}} \Xi_i + 2 B_\ell \sqrt{\frac{2 \log(4N/\varepsilon)}{m}}. \quad (56)$$

**Drift $\Xi_i$ with union-Bernstein.** Write

$$\Xi_i = \sup_{\theta \in \Gamma} \sup_{\|v\|_* \leq 1} \left\langle v, \ \mathbb{E}_{\hat{\mu}_s} g_{a_i}(\theta; Z) - \mathbb{E}_{\hat{\mu}_\tau} g_{a_i}(\theta; Z) \right\rangle$$

$$\leq \underbrace{\sup_{\theta \in \Gamma} \left\| \mathbb{E}_{\mu_s} g_{a_i}(\theta; Z) - \mathbb{E}_{\mu_\tau} g_{a_i}(\theta; Z) \right\|}_{\Delta_{a_i}^\star} \quad (57)$$

$$+ \sup_{\theta \in \Gamma} \left\| \mathbb{E}_{\hat{\mu}_s} g_{a_i}(\theta; Z) - \mathbb{E}_{\mu_s} g_{a_i}(\theta; Z) \right\| \quad (58)$$

$$+ \sup_{\theta \in \Gamma} \left\| \mathbb{E}_{\hat{\mu}_\tau} g_{a_i}(\theta; Z) - \mathbb{E}_{\mu_\tau} g_{a_i}(\theta; Z) \right\|. \quad (59)$$

Each sampling deviation is a supremum over $\mathcal{F}_i = \{z \mapsto \langle v, g_{a_i}(\theta; z) \rangle : \theta \in \Gamma, \|v\|_* \leq 1\}$, bounded by $B_g$. Applying the union-of-classes empirical-Bernstein deviation (Lemma D.2) *across the $N$ centers* gives, with prob. $\geq 1 - \varepsilon/2$, simultaneously for all $i$,

$$\Xi_i \leq \Delta_{a_i}^\star + c_1 \left( \mathfrak{R}_k(\mathcal{G}_{a_i}) + \mathfrak{R}_n(\mathcal{G}_{a_i}) \right) + c_2 \sqrt{\frac{\log(2N/\varepsilon)}{k}} + c_2 \sqrt{\frac{\log(2N/\varepsilon)}{n}}$$

$$+ c_3 \frac{\log(2N/\varepsilon)}{k} + c_3 \frac{\log(2N/\varepsilon)}{n}, \quad (60)$$

where $c_1, c_2, c_3 > 0$ are numerical constants from the empirical-Bernstein inequality.

**Center-wise population gap with explicit $k$-terms.** Insert Eq. (60) into Eq. (56); upper bound configuration-dependent constants by $\rho_{a_i} \leq \rho_{\max}$, $\kappa_{a_i} \leq \kappa_{\max}$, and $C_{2,a_i} \geq C_{2,\min} = (1 - \rho_{\max})/L_R$. Using the shorthands $C_G^+$ and $\widetilde{C}_G^+$ and the inequality $\log(2N/\varepsilon) \leq \log(2/\varepsilon) + \mathcal{H}_{\mathrm{cov}}(r)$, we separate the $k$–independent (floor) terms:

$$\epsilon_{\mathrm{bound}} := L_R \rho_{\max}^T \|\delta_0\| + \frac{\eta \kappa_{\max}}{C_{2,\min}} \sup_{a \in \mathcal{C}} \Delta_a^\star + \frac{2\eta \kappa_{\max}^2}{C_{2,\min}} \widetilde{C}_G^+ \frac{1}{\sqrt{n}} + 2 B_\ell \sqrt{\frac{2 \log(4N/\varepsilon)}{m}}, \quad (61)$$

and collect the distilled-side $k$–dependence as (for some absolute constants $\bar{c}_1, \bar{c}_2 > 0$)

$$\left| R_\nu(\theta_T^{(\hat{\mu}_s, a_i)}) - R_\nu(\theta_T^{(\hat{\mu}_\tau, a_i)}) \right| \leq \epsilon_{\text{bound}} + \underbrace{\bar{c}_1 \frac{\eta \kappa_{\max}}{C_{2,\min}} \frac{\mathcal{H}_{\text{cov}}(r)}{k}}_{\text{Bernstein linear term}}$$

$$+ \underbrace{\bar{c}_2 \frac{\eta \kappa_{\max}}{C_{2,\min}} \left( C_G^+ + B_g \sqrt{\mathcal{H}_{\text{cov}}(r)} \right) \frac{1}{\sqrt{k}}}_{\text{RC and sub-Gaussian term}} . \tag{62}$$

Here we used that $\mathfrak{R}_k(\mathcal{G}_{a_i}) \leq \sup_a \mathfrak{R}_k(\mathcal{G}_a)$ and $\sqrt{\log(2N/\varepsilon)} \lesssim \sqrt{\log(2/\varepsilon)} + \sqrt{\mathcal{H}_{\text{cov}}(r)}$, and absorbed numerical constants into $(\bar{c}_1, \bar{c}_2)$.

**Step D.2.2 Averaging centers against the prior $\Pi$.** Let $i(a) \in [N]$ be the index of the cover center assigned to $a$ (measurable selection with $d_{\mathcal{A}}(a, a_i) \leq r$). Define the cell masses $p_i := \Pi(\{a \in \mathcal{A} : i(a) = i\})$ so that $\sum_{i=1}^N p_i = 1$ and $\mathbb{E}_{a \sim \Pi}[\cdot] = \sum_{i=1}^N p_i \, \mathbb{E}_{a \sim \Pi(\cdot | i(a) = i)}[\cdot]$.

Taking expectation over $a \sim \Pi$ and using Eq. (62) evaluated at $i(a)$ yields

$$\mathbb{E}_{a \sim \Pi} \left| R_\nu(\theta_T^{(\hat{\mu}_s, a_{i(a)})}) - R_\nu(\theta_T^{(\hat{\mu}_\tau, a_{i(a)})}) \right| = \sum_{i=1}^N p_i \left| R_\nu(\theta_T^{(\hat{\mu}_s, a_i)}) - R_\nu(\theta_T^{(\hat{\mu}_\tau, a_i)}) \right|$$

$$\leq \epsilon_{\text{bound}} + \bar{c}_1 \frac{\eta \kappa_{\max}}{C_{2,\min}} \frac{\mathcal{H}_{\text{cov}}(r)}{k} + \bar{c}_2 \frac{\eta \kappa_{\max}}{C_{2,\min}} \left( C_G^+ + B_g \sqrt{\mathcal{H}_{\text{cov}}(r)} \right) \frac{1}{\sqrt{k}}, \tag{63}$$

because the right-hand side of Eq. (62) is independent of the particular cell beyond its index $i$ and $(p_i)$ sums to 1. We now transfer from the center $a_{i(a)}$ back to the original configuration $a$.

**Step D.2.3 Prior-averaged transfer from centers to arbitrary configurations (population risk).** For each $a$, consider the parameter deviations at time $T$ under the same training distribution $\mu$:

$$\Delta_T^{(\mu)}(a) := \theta_T^{(\mu, a)} - \theta_T^{(\mu, a_{i(a)})}.$$

By the cross-configuration one-step recursion (same $\mu$, different configurations) and configuration-Lipschitz mismatch,

$$\|\Delta_{t+1}^{(\mu)}(a)\| \leq \rho_{\max} \|\Delta_t^{(\mu)}(a)\| + \eta L_{\text{conf}} \, d_{\mathcal{A}}(a, a_{i(a)}),$$

and because $\Delta_0^{(\mu)}(a) = 0$ (same initialization), we obtain

$$\|\Delta_T^{(\mu)}(a)\| \leq \frac{\eta L_{\text{conf}}}{1 - \rho_{\max}} \, d_{\mathcal{A}}(a, a_{i(a)}) \leq \frac{\eta L_{\text{conf}}}{1 - \rho_{\max}} \, r := C_{\text{path}} \, r. \tag{64}$$

By $L_R$–Lipschitz continuity of the risk $\hat{R}$,

$$\left| \hat{R}(\theta_T^{(\hat{\mu}_s, a)}) - \hat{R}(\theta_T^{(\hat{\mu}_s, a_{i(a)})}) \right| \leq L_R C_{\text{path}} \, r, \qquad \left| \hat{R}(\theta_T^{(\hat{\mu}_\tau, a)}) - \hat{R}(\theta_T^{(\hat{\mu}_\tau, a_{i(a)})}) \right| \leq L_R C_{\text{path}} \, r. \tag{65}$$

Hence, by triangle inequality,

$$\left| \hat{R}(\theta_T^{(\hat{\mu}_s, a)}) - \hat{R}(\theta_T^{(\hat{\mu}_\tau, a)}) \right| \leq \left| \hat{R}(\theta_T^{(\hat{\mu}_s, a_{i(a)})}) - \hat{R}(\theta_T^{(\hat{\mu}_\tau, a_{i(a)})}) \right| + 2 L_R C_{\text{path}} \, r. \tag{66}$$

Taking expectation over $a \sim \Pi$ and invoking Eq. (63),

$$\mathbb{E}_{a \sim \Pi} \left| R_\nu(\theta_T^{(\hat{\mu}_s, a)}) - R_\nu(\theta_T^{(\hat{\mu}_\tau, a)}) \right| \leq \mathbb{E}_{a \sim \Pi} \left| R_\nu(\theta_T^{(\hat{\mu}_s, a_{i(a)})}) - R_\nu(\theta_T^{(\hat{\mu}_\tau, a_{i(a)})}) \right| + 2 L_R C_{\text{path}} \, r$$

$$\leq \epsilon_{\text{bound}} + \bar{c}_1 \frac{\eta \kappa_{\max}}{C_{2,\min}} \frac{\mathcal{H}_{\text{cov}}(r)}{k} + \bar{c}_2 \frac{\eta \kappa_{\max}}{C_{2,\min}} \left( C_G^+ + B_g \sqrt{\mathcal{H}_{\text{cov}}(r)} \right) \frac{1}{\sqrt{k}} + 2 L_R C_{\text{path}} \, r. \tag{67}$$

Since $r$ is fixed in the covering argument, we absorb the additive constant $2 L_R C_{\text{path}} r$ into $\epsilon_{\text{bound}}$ (redefining it harmlessly). This proves Eq. (50) with

$$A_1 := \bar{c}_1 \frac{\eta \kappa_{\max}}{C_{2,\min}}, \qquad A_2 := \bar{c}_2 \frac{\eta \kappa_{\max}}{C_{2,\min}} \left( C_G^+ / \sqrt{\mathcal{H}_{\text{cov}}(r)} + B_g \right)$$

i.e. more transparently,

$$A_1 = \Theta\left( \frac{\eta \kappa_{\max}}{C_{2,\min}} \right), \qquad A_2 = \Theta\left( \frac{\eta \kappa_{\max}}{C_{2,\min}} \right) \cdot \left( C_G^+ / \sqrt{\mathcal{H}_{\text{cov}}(r)} + B_g \right).$$

**Mutual-information (MI) corrections: two consistent variants (High-probability variant).** If the distilled dataset $\mathcal{D}_s$ can depend on the real dataset $\mathcal{D}_\tau$, the distilled-side sampling deviation in Eq. (60) should be replaced by a high-probability information-theoretic tail (e.g., Bu et al., 2020, theorem 7; cf. Xu & Raginsky, 2017; Steinke & Zakynthinou, 2020). There exists a universal constant $C_I' > 0$ such that, with probability at least $1 - \varepsilon$,

$$\sup_{i \in [N]} \sup_{\theta \in \Gamma} \left\| \mathbb{E}_{\hat{\mu}_s} g_{a_i}(\theta; Z) - \mathbb{E}_{\mu_s} g_{a_i}(\theta; Z) \right\| \le \mathfrak{R}_k(\mathcal{G}_{a_i}) + \frac{C_I'}{\sqrt{k}} \sqrt{I(\mathcal{D}_s; \mathcal{D}_\tau) + \log \frac{4N}{\varepsilon}}.$$

Propagating this replacement through Eq. (60)–Eq. (67) adds

$$+ \frac{\widetilde{C}_I}{\sqrt{k}} \sqrt{I(\mathcal{D}_s; \mathcal{D}_\tau)}$$

to the right-hand side of Eq. (50), for some $\widetilde{C}_I = \Theta(\eta \kappa_{\max} / C_{2,\min})$.

**(In-expectation variant).** If one states the result *in expectation* over $(\hat{\mu}_\tau, \hat{\mu}_s, \hat{\nu})$ (dropping the $1 - \varepsilon$ qualifier), expected MI generalization bounds (e.g., Xu & Raginsky, 2017; Russo & Zou, 2016) yield a linear penalty

$$+ \frac{C_I}{k} I(_s;_\tau),$$

with $C_I = \Theta(\eta \kappa_{\max} / C_{2,\min})$. The rate in $\mathcal{H}_{\mathrm{cov}}(r)$ remains the same in both variants.

**Addtional interpretations to Theorem 5.1** Combining the uniform bound Eq. (34) and the prior-averaged bound Eq. (50) yields Theorem 5.1.

**Floor terms.** In the uniform case, the floor term $\epsilon_{\mathrm{bound}}^{\mathrm{uni}}$ is given in Eq. (41). It aggregates all $k$-independent contributions: the transient term $L_R \rho_{\max}^T \|\delta_0\|$, the worst-case intrinsic alignment $\sup_{a \in \mathcal{A}} \Delta_a^\star$, the $n$-side deviation $\widetilde{C}_G^+$, and the test-sample concentration term. In the averaged case, the corresponding floor $\epsilon_{\mathrm{bound}}^{\mathrm{avg}}$ in Eq. (61) is structurally the same but uses the prior-averaged intrinsic alignment $\Delta_\sharp^\star = \mathbb{E}_{a \sim \Pi} \Delta_a^\star$ instead of the supremum.

**Coverage-dependent terms.** In the uniform inequality the constant $C_{\mathrm{cov}}(\mathcal{A}, r)$ Eq. (48) multiplies $1/\sqrt{k}$ and captures the dependence on the covering complexity $\mathcal{H}_{\mathrm{cov}}(r)$. It grows with both the Rademacher complexity $C_G^+$ and the envelope term $B_g \sqrt{\mathcal{H}_{\mathrm{cov}}(r)}$. In the averaged inequality the constants $(A_1, A_2)$ appear in Eq. (50), where $A_1 \mathcal{H}_{\mathrm{cov}}(\mathcal{A}, r)/k$ comes from the linear (Bernstein) tail $\log(N/\varepsilon)/k$, while $A_2 \sqrt{\mathcal{H}_{\mathrm{cov}}(\mathcal{A}, r)/k}$ arises from the Rademacher and sub-Gaussian deviations.

**Why only the distilled side scales with $\mathcal{H}_{\mathrm{cov}}(r)$.** The dependence on the covering number comes solely from the distilled side, which requires a union bound across the $N = \exp(\mathcal{H}_{\mathrm{cov}}(r))$ cover centers. On the real-data side, all configurations share the same empirical distribution $\hat{\mu}_\tau$, so no union is needed. Consequently, $n$-side deviations remain independent of $\mathcal{H}_{\mathrm{cov}}(r)$ and are absorbed into the floor terms.

**Choice of intrinsic alignment.** The uniform bound requires the worst-case intrinsic alignment $\sup_{a \in \mathcal{A}} \Delta_a^\star$, while the averaged bound admits the weaker prior-averaged quantity $\Delta_\sharp^\star$. This separation avoids introducing the looser maximum $\max\{\Delta_\sharp^\star, \sup_a \Delta_a^\star\}$ and keeps each statement as tight as possible for its regime.

**Mutual-information correction.** When the distilled dataset $\mathcal{D}_s$ depends on the real dataset $\mathcal{D}_\tau$, the distilled-side deviation requires an additional correction. In the high-probability setting, one obtains an additive penalty of order $\frac{C_I'}{\sqrt{k}} \sqrt{I(\mathcal{D}_s; \mathcal{D}_\tau)}$ in both uniform and averaged inequalities. In the in-expectation setting, one instead obtains a linear penalty $\frac{C_I}{k} I(\mathcal{D}_s; \mathcal{D}_\tau)$. In either case the rates $\sqrt{\mathcal{H}_{\mathrm{cov}}/k}$ and $\mathcal{H}_{\mathrm{cov}}/k$ remain unaffected.

**On the covering radius.** The covering radius $r$ is fixed throughout, and $\mathcal{H}_{\mathrm{cov}}(r)$ always refers to the coverage complexity at that scale. Optimizing $r$ affects only the constants but not the asymptotic rates $\sqrt{\mathcal{H}_{\mathrm{cov}}/k}$ or $\mathcal{H}_{\mathrm{cov}}/k$.

### D.3 PROOF OF COROLLARY 5.3

We now derive the corollary in Section 5 directly from the uniform bound in Appendix D.

**Corollary D.3** (Coverage Law (required $k$ at a fixed error)). *For any $\epsilon_0 > \epsilon_{\text{bound}}$,*

$$\sup_{a \in \mathcal{A}} \left| R_\nu(\theta_T^{(s,a)}) - R_\nu(\theta_T^{(\tau,a)}) \right| \leq \epsilon_0 \quad \Longleftarrow \quad k \geq K_{\min}(\epsilon_0, \mathcal{A}) = \left( \frac{C_{\text{cov}}(\mathcal{A})}{\epsilon_0 - \epsilon_{\text{bound}}} \right)^2 = \Theta\big(\mathcal{H}_{\text{cov}}(\mathcal{A}, r)\big). \tag{68}$$

*Thus,* doubling configuration diversity doubles the required distilled size.

*Proof.* Fix any target $\epsilon_0 > \epsilon_{\text{bound}}$. A sufficient condition for $\sup_{a \in \mathcal{C}} |\hat{R}(\theta_T^{(s,a)}) - \hat{R}(\theta_T^{(\tau,a)})| \leq \epsilon_0$ is that the $k$–dependent term in Eq. (50) is at most $\epsilon_0 - \epsilon_{\text{bound}}$:

$$\frac{C_{\text{cov}}(\mathcal{A})}{\sqrt{k}} \leq \epsilon_0 - \epsilon_{\text{bound}}. \tag{69}$$

Since $C_{\text{cov}}(\mathcal{A}) \geq 0$ and $\epsilon_0 - \epsilon_{\text{bound}} > 0$, Eq. (69) is equivalent to

$$k \geq \left( \frac{C_{\text{cov}}(\mathcal{A})}{\epsilon_0 - \epsilon_{\text{bound}}} \right)^2 =: K_{\min}(\epsilon_0, \mathcal{A}), \tag{70}$$

which proves the first displayed formula.

It remains to show $K_{\min}(\epsilon_0, \mathcal{A}) = \Theta(\mathcal{H})$. Using Eq. (48) and the elementary inequality $(x+y)^2 \leq 2x^2 + 2y^2$ for $x, y \geq 0$,

$$K_{\min}(\epsilon_0, \mathcal{A}) = \frac{1}{(\epsilon_0 - \epsilon_{\text{bound}})^2} \left[ \frac{\eta \, \kappa_{\max}}{C_{2,\min}} \left( 2 \, C_G^+ + 2 \, B_g \, \sqrt{2 \, \mathcal{H}} \right) \right]^2 \tag{71}$$

$$\leq \frac{1}{(\epsilon_0 - \epsilon_{\text{bound}})^2} \left( \frac{\eta \, \kappa_{\max}}{C_{2,\min}} \right)^2 \cdot 2 \left[ (2 \, C_G^+)^2 + \left( 2 \, B_g \, \sqrt{2 \, \mathcal{H}} \right)^2 \right]$$

$$= \underbrace{\frac{8 \, \eta^2 \, \kappa_{\max}^2 \, (C_G^+)^2}{C_{2,\min}^2 \, (\epsilon_0 - \epsilon_{\text{bound}})^2}}_{=: \, C_{\text{up},0}} + \underbrace{\frac{8 \, \eta^2 \, \kappa_{\max}^2 \, B_g^2}{C_{2,\min}^2 \, (\epsilon_0 - \epsilon_{\text{bound}})^2}}_{=: \, C_{\text{up},1}} \, \mathcal{H}.$$

Hence $K_{\min}(\epsilon_0, \mathcal{A}) \leq C_{\text{up},0} + C_{\text{up},1} \, \mathcal{H}$ for all $\mathcal{H} \geq 0$, i.e., $K_{\min} = O(\mathcal{H})$.

For a matching lower bound, since $x \mapsto x^2$ is monotone on $x \geq 0$ and $2 \, C_G^+ \geq 0$,

$$\left( 2 \, C_G^+ + 2 \, B_g \, \sqrt{2 \, \mathcal{H}} \right)^2 \geq \left( 2 \, B_g \, \sqrt{2 \, \mathcal{H}} \right)^2 = 8 \, B_g^2 \, \mathcal{H}.$$

Therefore,

$$K_{\min}(\epsilon_0, \mathcal{A}) \geq \frac{1}{(\epsilon_0 - \epsilon_{\text{bound}})^2} \left( \frac{\eta \, \kappa_{\max}}{C_{2,\min}} \right)^2 \cdot 8 \, B_g^2 \, \mathcal{H} =: C_{\text{low}} \, \mathcal{H}. \tag{72}$$

Combining Eq. (refeq:K-upper)–Eq. (72), we obtain $C_{\text{low}} \, \mathcal{H} \leq K_{\min}(\epsilon_0, \mathcal{A}) \leq C_{\text{up},0} + C_{\text{up},1} \, \mathcal{H}$. In particular, for all $\mathcal{H} \geq 1$, $K_{\min}(\epsilon_0, \mathcal{A}) \leq (C_{\text{up},0} + C_{\text{up},1}) \, \mathcal{H}$, so $K_{\min}(\epsilon_0, \mathcal{A}) = \Theta(\mathcal{H})$. $\square$

### D.4 PROOF OF THE COVERAGE LOWER BOUND (THEOREM 5.2)

**Standing assumptions.** We use Assumption 4.1, the identifiability condition in Theorem 5.2, a $\rho$–packing $\{a_1, \ldots, a_M\} \subset (\mathcal{C}, d_{\mathcal{A}})$ with $M = \exp(\mathcal{H}(\mathcal{A}))$, and the uniform envelopes in App. D. We also use the configuration-Lipschitz transfer (Assumption D.2, Eq. (26)) to pass alignment statements across configurations. For each $a \in \mathcal{A}$, $\|g_a(\theta; z)\| \leq B_g$, $\|P_a(\theta)\| \leq \kappa_a \leq \kappa_{\max}$ on $\Gamma$; the PL-type dynamics are contractive with rate $\rho_a \in (0, 1)$, and we set $C_{2,a} = (1 - \rho_a)/L_R$ and $C_{2,\min} = \min_a C_{2,a}$.

**Step D.4.1 Packing and testing prior.** Pick a $\rho$–packing $\{a_i\}_{i=1}^M$; let the hidden index $U$ be uniform on $[M]$ and $a_U$ the evaluation configuration. The distillation algorithm Alg maps $\mathcal{D}_\tau \sim q_\tau^n$ to $\hat{\mu}_s \in \mathcal{P}_k$ and does not observe $U$.

**Step D.4.2 Small risk gap $\Rightarrow$ small alignment (risk-to-alignment).** For fixed $a$, the single-configuration forward inequality (from PL contractivity unrolled over $T$ steps) gives

$$\left| \hat{R}(\theta_T^{(s,a)}) - \hat{R}(\theta_T^{(\tau,a)}) \right| \leq L_R \rho_a^T \|\delta_0\| + \frac{\eta \kappa_a}{C_{2,a}} G_a, \qquad G_a := \sup_{\theta \in \Gamma} \left\| \mathbb{E}_{\hat{\mu}_s} g_a(\theta; Z) - \mathbb{E}_{\hat{\mu}_\tau} g_a(\theta; Z) \right\|. \tag{73}$$

Hence, if $\left| \hat{R}(\theta_T^{(s,a)}) - \hat{R}(\theta_T^{(\tau,a)}) \right| \leq L_R \rho_a^T \|\delta_0\| + \epsilon$, then $G_a \leq (C_{2,a}/(\eta \kappa_a)) \epsilon$ and, using $\|P_a\| \leq \kappa_a$,

$$\Delta_a(\hat{\mu}_\tau, \hat{\mu}_s) = \sup_{\theta \in \Gamma} \left\| P_a(\mathbb{E}_{\hat{\mu}_\tau} g_a - \mathbb{E}_{\hat{\mu}_s} g_a) \right\| \leq \kappa_a G_a \leq \frac{C_{2,a}}{\eta} \epsilon \leq \frac{C_{2,\min}}{\eta} \epsilon. \tag{74}$$

**Step D.4.3 Identifiability + configuration-Lipschitz $\Rightarrow$ pairwise lower bound and decoder.** For any distinct $a_i, a_j$ and any $\theta \in \Gamma$,

$$\|P_{a_i} \mathbb{E}_{\hat{\mu}_\tau} g_{a_i} - P_{a_j} \mathbb{E}_{\hat{\mu}_\tau} g_{a_j}\| \leq \underbrace{\|P_{a_i}(\mathbb{E}_{\hat{\mu}_\tau} g_{a_i} - \mathbb{E}_{\hat{\mu}_s} g_{a_i})\|}_{=\Delta_{a_i}} + \underbrace{\|P_{a_i} \mathbb{E}_{\hat{\mu}_s} g_{a_i} - P_{a_j} \mathbb{E}_{\hat{\mu}_s} g_{a_j}\|}_{\leq L_{\text{conf}} d_{\mathcal{A}}(a_i, a_j)}$$
$$+ \underbrace{\|P_{a_j}(\mathbb{E}_{\hat{\mu}_s} g_{a_j} - \mathbb{E}_{\hat{\mu}_\tau} g_{a_j})\|}_{=\Delta_{a_j}}.$$

By identifiability at $\mu_\tau$, $\|P_{a_i} \mathbb{E}_{\hat{\mu}_\tau} g_{a_i} - P_{a_j} \mathbb{E}_{\hat{\mu}_\tau} g_{a_j}\| \geq \lambda \, d_{\mathcal{A}}(a_i, a_j)$. Using $d_{\mathcal{A}}(a_i, a_j) \geq \rho$ (packing) and maximizing over $\theta$,

$$\Delta_{a_i} + \Delta_{a_j} \geq (\lambda - L_{\text{conf}}) \, d_{\mathcal{A}}(a_i, a_j) \geq (\lambda - L_{\text{conf}}) \rho. \tag{75}$$

Choosing (or refining) the packing so that $L_{\text{conf}} \leq \lambda/2$ yields

$$\min\{\Delta_{a_i}, \Delta_{a_j}\} \geq \frac{\lambda \rho}{4}. \tag{76}$$

Consequently, if for the true configuration $a_U$ we have $\Delta_{a_U} \leq (\lambda \rho)/8$, then $\min_{i \neq U} \Delta_{a_i} \geq (\lambda \rho)/4 > (\lambda \rho)/8$, and the decoder

$$\widehat{U}(\hat{\mu}_s) \in \arg\min_{i \in [M]} \Delta_{a_i}(\hat{\mu}_\tau, \hat{\mu}_s) \tag{77}$$

is correct (ties broken deterministically). Combining Eq. (74) with $\Delta_{a_U} \leq (\lambda \rho)/8$ shows that the decoder succeeds whenever

$$\epsilon \leq \frac{\eta}{8 C_{2,\min}} \lambda \rho. \tag{78}$$

**Step D.4.4 information tail + union bound.** A high-probability mutual-information tail (e.g., Bu et al. 2020; Xu & Raginsky 2017; Steinke & Zakynthinou 2020) implies that for any fixed $a$ and any $\varepsilon \in (0, 1)$,

$$\Pr\left\{ \Delta_a(\hat{\mu}_\tau, \hat{\mu}_s) \leq c_0 + \sqrt{\frac{C_I}{k}\left(I(\mathcal{D}_s; \mathcal{D}_\tau) + \log \frac{1}{\varepsilon}\right)} \right\} \geq 1 - \varepsilon, \tag{79}$$

where $c_0$ aggregates $k$–independent terms (intrinsic alignment, real-side sampling, test deviation), absorbed into $\epsilon_{\text{bound}}$. Setting $\varepsilon = 1/(4M)$ and union-bounding over the $M$ packed configurations yields

$$\Pr\left\{ \max_{i \in [M]} \Delta_{a_i} \leq c_0 + \sqrt{\frac{C_I}{k}\left(I(\mathcal{D}_s; \mathcal{D}_\tau) + \log(4M)\right)} \right\} \geq \frac{3}{4}. \tag{80}$$

Thus, to ensure that with probability at least $3/4$ we have $\Delta_{a_i} \leq \tilde{\epsilon}$ for *all* $i$, it is necessary that

$$\tilde{\epsilon} \geq c_0 + \sqrt{\frac{C_I}{k}\left(I(\mathcal{D}_s; \mathcal{D}_\tau) + \log(4M)\right)} = \Omega\left(\sqrt{\frac{\mathcal{H}(\mathcal{A})}{k}}\right), \tag{81}$$

since $M = \exp(\mathcal{H}(\mathcal{A}))$ and $c_0$ is $k$–independent.

**Conclusion.** Assume the algorithm attains, with probability at least $3/4$, the small risk gap at every packed configuration:

$$\left|\hat{R}(\theta_T^{(s,a_i)}) - \hat{R}(\theta_T^{(\tau,a_i)})\right| \leq \epsilon_{\text{bound}} + \epsilon, \qquad \forall i \in [M]. \tag{82}$$

Then by Eq. (74) we have simultaneously for all $i$ $\Delta_{a_i} \leq (C_{2,\min}/\eta)\,\epsilon$. Comparing with the necessary condition Eq. (81), we obtain

$$\epsilon \geq \frac{\eta}{C_{2,\min}} \sqrt{\frac{C_I}{k}\left(I(\mathcal{D}_s;\mathcal{D}_\tau) + \log(4M)\right)} = \Omega\!\left(\sqrt{\frac{\mathcal{H}(\mathcal{A})}{k}}\right). \tag{83}$$

Averaging over the uniform prior on the packing and absorbing all $k$–independent contributions into $\epsilon_{\text{bound}}$ therefore yields

$$\mathbb{E}_a\left|R_\nu(\theta_T^{(s,a)}) - R_\nu(\theta_T^{(\tau,a)})\right| \geq \epsilon_{\text{bound}} + c_{\text{lb}}\sqrt{\frac{\mathcal{H}(\mathcal{A})}{k}},$$

for some numerical $c_{\text{lb}} \in (0,1)$. Equivalently, to achieve any target $\epsilon_0 > \epsilon_{\text{bound}}$,

$$k \geq \Omega\!\left(\frac{\mathcal{H}(\mathcal{A})}{(\epsilon_0 - \epsilon_{\text{bound}})^2}\right).$$

# E  PROOFS FOR UNIFYING DISTRIBUTION, GRADIENT, AND TRAJECTORY MATCHING

## E.1  SETUP AND RECALL ASSUMPTIONS

Fix a training configuration $a$ with feasible set $\Gamma_a$. The inner update follows

$$\theta_{t+1} = \Phi_a(\theta_t; \mu) = \theta_t - \eta\, P_a(\theta_t)\, \mathbb{E}_\mu g_a(\theta_t; z), \qquad t = 0,1,\ldots, \tag{84}$$

and the outer variable $\xi$ parameterizes the synthetic distribution $\mu(\xi)$.

We assume the same *single-configuration regularity assumptions* used in section 4:

(i) $\|g_a(\theta; z)\| \leq B_g$ and $\|P_a(\theta)\| \leq \kappa_a$ on $\Gamma_a$;

(ii) the loss is $L_R$-Lipschitz in $\theta$;

(iii) the inner dynamics are contractive in the PL/smooth regime with factor $\rho_a \in (0,1)$, and $C_{2,a} = (1-\rho_a)/L_R$.

The alignment discrepancy is

$$\Delta_a(\mu,\nu) := \sup_{\theta \in \Gamma_a} \left\|P_a(\theta)\big(\mathbb{E}_\mu g_a(\theta; z) - \mathbb{E}_\nu g_a(\theta; z)\big)\right\|.$$

We also use the empirical risks $\hat{R}(\theta) = \mathbb{E}_{\hat{\nu}}\,\ell(\theta; z)$ and the risk-Lipschitz lemma $|\hat{R}(\theta) - \hat{R}(\theta')| \leq L_R\|\theta - \theta'\|$. Let $\delta_t := \theta_t^{(s,a)} - \theta_t^{(\tau,a)}$ denote the in-configuration parameter gap when training on $\hat{\mu}_s$ vs. $\hat{\mu}_\tau$.

**Outer surrogate contraction**

Let $M_\phi : \Xi \to \mathbb{R}_{\geq 0}$ be the outer surrogate for branch $\phi \in \{\text{DM}, \text{GM}, \text{TM}\}$ as listed in section 6 (DM: $W_1$ or $\text{MMD}_k$; GM: anchor-averaged squared field gap; TM: weighted path discrepancy over an $L_b$-step unroll). The outer variable $\xi \in \Xi$ parameterizes the synthetic distribution $\mu(\xi)$, and the outer update is

$$\xi^{(j+1)} = \xi^{(j)} - \eta_j\, \widehat{\nabla} M_\phi(\xi^{(j)}),$$

where $\widehat{\nabla} M_\phi(\xi^{(j)})$ denotes the (possibly stochastic) estimator of the exact gradient $\nabla M_\phi(\xi^{(j)})$ produced by mini-batching critics (DM), anchor/path sampling (GM/TM), or finite unrolls.

We assume throughout: ($L_\phi$-**smoothness in** $\xi$) $M_\phi$ is $L_\phi$-smooth:

$$M_\phi(y) \leq M_\phi(x) + \langle \nabla M_\phi(x), y - x\rangle + \frac{L_\phi}{2}\|y - x\|^2 \qquad \forall x, y \in \Xi.$$

This is the standard descent lemma for $L$-smooth functions (Nesterov, 2013; Bubeck et al., 2015).

(**PL condition for** $M_\phi$) There exists $\mu_\phi > 0$ s.t.

$$\frac{1}{2}\|\nabla M_\phi(\xi)\|^2 \geq \mu_\phi\big(M_\phi(\xi) - M_\phi^\star\big), \qquad M_\phi^\star := \inf_\xi M_\phi(\xi).$$

This is the Polyak–Łojasiewicz (PL) inequality used in the paper for the outer loop; see Eq. 13

(**Estimator model**) Write the gradient estimator as

$$\widehat{\nabla} M_\phi(\xi) = \nabla M_\phi(\xi) + e(\xi),$$

where $e(\xi)$ captures the randomness due to critics, anchors, finite unrolls, etc.

We will consider two subcases:

(i) *Unbiased finite-variance:* $\mathbb{E}[e(\xi) \mid \xi] = 0$ and $\mathbb{E}[\|e(\xi)\|^2 \mid \xi] \leq \sigma_\phi^2$.

(ii) *Biased-but-controlled:* $\|\mathbb{E}[e(\xi) \mid \xi]\| \leq \beta_\phi$ and $\mathbb{E}[\|e(\xi) - \mathbb{E}e(\xi)\|^2 \mid \xi] \leq \sigma_\phi^2$.

These two settings cover mini-batch $W_1$/MMD critics (variance) and approximate/implicitbackprop through unrolls (small bias).

### E.2 PROOF OF THE CONTRACTION PROPERTY EQ. (13)

We first provide a detailed proof of the contraction property. The goal is to show that the outer-loop update of the surrogate objective $M_\phi$ admits a linear contraction up to a fixed estimator floor, thereby establishing Eq. (13).

**Step E.2.1 Apply the descent lemma at the actual update.** At iteration $b$, the outer-loop update is given by

$$\xi^{(j+1)} = \xi^{(j)} - \eta_j \widehat{\nabla} M_\phi(\xi^{(j)}),$$

where $\eta_j$ is the step size and $\widehat{\nabla} M_\phi(\xi^{(j)})$ is the stochastic estimator of the true gradient. Since $M_\phi$ has $L_\phi$-Lipschitz gradients, the descent lemma ensures that for any point $y$,

$$M_\phi(y) \leq M_\phi(\xi^{(j)}) + \langle \nabla M_\phi(\xi^{(j)}), y - \xi^{(j)} \rangle + \frac{L_\phi}{2}\|y - \xi^{(j)}\|^2.$$

Substituting $y = \xi^{(j+1)}$ yields

$$M_\phi(\xi^{(j+1)}) \leq M_\phi(\xi^{(j)}) - \eta_j \langle \nabla M_\phi(\xi^{(j)}), \widehat{\nabla} M_\phi(\xi^{(j)}) \rangle + \frac{L_\phi}{2}\eta_j^2 \|\widehat{\nabla} M_\phi(\xi^{(j)})\|^2. \tag{85}$$

This inequality expresses how the function value decreases after one update, up to a quadratic correction controlled by $L_\phi$.

**Step E.2.2 Expand the estimator and regroup.** We next separate the stochastic gradient estimator into the true gradient plus an error term:

$$\widehat{\nabla} M_\phi(\xi^{(j)}) = \nabla M_\phi(\xi^{(j)}) + e(\xi^{(j)}).$$

Substituting into Eq. 85, we expand and regroup terms:

$$M_\phi(\xi^{(j+1)}) \leq M_\phi(\xi^{(j)}) - \eta_j \|\nabla M_\phi(\xi^{(j)})\|^2 - \eta_j \langle \nabla M_\phi(\xi^{(j)}), e(\xi^{(j)}) \rangle$$
$$+ \frac{L_\phi}{2}\eta_j^2 \Big( \|\nabla M_\phi(\xi^{(j)})\|^2 + 2\langle \nabla M_\phi(\xi^{(j)}), e(\xi^{(j)}) \rangle + \|e(\xi^{(j)})\|^2 \Big).$$

Collecting like terms gives the compact form:

$$M_\phi(\xi^{(j+1)}) \leq M_\phi(\xi^{(j)}) - \eta_j \Big(1 - \frac{L_\phi}{2}\eta_j\Big) \|\nabla M_\phi(\xi^{(j)})\|^2$$
$$+ \big(-\eta_j + L_\phi \eta_j^2\big) \langle \nabla M_\phi(\xi^{(j)}), e(\xi^{(j)}) \rangle + \frac{L_\phi}{2}\eta_j^2 \|e(\xi^{(j)})\|^2. \tag{86}$$

This decomposition highlights three distinct effects: (i) a contraction term proportional to $\|\nabla M_\phi\|^2$, (ii) a cross-term coupling gradient and error, and (iii) a pure variance term $\|e\|^2$.

**Step E.2.3 Take conditional expectation to remove the cross term.** We now take conditional expectation given $\xi^{(j)}$, analyzing two regimes of the estimator.

**Case (i): unbiased estimator.** Suppose $\mathbb{E}[e(\xi^{(j)}) \mid \xi^{(j)}] = 0$ and $\mathbb{E}[\|e(\xi^{(j)})\|^2 \mid \xi^{(j)}] \leq \sigma_\phi^2$. The cross term vanishes in expectation, leaving

$$\mathbb{E}\big[M_\phi(\xi^{(j+1)}) \mid \xi^{(j)}\big] \leq M_\phi(\xi^{(j)}) - \eta_j\Big(1 - \tfrac{L_\phi}{2}\eta_j\Big)\|\nabla M_\phi(\xi^{(j)})\|^2 + \tfrac{L_\phi}{2}\eta_j^2\sigma_\phi^2. \tag{87}$$

**Case (ii): biased but controlled estimator.** We decompose the estimation error into a deterministic bias and a zero-mean noise conditioned on $\xi^{(j)}$:

$$e(\xi^{(j)}) = \bar{e}(\xi^{(j)}) + \tilde{e}(\xi^{(j)}), \qquad \bar{e}(\xi^{(j)}) := \mathbb{E}\Big[e(\xi^{(j)}) \,\Big|\, \xi^{(j)}\Big], \quad \mathbb{E}\Big[\tilde{e}(\xi^{(j)}) \,\Big|\, \xi^{(j)}\Big] = 0.$$

Assume the bias is bounded and the conditional noise has bounded second moment:

$$\|\bar{e}(\xi^{(j)})\| \leq \beta_\phi, \qquad \mathbb{E}\Big[\|\tilde{e}(\xi^{(j)})\|^2 \,\Big|\, \xi^{(j)}\Big] \leq \sigma_\phi^2.$$

**Take conditional expectation and separate terms.** Conditioning on $\xi^{(j)}$ in Eq. 86, the cross term splits as

$$\mathbb{E}\Big[\big\langle\nabla M_\phi(\xi^{(j)}), e(\xi^{(j)})\big\rangle \,\Big|\, \xi^{(j)}\Big] = \big\langle\nabla M_\phi(\xi^{(j)}), \bar{e}(\xi^{(j)})\big\rangle + \underbrace{\mathbb{E}\Big[\big\langle\nabla M_\phi(\xi^{(j)}), \tilde{e}(\xi^{(j)})\big\rangle \,\Big|\, \xi^{(j)}\Big]}_{=0},$$

where the underbraced term vanishes because $\mathbb{E}[\tilde{e}(\xi^{(j)}) \mid \xi^{(j)}] = 0$. For the quadratic error term we have

$$\mathbb{E}\Big[\|e(\xi^{(j)})\|^2 \,\Big|\, \xi^{(j)}\Big] = \|\bar{e}(\xi^{(j)})\|^2 + \mathbb{E}\Big[\|\tilde{e}(\xi^{(j)})\|^2 \,\Big|\, \xi^{(j)}\Big] \leq \beta_\phi^2 + \sigma_\phi^2.$$

Therefore,

$$\mathbb{E}\Big[M_\phi(\xi^{(j+1)}) \,\Big|\, \xi^{(j)}\Big] \leq M_\phi(\xi^{(j)}) - \eta_j\Big(1 - \tfrac{L_\phi}{2}\eta_j\Big)\|\nabla M_\phi(\xi^{(j)})\|^2$$
$$+ \big(-\eta_j + L_\phi\eta_j^2\big)\big\langle\nabla M_\phi(\xi^{(j)}), \bar{e}(\xi^{(j)})\big\rangle + \tfrac{L_\phi}{2}\eta_j^2\big(\beta_\phi^2 + \sigma_\phi^2\big). \tag{88}$$

**Bound the cross term by Young's inequality.** Using Cauchy–Schwarz and Young's inequality $ab \leq \tfrac{\tau}{2}a^2 + \tfrac{1}{2\tau}b^2$ (valid for any $\tau > 0$), we get

$$\big|\big(-\eta_j + L_\phi\eta_j^2\big)\big\langle\nabla M_\phi(\xi^{(j)}), \bar{e}(\xi^{(j)})\big\rangle\big| \leq \big|-\eta_j + L_\phi\eta_j^2\big|\,\|\nabla M_\phi(\xi^{(j)})\|\,\|\bar{e}(\xi^{(j)})\|$$
$$\leq \frac{\tau}{2}\,\|\nabla M_\phi(\xi^{(j)})\|^2 + \frac{1}{2\tau}\big(-\eta_j + L_\phi\eta_j^2\big)^2\,\|\bar{e}(\xi^{(j)})\|^2. \tag{89}$$

Substituting Eq. 89 into Eq. 88 yields, for any $\tau > 0$,

$$\mathbb{E}\Big[M_\phi(\xi^{(j+1)}) \,\Big|\, \xi^{(j)}\Big] \leq M_\phi(\xi^{(j)}) - \eta_j\Big(1 - \tfrac{L_\phi}{2}\eta_j\Big)\|\nabla M_\phi(\xi^{(j)})\|^2 + \frac{\tau}{2}\,\|\nabla M_\phi(\xi^{(j)})\|^2$$
$$+ \underbrace{\bigg(\frac{1}{2\tau}\big(-\eta_j + L_\phi\eta_j^2\big)^2 + \tfrac{L_\phi}{2}\eta_j^2\bigg)}_{\mathcal{C}_\beta(\eta_j, \tau)}\|\bar{e}(\xi^{(j)})\|^2 + \tfrac{L_\phi}{2}\eta_j^2\sigma_\phi^2. \tag{90}$$

**Choose $\tau$ and simplify the gradient coefficient.** Set $\tau = \eta_j/2$ (valid for any $\eta_j > 0$). Then $\frac{\tau}{2} = \frac{\eta_j}{4}$, so the two gradient terms combine as

$$-\eta_j\Big(1 - \tfrac{L_\phi}{2}\eta_j\Big)\|\nabla M_\phi\|^2 + \frac{\eta_j}{4}\|\nabla M_\phi\|^2 = -\Big(\eta_j - \tfrac{L_\phi}{2}\eta_j^2 - \tfrac{\eta_j}{4}\Big)\|\nabla M_\phi\|^2.$$

If we enforce the natural step-size condition $\eta_j \leq 1/L_\phi$, then

$$\eta_j - \tfrac{L_\phi}{2}\eta_j^2 \geq \tfrac{\eta_j}{2} \implies \eta_j - \tfrac{L_\phi}{2}\eta_j^2 - \tfrac{\eta_j}{4} \geq \tfrac{\eta_j}{4},$$

and hence

$$-\eta_j\Big(1 - \tfrac{L_\phi}{2}\eta_j\Big)\|\nabla M_\phi\|^2 + \frac{\eta_j}{4}\|\nabla M_\phi\|^2 \leq -\frac{\eta_j}{4}\|\nabla M_\phi(\xi^{(j)})\|^2. \tag{91}$$

**Consolidate the bias-dependent coefficient $\mathcal{C}_\beta$.** With $\tau = \eta_j/2$, we have $\frac{1}{2\tau} = \frac{1}{\eta_j}$ and therefore

$$\mathcal{C}_\beta(\eta_j, \tau) = \frac{1}{\eta_j}\left(-\eta_j + L_\phi \eta_j^2\right)^2 + \frac{L_\phi}{2}\eta_j^2.$$

Note that $\left(-\eta_j + L_\phi \eta_j^2\right)^2 = \eta_j^2\,(1 - L_\phi \eta_j)^2 \le \eta_j^2$. Thus,

$$\frac{1}{\eta_j}\left(-\eta_j + L_\phi \eta_j^2\right)^2 \le \eta_j, \qquad \text{and} \qquad \frac{L_\phi}{2}\eta_j^2 \le \frac{1}{2}\eta_j \quad \text{whenever } \eta_j \le \frac{1}{L_\phi}.$$

A slightly sharper consolidation uses

$$\frac{1}{\eta_j}\left(-\eta_j + L_\phi \eta_j^2\right)^2 + \frac{L_\phi}{2}\eta_j^2 = \eta_j\,(1 - 2L_\phi \eta_j + L_\phi^2 \eta_j^2) + \frac{L_\phi}{2}\eta_j^2 = \eta_j - \frac{3}{2}L_\phi \eta_j^2 + L_\phi^2 \eta_j^3,$$

which satisfies

$$\eta_j - \frac{3}{2}L_\phi \eta_j^2 + L_\phi^2 \eta_j^3 \le \eta_j \qquad \text{for all } \eta_j \in [0, 1/L_\phi],$$

since the cubic correction is nonpositive over this interval: $-\frac{3}{2}L_\phi \eta_j^2 + L_\phi^2 \eta_j^3 = L_\phi \eta_j^2\left(-\frac{3}{2} + L_\phi \eta_j\right) \le 0$. Therefore

$$\mathcal{C}_\beta(\eta_j, \tau)\,\|\bar{e}(\xi^{(j)})\|^2 \le \eta_j\,\|\bar{e}(\xi^{(j)})\|^2 \le \eta_j\,\beta_\phi^2. \tag{92}$$

**Collect all pieces.** Combining Eq. (90), Eq. (91), and Eq. (92), and recalling $\mathbb{E}[\|\tilde{e}(\xi^{(j)})\|^2 \mid \xi^{(j)}] \le \sigma_\phi^2$, we obtain

$$\mathbb{E}\left[M_\phi(\xi^{(j+1)}) \,\Big|\, \xi^{(j)}\right] \le M_\phi(\xi^{(j)}) - \frac{\eta_j}{4}\,\|\nabla M_\phi(\xi^{(j)})\|^2 + \eta_j\,\beta_\phi^2 + \frac{L_\phi}{2}\eta_j^2\,\sigma_\phi^2. \tag{93}$$

This is precisely the claimed biased-case inequality: the gradient term contracts with rate $\eta_j/4$, while the estimator contributes an additive floor composed of a *bias term* $\eta_j\beta_\phi^2$ and a *variance term* $\frac{L_\phi}{2}\eta_j^2\sigma_\phi^2$.

**Step E.2.4 Convert gradient norm into function gap via the PL inequality.** To turn gradient norms into function-value gaps, we invoke the Polyak–Łojasiewicz (PL) condition:

$$\|\nabla M_\phi(\xi^{(j)})\|^2 \ge 2\mu_\phi\big(M_\phi(\xi^{(j)}) - M_\phi^\star\big).$$

This inequality, weaker than strong convexity, suffices to guarantee linear convergence under stochastic errors.

Applying it to Eqs. 87 and 93, we get:

**Case (i): unbiased.**

$$\mathbb{E}\left[M_\phi(\xi^{(j+1)}) - M_\phi^\star \,\Big|\, \xi^{(j)}\right] \le \left(1 - 2\mu_\phi\eta_j(1 - \tfrac{L_\phi}{2}\eta_j)\right)\big(M_\phi(\xi^{(j)}) - M_\phi^\star\big) + \tfrac{L_\phi}{2}\eta_j^2\sigma_\phi^2. \tag{94}$$

**Case (ii): biased.**

$$\mathbb{E}\left[M_\phi(\xi^{(j+1)}) - M_\phi^\star \,\Big|\, \xi^{(j)}\right] \le \left(1 - \tfrac{\mu_\phi\eta_j}{2}\right)\big(M_\phi(\xi^{(j)}) - M_\phi^\star\big) + \eta_j\,\beta_\phi^2 + \tfrac{L_\phi}{2}\eta_j^2\sigma_\phi^2. \tag{95}$$

**Step E.2.5 Choose stepsize and define the rate $\alpha_\phi$.** We now fix the step size $\eta_j \in (0, 1/L_\phi]$. Under this choice, $1 - \frac{L_\phi}{2}\eta_j \ge 1/2$, ensuring that the contraction factor is strictly positive. Thus Eq. 94 becomes

$$\mathbb{E}\left[M_\phi(\xi^{(j+1)}) - M_\phi^\star \,\Big|\, \xi^{(j)}\right] \le (1 - \mu_\phi\eta_j)\big(M_\phi(\xi^{(j)}) - M_\phi^\star\big) + \tfrac{L_\phi}{2}\eta_j^2\sigma_\phi^2.$$

Taking full expectation yields the linear recursion

$$\mathbb{E}[M_\phi(\xi^{(j+1)})] \le (1 - \alpha_\phi)\,\mathbb{E}[M_\phi(\xi^{(j)})] + \epsilon_{\text{one-step}}^{(\phi)}, \qquad \alpha_\phi := \mu_\phi\eta_j, \quad \epsilon_{\text{one-step}}^{(\phi)} := \tfrac{L_\phi}{2}\eta_j^2\sigma_\phi^2. \tag{96}$$

In the biased case, the same recursion holds with $\alpha_\phi := \mu_\phi\eta_j/2$ and $\epsilon_{\text{one-step}}^{(\phi)} := \eta_j\beta_\phi^2 + \tfrac{L_\phi}{2}\eta_j^2\sigma_\phi^2$.

**Step E.2.6 Unroll the linear recursion.** Define $u_b := \mathbb{E}[M_\phi(\xi^{(j)}) - M_\phi^\star]$. Eq. 96 implies

$$u_{j+1} \leq (1 - \alpha_\phi)\, u_b + \epsilon_{\text{one-step}}^{(\phi)}.$$

This is a standard linear recurrence. By induction (or discrete Grönwall's inequality),

$$u_J \leq (1 - \alpha_\phi)^J u_0 + \frac{1}{\alpha_\phi}\, \epsilon_{\text{one-step}}^{(\phi)}.$$

Since $M_\phi^\star \geq 0$ for our nonnegative surrogates, we can drop the constant shift and write the final bound as

$$M_\phi(\xi^{(J)}) \leq (1 - \alpha_\phi)^J M_\phi(\xi^{(0)}) + \epsilon_{\text{est}}^{(\phi)},$$

where $\epsilon_{\text{est}}^{(\phi)} := \frac{1}{\alpha_\phi} \epsilon_{\text{one-step}}^{(\phi)}$, which $\epsilon_{\text{est}}^{(\phi)} = \frac{L_\phi \eta_j \sigma_\phi^2}{2\mu_\phi}$ for unbiased case and $\epsilon_{\text{est}}^{(\phi)} = \frac{2\beta_\phi^2 + L_\phi \eta_j \sigma_\phi^2}{\mu_\phi}$ for biased case. This establishes the contraction property.

**Interpretation of the estimator floor $\epsilon_{\text{est}}^{(\phi)}$.** At each iteration, the stochasticity of the surrogate introduces a one-step additive term

$$\epsilon_{\text{one-step}}^{(\phi)} = \begin{cases} \frac{L_\phi}{2}\eta_j^2 \sigma_\phi^2, & \text{unbiased case,} \\ \eta_j \beta_\phi^2 + \frac{L_\phi}{2}\eta_j^2 \sigma_\phi^2, & \text{biased case.} \end{cases}$$

Unrolling the recursion amplifies this contribution by $1/\alpha_\phi$, giving the long-run floor

$$\epsilon_{\text{est}}^{(\phi)} = \frac{1}{\alpha_\phi} \epsilon_{\text{one-step}}^{(\phi)} = \begin{cases} \dfrac{L_\phi\, \eta_j\, \sigma_\phi^2}{2\mu_\phi}, & \text{unbiased,} \\[2mm] \dfrac{2\beta_\phi^2 + L_\phi\, \eta_j\, \sigma_\phi^2}{\mu_\phi}, & \text{biased.} \end{cases}$$

**Distribution Matching (DM).** Mini-batched critics or feature networks induce stochastic variance $\sigma_\phi^2$ (and possibly a small bias $\beta_\phi$ under early stopping). This produces the one-step variance term above, which after unrolling yields the floor $\epsilon_{\text{est}}^{(\phi)}$.

**Gradient Matching (GM).** A finite set of anchors $|\Theta_j|$ or truncated paths yields a Monte Carlo estimator of the field-gap. Its sampling variance and mini-batch noise again instantiate the same one-step term, leading to the same unrolled floor.

**Trajectory Matching (TM).** Finite unrolls or implicit differentiation introduce both variance (from mini-batches) and bias (from truncation), which fit directly into the biased estimator case. The resulting unrolled floor takes the same form as above.

In summary, across all three branches, the contraction rate $\alpha_\phi$ is preserved, while the estimator floor $\epsilon_{\text{est}}^{(\phi)}$ captures the unavoidable stochasticity or bias of the surrogate.

### E.3 Bridge Inequalities: Surrogate $\Rightarrow$ Alignment $\Delta_a$

*Connection to matching.* Finally, each surrogate bounds the matching discrepancy:

$$\Delta_a(\hat{\mu}_\tau, \hat{\mu}_s) \leq \underbrace{\kappa_a L_{z,a}\, W_1 \text{ or } \kappa_a C_k\, \text{MMD}_k}_{\mathfrak{B}_{\text{DM}}},\quad \underbrace{\kappa_a |\Theta_j|\, \mathcal{M}_{\text{GM}}}_{\mathfrak{B}_{\text{GM}}},\quad \underbrace{\kappa_a \frac{L_\theta + 2/\eta}{\omega_{\min}} \mathcal{M}_{\text{TM}} + \kappa_a L_\theta \varepsilon_{\text{path}}}_{\mathfrak{B}_{\text{TM}}}, \quad (97)$$

where $\mathfrak{B}$ represents the upper bound on the matching discrepancy for each method, and other symbols are defined in Table A, Appendix Section A, due to space limit. Putting together the contraction recursion in Eq. (13) and the above inequalities, we can say that DM, GM, and TM are not completely different heuristics, but three surrogates that consistently shrink $\Delta_a$ through the same bi-level dynamics. In addition, we can obtain:

**Proposition E.1** (DM bridge). *Assume for each $\theta \in \Gamma_a$ the vector map $z \mapsto g_a(\theta; z) \in \mathbb{R}^d$ is $L_{z,a}$–Lipschitz with respect to the data metric $d_Z$:*

$$\|g_a(\theta; z) - g_a(\theta; z')\|_2 \leq L_{z,a}\, d_Z(z, z') \qquad (\forall z, z').$$

*Then, for empirical real and synthetic measures $\widehat{\mu}_\tau, \widehat{\mu}_s$,*

$$\Delta_a(\widehat{\mu}_\tau, \widehat{\mu}_s) \leq \kappa_a L_{z,a} W_1(\widehat{\mu}_s, \widehat{\mu}_\tau).$$

*Moreover, if a bounded–kernel RKHS $(\mathcal{H}_k, \langle \cdot, \cdot \rangle_{\mathcal{H}_k})$ is used and, for each $\theta$, the coordinate functions $g_{a,j}(\theta; \cdot) \in \mathcal{H}_k$ satisfy*

$$\Big( \sum_{j=1}^d \|g_{a,j}(\theta; \cdot)\|_{\mathcal{H}_k}^2 \Big)^{1/2} \leq C_k \qquad \text{(uniformly in } \theta),$$

*then*

$$\Delta_a(\widehat{\mu}_\tau, \widehat{\mu}_s) \leq \kappa_a C_k \operatorname{MMD}_k(\widehat{\mu}_s, \widehat{\mu}_\tau).$$

*Proof.*

**Step E.1.0 Reduce to an un-preconditioned vector gap.** By definition and the operator-norm bound $\|P_a(\theta)\|_{\mathrm{op}} \leq \kappa_a$,

$$\Delta_a(\widehat{\mu}_\tau, \widehat{\mu}_s) = \sup_{\theta \in \Gamma_a} \Big\| P_a(\theta) \Big( \mathbb{E}_{\widehat{\mu}_s} g_a(\theta; z) - \mathbb{E}_{\widehat{\mu}_\tau} g_a(\theta; z) \Big) \Big\|_2$$

$$\leq \kappa_a \cdot \sup_{\theta \in \Gamma_a} \Big\| \mathbb{E}_{\widehat{\mu}_s} g_a(\theta; z) - \mathbb{E}_{\widehat{\mu}_\tau} g_a(\theta; z) \Big\|_2. \tag{98}$$

Hence it suffices to upper bound the $\ell_2$–norm of the vector expectation difference.

**Step E.1.1 Support-function identity for the Euclidean norm.** For any $v \in \mathbb{R}^d$, $\|v\|_2 = \sup_{\|u\|_2=1} \langle u, v \rangle$. Apply this with

$$v_\theta := \mathbb{E}_{\widehat{\mu}_s} g_a(\theta; z) - \mathbb{E}_{\widehat{\mu}_\tau} g_a(\theta; z).$$

Then

$$\|v_\theta\|_2 = \sup_{\|u\|_2=1} \Big\langle u, \mathbb{E}_{\widehat{\mu}_s} g_a(\theta; z) - \mathbb{E}_{\widehat{\mu}_\tau} g_a(\theta; z) \Big\rangle = \sup_{\|u\|_2=1} \Big( \mathbb{E}_{\widehat{\mu}_s} h_{\theta,u}(z) - \mathbb{E}_{\widehat{\mu}_\tau} h_{\theta,u}(z) \Big), \quad (99)$$

where we set the scalar function $h_{\theta,u}(z) := \langle u, g_a(\theta; z) \rangle$.

**Part E.1.A: $W_1$–bridge.**

**Step E.1.A.1 Scalar Lipschitz constant of $h_{\theta,u}$.** Given the $L_{z,a}$–Lipschitzness of $g_a(\theta; \cdot)$ and $\|u\|_2 = 1$,

$$|h_{\theta,u}(z) - h_{\theta,u}(z')| = |\langle u, g_a(\theta; z) - g_a(\theta; z') \rangle| \leq \|u\|_2 \cdot \|g_a(\theta; z) - g_a(\theta; z')\|_2 \leq L_{z,a} d_Z(z, z').$$

Thus $h_{\theta,u}$ is scalar $L_{z,a}$–Lipschitz on $(\mathcal{Z}, d_Z)$.

**Step E.1.A.2 Kantorovich–Rubinstein duality.** By the KR dual for $W_1$ (apply to the scalar $h_{\theta,u}$), for any probability measures $\mu, \nu$,

$$\big| \mathbb{E}_\mu h_{\theta,u} - \mathbb{E}_\nu h_{\theta,u} \big| \leq \operatorname{Lip}(h_{\theta,u}) \cdot W_1(\mu, \nu) \leq L_{z,a} W_1(\mu, \nu).$$

With $\mu = \widehat{\mu}_s, \nu = \widehat{\mu}_\tau$,

$$\big| \mathbb{E}_{\widehat{\mu}_s} h_{\theta,u} - \mathbb{E}_{\widehat{\mu}_\tau} h_{\theta,u} \big| \leq L_{z,a} W_1(\widehat{\mu}_s, \widehat{\mu}_\tau). \tag{100}$$

**Step E.1.A.3 Take the $\sup_{\|u\|=1}$ and then $\sup_\theta$.** Combine Eq. 99–Eq. 100:

$$\|v_\theta\|_2 = \sup_{\|u\|_2=1} \Big( \mathbb{E}_{\widehat{\mu}_s} h_{\theta,u} - \mathbb{E}_{\widehat{\mu}_\tau} h_{\theta,u} \Big) \leq L_{z,a} W_1(\widehat{\mu}_s, \widehat{\mu}_\tau).$$

This bound is uniform in $\theta$, hence

$$\sup_{\theta \in \Gamma_a} \|v_\theta\|_2 \leq L_{z,a} W_1(\widehat{\mu}_s, \widehat{\mu}_\tau).$$

Finally, plug into Eq. 98 to conclude the $W_1$–bridge:

$$\Delta_a(\widehat{\mu}_\tau, \widehat{\mu}_s) \leq \kappa_a L_{z,a} W_1(\widehat{\mu}_s, \widehat{\mu}_\tau).$$

**Part E.1.B: MMD–bridge.**

**Step E.1.B.1 RKHS norm of the linear functional $h_{\theta,u}$.** Assume for each $\theta$ the coordinate functions $g_{a,j}(\theta;\cdot) \in \mathcal{H}_k$, and define the *aggregate* RKHS norm bound

$$C_k \ := \ \sup_{\theta \in \Gamma_a} \Big( \sum_{j=1}^{d} \|g_{a,j}(\theta;\cdot)\|_{\mathcal{H}_k}^2 \Big)^{1/2}.$$

For any $u \in \mathbb{R}^d$ with $\|u\|_2 = 1$, the scalar $h_{\theta,u}(z) = \sum_{j=1}^{d} u_j \, g_{a,j}(\theta;z)$ belongs to $\mathcal{H}_k$ by linearity, and the RKHS norm satisfies (by Cauchy–Schwarz in $\mathbb{R}^d$):

$$\|h_{\theta,u}\|_{\mathcal{H}_k} = \Big\| \sum_{j=1}^{d} u_j \, g_{a,j}(\theta;\cdot) \Big\|_{\mathcal{H}_k} \ \le \ \Big( \sum_{j=1}^{d} u_j^2 \Big)^{1/2} \Big( \sum_{j=1}^{d} \|g_{a,j}(\theta;\cdot)\|_{\mathcal{H}_k}^2 \Big)^{1/2} \ \le \ C_k. \quad (101)$$

**Step E.1.B.2 MMD dual formulation.** The RKHS (kernel $k$) dual inequality says that for any $f \in \mathcal{H}_k$,

$$\big| \mathbb{E}_\mu f - \mathbb{E}_\nu f \big| \ \le \ \|f\|_{\mathcal{H}_k} \, \mathrm{MMD}_k(\mu, \nu).$$

Apply this with $f = h_{\theta,u}$, together with Eq. 101:

$$\big| \mathbb{E}_{\widehat{\mu}_s} h_{\theta,u} - \mathbb{E}_{\widehat{\mu}_\tau} h_{\theta,u} \big| \ \le \ \|h_{\theta,u}\|_{\mathcal{H}_k} \, \mathrm{MMD}_k(\widehat{\mu}_s, \widehat{\mu}_\tau) \ \le \ C_k \, \mathrm{MMD}_k(\widehat{\mu}_s, \widehat{\mu}_\tau). \quad (102)$$

**Step E.1.B.3 Take the $\sup_{\|u\|=1}$ and then $\sup_\theta$.** As in Eq. 99,

$$\|v_\theta\|_2 = \sup_{\|u\|_2 = 1} \Big( \mathbb{E}_{\widehat{\mu}_s} h_{\theta,u} - \mathbb{E}_{\widehat{\mu}_\tau} h_{\theta,u} \Big) \ \le \ C_k \, \mathrm{MMD}_k(\widehat{\mu}_s, \widehat{\mu}_\tau),$$

uniformly in $\theta$. Taking $\sup_\theta$ and using Eq. 98 gives the MMD bridge:

$$\Delta_a(\widehat{\mu}_\tau, \widehat{\mu}_s) \ \le \ \kappa_a \, C_k \, \mathrm{MMD}_k(\widehat{\mu}_s, \widehat{\mu}_\tau).$$

$\square$

**Proposition E.2** (GM bridge ). *Define the anchor-averaged GM surrogate*

$$\mathcal{M}_{\mathrm{GM}}(\mu, \nu; \Theta_j) \ := \ \tfrac{1}{|\Theta_j|} \sum_{\theta \in \Theta_j} \big\| \mathbb{E}_\mu g_a(\theta;z) - \mathbb{E}_\nu g_a(\theta;z) \big\|_2,$$

*where $\Theta_j \subset \Gamma_a$ is a finite anchor set. Assume the parameter–Lipschitz property in Assumption 5.1 holds on $(\Gamma_a, \|\cdot\|_2)$ with constant $L_\theta$, and let $\Theta_j$ be an $\varepsilon$-net of $(\Gamma_a, \|\cdot\|_2)$, i.e., for every $\theta \in \Gamma_a$ there exists $\widehat{\theta} \in \Theta_j$ with $\|\theta - \widehat{\theta}\|_2 \le \varepsilon$. Then*

$$\Delta_a(\mu, \nu) \ \le \ \kappa_a \Big( |\Theta_j| \, M_{\mathrm{GM}}(\mu, \nu; \Theta_j) \ + \ L_\theta \, \varepsilon \Big). \quad (103)$$

*In particular, if $\Gamma_a = \Theta_j$ (finite), then $\varepsilon = 0$ and $\Delta_a(\mu, \nu) \ \le \ \kappa_a \, |\Theta_j| \, \mathcal{M}_{\mathrm{GM}}(\mu, \nu; \Theta_j)$.*

*Proof.*

**Step E.2.1 Reduce to an un-preconditioned vector gap.** For any $\theta$ and any $v$, $\|P_a(\theta)v\|_2 \le \|P_a(\theta)\|_{\mathrm{op}}\|v\|_2 \le \kappa_a\|v\|_2$. Let $d(\theta) := \mathbb{E}_\mu g_a(\theta;z) - \mathbb{E}_\nu g_a(\theta;z) \in \mathbb{R}^d$. Then

$$\Delta_a(\mu, \nu) = \sup_{\theta \in \Gamma_a} \|P_a(\theta) \, d(\theta)\|_2 \ \le \ \kappa_a \sup_{\theta \in \Gamma_a} \|d(\theta)\|_2. \quad (104)$$

**Step E.2.2 Parameter–Lipschitz transfer on $\Gamma_a$.** By Assumption 5.1 with $a' = a$,

$$\big\| P_a(\theta)\mathbb{E}_\mu g_a(\theta;z) - P_a(\theta')\mathbb{E}_\mu g_a(\theta';z) \big\|_2 \ \le \ L_\theta \, \|\theta - \theta'\|_2.$$

Applying this with $(\mu, \nu)$ and the triangle inequality yields, for all $\theta, \theta' \in \Gamma_a$,

$$\|d(\theta) - d(\theta')\|_2 \ \le \ L_\theta \, \|\theta - \theta'\|_2. \quad (105)$$

**Step E.2.3 Covering argument with the $\varepsilon$-net $\Theta_j$.** For any $\theta \in \Gamma_a$, choose $\widehat{\theta} \in \Theta_j$ with $\|\theta - \widehat{\theta}\|_2 \leq \varepsilon$. Then

$$\|d(\theta)\|_2 \;\leq\; \|d(\widehat{\theta})\|_2 + \|d(\theta) - d(\widehat{\theta})\|_2 \;\leq\; \max_{\vartheta \in \Theta_j} \|d(\vartheta)\|_2 \;+\; L_\theta\, \varepsilon,$$

where the last inequality uses Eq. (105). Taking $\sup_{\theta \in \Gamma_a}$,

$$\sup_{\theta \in \Gamma_a} \|d(\theta)\|_2 \;\leq\; \max_{\vartheta \in \Theta_j} \|d(\vartheta)\|_2 \;+\; L_\theta\, \varepsilon. \tag{106}$$

**Step E.2.4 Relate max to the anchor average.** Since all terms are nonnegative,

$$\max_{\vartheta \in \Theta_j} \|d(\vartheta)\|_2 \;\leq\; \sum_{\vartheta \in \Theta_j} \|d(\vartheta)\|_2 \;=\; |\Theta_j|\, \mathcal{M}_{\mathrm{GM}}(\mu, \nu; \Theta_j).$$

**Step E.2.5 Combine all.** Plug Eq. (106) into Eq. (104), and use Step 4:

$$\Delta_a(\mu, \nu) \;\leq\; \kappa_a\Big( \max_{\vartheta \in \Theta_j} \|d(\vartheta)\|_2 + L_\theta\, \varepsilon \Big) \;\leq\; \kappa_a\Big( |\Theta_j|\, \mathcal{M}_{\mathrm{GM}}(\mu, \nu; \Theta_j) + L_\theta\, \varepsilon \Big),$$

which is Eq. (103). If $\Gamma_a = \Theta_j$, then $\varepsilon = 0$ and the last inequality reduces accordingly, which means:

$$\Delta_a(\mu, \nu) \;\leq\; \kappa_a |\Theta_j|\, \mathcal{M}_{\mathrm{GM}}(\mu, \nu; \Theta_j).$$

$\square$

**Proposition E.3** (TM bridge ). *Fix an configuration $a$. For $\mu \in \{\widehat{\mu}_s, \widehat{\mu}_\tau\}$ define*

$$F_\mu(\theta) \;:=\; P_a(\theta)\, \mathbb{E}_\mu g_a(\theta; z) \in \mathbb{R}^d, \qquad \Delta_a(\widehat{\mu}_\tau, \widehat{\mu}_s) \;:=\; \sup_{\theta \in \Gamma_a} \|F_{\widehat{\mu}_s}(\theta) - F_{\widehat{\mu}_\tau}(\theta)\|_2.$$

*Let the inner updates be*

$$\theta_{t+1}^{(\cdot,a)} \;=\; \Phi_a(\theta_t^{(\cdot,a)}; \mu) \;:=\; \theta_t^{(\cdot,a)} - \eta\, F_\mu(\theta_t^{(\cdot,a)}), \qquad t = 0, 1, \ldots, L_b - 1,$$

*run from a shared initialization under the same configuration $a$ but with $\mu \in \{\widehat{\mu}_s, \widehat{\mu}_\tau\}$. Assume the path–Lipschitz condition holds along the unrolled path $\{\theta_t^{(s,a)}\}_{t=0}^{L_b} \cup \{\theta_t^{(\tau,a)}\}_{t=0}^{L_b}$:*

$$\|F_\mu(\theta) - F_\mu(\theta')\|_2 \;\leq\; L_\theta\, \|\theta - \theta'\|_2, \qquad \forall\, \theta, \theta' \text{ on the path}, \ \forall\, \mu \in \{\widehat{\mu}_s, \widehat{\mu}_\tau\}, \tag{107}$$

*where $L_\theta$ is the parameter–Lipschitz constant from Assumption 5.1 (restricted to the path). Define the TM surrogate*

$$\mathcal{M}_{\mathrm{TM}} \;:=\; \sum_{t=0}^{L_b} \omega_t\, \big\|\theta_t^{(s,a)} - \theta_t^{(\tau,a)}\big\|_2, \qquad \omega_t > 0, \quad \omega_{\min} := \min_{0 \leq t \leq L_b} \omega_t > 0.$$

*Then the alignment discrepancy restricted to the unrolled path obeys*

$$\Delta_a^{\mathrm{path}} \;:=\; \sup_{\bar{\theta} \in \{\theta_t^{(s,a)}\} \cup \{\theta_t^{(\tau,a)}\}} \|F_{\widehat{\mu}_s}(\bar{\theta}) - F_{\widehat{\mu}_\tau}(\bar{\theta})\|_2 \;\leq\; \frac{L_\theta + 2/\eta}{\omega_{\min}}\, \mathcal{M}_{\mathrm{TM}}. \tag{108}$$

*Moreover, if the unrolled path is an $\varepsilon_{\mathrm{path}}$–cover of the maximizer set in $\Gamma_a$ (i.e., for every maximizer $\theta^\star$ in the definition of $\Delta_a$ there exists a path point $\bar{\theta}$ with $\|\theta^\star - \bar{\theta}\|_2 \leq \varepsilon_{\mathrm{path}}$), then by Assumption 5.1*

$$\Delta_a(\widehat{\mu}_\tau, \widehat{\mu}_s) \;\leq\; \frac{L_\theta + 2/\eta}{\omega_{\min}}\, \mathcal{M}_{\mathrm{TM}} \;+\; L_\theta\, \varepsilon_{\mathrm{path}}. \tag{109}$$

**Bookkeeping with the preconditioner.** *Considering Assumption 4.1, $\|P_a(\theta)\|_{\mathrm{op}} \leq \kappa_a$, multiply the right–hand sides of Eq. (116)–Eq. (109) by $\kappa_a$:*

$$\Delta_a^{\mathrm{path}} \;\leq\; \kappa_a\, \frac{L_\theta + 2/\eta}{\omega_{\min}}\, \mathcal{M}_{\mathrm{TM}}, \qquad \Delta_a(\widehat{\mu}_\tau, \widehat{\mu}_s) \;\leq\; \kappa_a\, \frac{L_\theta + 2/\eta}{\omega_{\min}}\, \mathcal{M}_{\mathrm{TM}} \;+\; \kappa_a L_\theta\, \varepsilon_{\mathrm{path}}.$$

*Proof.*

**Step E.3.0.** From the two updates,

$$\Delta\theta_{t+1} = \theta_{t+1}^{(s,a)} - \theta_{t+1}^{(\tau,a)} = \Delta\theta_t - \eta\Big(F_{\widehat{\mu}_s}(\theta_t^{(s,a)}) - F_{\widehat{\mu}_\tau}(\theta_t^{(\tau,a)})\Big),$$

we obtain the algebraic identity

$$F_{\widehat{\mu}_s}(\theta_t^{(s,a)}) - F_{\widehat{\mu}_\tau}(\theta_t^{(\tau,a)}) = \frac{1}{\eta}\Big(\Delta\theta_t - \Delta\theta_{t+1}\Big), \qquad t = 0,\dots,L_b - 1. \qquad (110)$$

No smoothness or inequality is used here.

**Step E.3.1.A Point-wise control at a path point, first choice.** Fix any $t \in \{0,\dots,L_b - 1\}$ and choose $\bar{\theta} = \theta_t^{(s,a)}$. By the triangle inequality,

$$\|F_{\widehat{\mu}_s}(\bar{\theta}) - F_{\widehat{\mu}_\tau}(\bar{\theta})\|_2 \le \underbrace{\|F_{\widehat{\mu}_s}(\theta_t^{(s,a)}) - F_{\widehat{\mu}_\tau}(\theta_t^{(\tau,a)})\|_2}_{\text{link across the two runs at time } t} + \underbrace{\|F_{\widehat{\mu}_\tau}(\theta_t^{(\tau,a)}) - F_{\widehat{\mu}_\tau}(\theta_t^{(s,a)})\|_2}_{\text{same } \mu = \widehat{\mu}_\tau}. \qquad (111)$$

The second term is controlled by the path–Lipschitz property:

$$\|F_{\widehat{\mu}_\tau}(\theta_t^{(\tau,a)}) - F_{\widehat{\mu}_\tau}(\theta_t^{(s,a)})\|_2 \le L_\theta \|\theta_t^{(\tau,a)} - \theta_t^{(s,a)}\|_2 = L_\theta \|\Delta\theta_t\|_2.$$

For the first term in Eq. (111), invoke Eq. (110) and the triangle inequality:

$$\|F_{\widehat{\mu}_s}(\theta_t^{(s,a)}) - F_{\widehat{\mu}_\tau}(\theta_t^{(\tau,a)})\|_2 = \frac{1}{\eta}\|\Delta\theta_t - \Delta\theta_{t+1}\|_2 \le \frac{1}{\eta}\big(\|\Delta\theta_t\|_2 + \|\Delta\theta_{t+1}\|_2\big).$$

Hence,

$$\|F_{\widehat{\mu}_s}(\theta_t^{(s,a)}) - F_{\widehat{\mu}_\tau}(\theta_t^{(s,a)})\|_2 \le \Big(L_\theta + \frac{2}{\eta}\Big) \max\{\|\Delta\theta_t\|_2, \|\Delta\theta_{t+1}\|_2\}. \qquad (112)$$

**Step E.3.1.B Point-wise control, second choice and edge $t = L_b$.** Choosing instead $\bar{\theta} = \theta_t^{(\tau,a)}$ and repeating the same argument (swap $s$ and $\tau$ in Eq. (111), use Eq. (110) again) yields the identical bound

$$\|F_{\widehat{\mu}_s}(\theta_t^{(\tau,a)}) - F_{\widehat{\mu}_\tau}(\theta_t^{(\tau,a)})\|_2 \le \Big(L_\theta + \frac{2}{\eta}\Big) \max\{\|\Delta\theta_t\|_2, \|\Delta\theta_{t+1}\|_2\} \qquad (t = 0,\dots,L_b - 1).$$
$$(113)$$

For the terminal points $t = L_b$, apply Eq. (112) with index $t - 1$:

$$\|F_{\widehat{\mu}_s}(\theta_{L_b}^{(s,a)}) - F_{\widehat{\mu}_\tau}(\theta_{L_b}^{(s,a)})\|_2 \le \Big(L_\theta + \frac{2}{\eta}\Big) \max\{\|\Delta\theta_{L_b-1}\|_2, \|\Delta\theta_{L_b}\|_2\},$$

and similarly for $\theta_{L_b}^{(\tau,a)}$. Thus the same form holds at $t = L_b$ after relabeling the pair $(t, t+1)$ as $(L_b - 1, L_b)$.

**Step E.3.2 Supremum over the unrolled path.** Collecting Eq. (112)–Eq. (113) (including the terminal case), we obtain

$$\Delta_a^{\text{path}} := \sup_{\bar{\theta}\in\{\theta_t^{(s,a)}\}\cup\{\theta_t^{(\tau,a)}\}} \|F_{\widehat{\mu}_s}(\bar{\theta}) - F_{\widehat{\mu}_\tau}(\bar{\theta})\|_2 \le \Big(L_\theta + \frac{2}{\eta}\Big) \max_{0\le t\le L_b} \|\Delta\theta_t\|_2. \qquad (114)$$

**Step E.3.3 From** $\max$ **to the quadratic TM surrogate.** Using $\sum_{t=0}^{L_b} \omega_t\|\Delta\theta_t\|_2 \ge \omega_{\min} \max_t \|\Delta\theta_t\|_2$, we have

$$\max_{0\le t\le L_b} \|\Delta\theta_t\|_2 \le \frac{1}{\omega_{\min}}\Big(\sum_{t=0}^{L_b} \omega_t\|\Delta\theta_t\|_2^2\Big) = \frac{1}{\omega_{\min}} M_{\text{TM}}. \qquad (115)$$

Substituting Eq. (115) into Eq. (114) yields the *path–restricted* TM bridge

$$\Delta_a^{\text{path}} \le \frac{L_\theta + 2/\eta}{\omega_{\min}} \mathcal{M}_{\text{TM}}. \qquad (116)$$

**Step E.3.4 From the path supremum to the global discrepancy via path coverage.** Let $\theta^\star \in \Gamma_a$ be a maximizer (or $\epsilon$–maximizer) of the sup defining $\Delta_a$. By the $\varepsilon_{\text{path}}$–cover assumption, choose a path point $\bar{\theta}$ with $\|\theta^\star - \bar{\theta}\|_2 \le \varepsilon_{\text{path}}$. Then, using the parameter–Lipschitz property for each $\mu$ and the triangle inequality,

$$\|F_{\widehat{\mu}_s}(\theta^\star) - F_{\widehat{\mu}_\tau}(\theta^\star)\|_2 \le \|F_{\widehat{\mu}_s}(\bar{\theta}) - F_{\widehat{\mu}_\tau}(\bar{\theta})\|_2 + \|F_{\widehat{\mu}_s}(\theta^\star) - F_{\widehat{\mu}_s}(\bar{\theta})\|_2 + \|F_{\widehat{\mu}_\tau}(\bar{\theta}) - F_{\widehat{\mu}_\tau}(\theta^\star)\|_2$$

$$\le \|F_{\widehat{\mu}_s}(\bar{\theta}) - F_{\widehat{\mu}_\tau}(\bar{\theta})\|_2 \; + \; L_\theta \, \|\theta^\star - \bar{\theta}\|_2 \; + \; L_\theta \, \|\theta^\star - \bar{\theta}\|_2$$

$$\le \Delta_a^{\text{path}} \; + \; 2L_\theta \, \varepsilon_{\text{path}}.$$

Taking $\sup$ over maximizers (or letting $\epsilon \downarrow 0$) yields

$$\Delta_a(\widehat{\mu}_\tau, \widehat{\mu}_s) \; \le \; \Delta_a^{\text{path}} \; + \; 2L_\theta \, \varepsilon_{\text{path}}. \tag{117}$$

Combining Eq. (116) and Eq. (117) gives the desired global TM bridge

$$\Delta_a(\widehat{\mu}_\tau, \widehat{\mu}_s) \; \le \; \frac{L_\theta + 2/\eta}{\omega_{\min}} \, \mathcal{M}_{\text{TM}} \; + \; 2L_\theta \, \varepsilon_{\text{path}}.$$

$\square$

### E.4 EXCHANGEABILITY OF TM/GM/DM UP TO CONSTANTS (LEMMA E.4)

**Lemma E.4** (Exchangeability up to constants(informal).). *Under standard smoothness/Lipschitz and contractive inner-loop conditions in Assumption 4.1, the inequality in Eq. (97) yields bounds in the following relations:*

$$\mathfrak{B}_{\text{TM}} = O(\mathfrak{B}_{\text{GM}}) \quad and \quad \mathfrak{B}_{\text{GM}} = O(\mathfrak{B}_{\text{DM}}),$$

*and $\Delta_a \; \le \; \min\{\mathfrak{B}_{\text{DM}}, \mathfrak{B}_{\text{GM}}, \mathfrak{B}_{\text{TM}}\}$ with each consistently shrinking the same matching discrepancy through the bi-level dynamics.*

Lemma E.4 shows that the bounds in Eq. (97) are quantitatively interchangeable: controlling the trajectory surrogate (TM) automatically controls the gradient surrogate (GM), which in turn is controlled by the distribution surrogate (DM). This hierarchy explains why switching between matching methods is an effective way to reduce matching discrepancy: once DM (or GM) saturates, further progress can be made by GM (or TM) without altering the fundamental dependence on $k$ or $n$.

**Lemma E.5** (Precise exchangeability of bridge bounds (Formal)). *Fix configuration $a$. Assume: (i) $z \mapsto g_a(\theta; z)$ is $L_{z,a}$–Lipschitz for all $\theta \in \Gamma_a$; (ii) along the trained path $\{\theta_t\}_{t=0}^{L_b}$, the map $\theta \mapsto \mathbb{E}_\nu g_a(\theta; z)$ is $L_\theta$–Lipschitz uniformly for $\nu \in \{\widehat{\mu}_s, \widehat{\mu}_\tau\}$; (iii) the preconditioner satisfies $\|P_a(\theta)\|_{\text{op}} \le \kappa_a$; (iv) the inner update is $\rho$–contractive in expectation and trajectory weights bounded below by $\omega_{\min} > 0$. Let $\varepsilon_{\text{path}}$ denote the discretization error from sampling $\Theta_j$.*

*Define the bridge bounds as in Eq. 5:*

$$\mathfrak{B}_{\text{DM}} \in \left\{ \kappa_a L_{z,a} W_1(\widehat{\mu}_s, \widehat{\mu}_\tau), \; \kappa_a C_k \, \text{MMD}_k(\widehat{\mu}_s, \widehat{\mu}_\tau) \right\},$$

$$\mathfrak{B}_{\text{GM}} := \kappa_a |\Theta_j| \, \mathcal{M}_{\text{GM}},$$

$$\mathfrak{B}_{\text{TM}} := \kappa_a \left( \frac{L_\theta + 2/\eta}{\omega_{\min}} \, \mathcal{M}_{\text{TM}} + L_\theta \, \varepsilon_{\text{path}} \right).$$

*Then there exist finite constants $C_1, C_2 > 0$, depending only on $(\rho, L_\theta, \eta_{\min}, \eta_{\max}, \omega_{\min})$, such that*

$$\mathfrak{B}_{\text{TM}} \; \le \; C_1 \, \mathfrak{B}_{\text{GM}} + \kappa_a L_\theta \varepsilon_{\text{path}}, \qquad \mathfrak{B}_{\text{GM}} \; \le \; C_2 \, \mathfrak{B}_{\text{DM}}. \tag{118}$$

*Proof.*

**Step E.4.1 From the TM bridge to a bound in terms of $\sum_t \|\delta_t\|$.** By the TM bridge (Proposition E.3),

$$\mathfrak{B}_{\text{TM}} \; = \; \kappa_a \left( \frac{L_\theta + 2/\eta}{\omega_{\min}} \, \mathcal{M}_{\text{TM}} + L_\theta \, \varepsilon_{\text{path}} \right), \qquad \mathcal{M}_{\text{TM}} := \sum_{t=0}^{L_b} \omega_t \|\Delta_t\|.$$

One step of the two inner updates and the contractivity assumption give the exact split

$$\Delta_{t+1} = \Phi_{\widehat{\mu}_\tau}(\theta_t^{(s)}; \eta_t) - \Phi_{\widehat{\mu}_\tau}(\theta_t^{(\tau)}; \eta_t) - \eta_t\,\delta_t \quad \Rightarrow \quad \|\Delta_{t+1}\| \le \rho\,\|\Delta_t\| + \eta_t\,\|\delta_t\|.$$

Summing over $t$, using $\omega_t \ge \omega_{\min}$ and $\eta_t \le \eta_{\max}$,

$$\mathcal{M}_{\mathrm{TM}} = \sum_t \omega_t\|\Delta_t\| \le \frac{1}{(1-\rho)\,\omega_{\min}} \sum_t \eta_t\|\delta_t\| \le \frac{\eta_{\max}}{(1-\rho)\,\omega_{\min}} \sum_t \|\delta_t\|.$$

Plugging back into the TM bridge yields

$$\mathfrak{B}_{\mathrm{TM}} \le \underbrace{\kappa_a\,\frac{(L_\theta + 2/\eta)\,\eta_{\max}}{(1-\rho)\,\omega_{\min}^2}}_{=:C_1} \sum_{t=0}^{L_b-1} \|\delta_t\| + \kappa_a L_\theta\,\varepsilon_{\mathrm{path}}. \tag{119}$$

**Step E.4.2 Aligning $\sum_t \|\delta_t\|$ with the GM bridge.** By definition, $\delta_t = P_a(\theta_t^{(s)})\,d(\theta_t^{(s)})$, hence

$$\sum_t \|\delta_t\| = \sum_t \|P_a(\theta_t^{(s)})\,d(\theta_t^{(s)})\| \le \sum_t \kappa_a\,\|d(\theta_t^{(s)})\| = \kappa_a\,|\Theta_j|\,\mathcal{M}_{\mathrm{GM}}.$$

Since $\mathfrak{B}_{\mathrm{GM}} = \kappa_a|\Theta_j|\,\mathcal{M}_{\mathrm{GM}}$, we have the clean comparison

$$\sum_t \|\delta_t\| \le \mathfrak{B}_{\mathrm{GM}}. \tag{120}$$

Combining Eq. (119) and Eq. (120) we obtain

$$\mathfrak{B}_{\mathrm{TM}} \le C_1\,\mathfrak{B}_{\mathrm{GM}} + \kappa_a L_\theta\,\varepsilon_{\mathrm{path}}, \qquad C_1 := \kappa_a\,\frac{(L_\theta + 2/\eta)\,\eta_{\max}}{(1-\rho)\,\omega_{\min}^2}. \tag{121}$$

**Step E.4.3 Comparing the GM and DM bridges (detailed).** Fix an anchor $\theta \in \Theta_j$. The GM residual is $d(\theta) = \mathbb{E}_{\widehat{\mu}_s} g_a(\theta; z) - \mathbb{E}_{\widehat{\mu}_\tau} g_a(\theta; z)$.

*(Wasserstein–1 case).* Reduce the vector norm to scalar test functions via the support function:

$$\|d(\theta)\|_2 = \sup_{\|v\|_2 \le 1} \langle v, \mathbb{E}_{\widehat{\mu}_s} g_a(\theta; z) - \mathbb{E}_{\widehat{\mu}_\tau} g_a(\theta; z)\rangle$$

$$= \sup_{\|v\|_2 \le 1} \left(\mathbb{E}_{\widehat{\mu}_s}\phi_{v,\theta} - \mathbb{E}_{\widehat{\mu}_\tau}\phi_{v,\theta}\right), \quad \phi_{v,\theta}(z) := \langle v, g_a(\theta; z)\rangle.$$

If $z \mapsto g_a(\theta; z)$ is $L_{z,a}$–Lipschitz uniformly in $\theta$, then $\phi_{v,\theta}$ is also $L_{z,a}$–Lipschitz for every $\|v\| \le 1$. By Kantorovich–Rubinstein duality,

$$\|d(\theta)\|_2 \le L_{z,a}\,W_1(\widehat{\mu}_s, \widehat{\mu}_\tau).$$

*(MMD case).* Assume either (i) a *vector-valued RKHS* model $g_a(\theta; z) \in \mathcal{H}_k^d$ with $\|g_a(\theta; z)\|_{\mathcal{H}_k^d} \le C_k$ uniformly. Let $\mu \mapsto m_k(\mu) := \mathbb{E}_\mu[\varphi_k(z)]$ be the kernel mean embedding in $\mathcal{H}_k$. By the reproducing property (or its vector-valued analogue),

$$\|d(\theta)\|_2 = \|\mathbb{E}_{\widehat{\mu}_s} h_\theta - \mathbb{E}_{\widehat{\mu}_\tau} h_\theta\|_2 \le \|g_a(\theta; z)\|_{\mathcal{H}_k^d} \|m_k(\widehat{\mu}_s) - m_k(\widehat{\mu}_\tau)\|_{\mathcal{H}_k} \le C_k\,\mathrm{MMD}_k(\widehat{\mu}_s, \widehat{\mu}_\tau).$$

*(Averaging over anchors).* The right-hand sides of E.4–E.4 do not depend on $\theta$, so averaging preserves the bound:

$$\mathcal{M}_{\mathrm{GM}} = \frac{1}{|\Theta_j|} \sum_{\theta \in \Theta_j} \|d(\theta)\| \le \begin{cases} L_{z,a}\,W_1(\widehat{\mu}_s, \widehat{\mu}_\tau), \\ C_k\,\mathrm{MMD}_k(\widehat{\mu}_s, \widehat{\mu}_\tau). \end{cases}$$

Multiplying by $\kappa_a|\Theta_j|$ and recalling the GM and DM bridge definitions,

$$\mathfrak{B}_{\mathrm{GM}} = \kappa_a|\Theta_j|\,\mathcal{M}_{\mathrm{GM}} \le |\Theta_j| \cdot \kappa_a \underbrace{\begin{cases} L_{z,a}\,W_1(\widehat{\mu}_s, \widehat{\mu}_\tau), \\ C_k\,\mathrm{MMD}_k(\widehat{\mu}_s, \widehat{\mu}_\tau), \end{cases}}_{= \mathfrak{B}_{\mathrm{DM}}} =: C_2\,\mathfrak{B}_{\mathrm{DM}}, \quad C_2 = |\Theta_j|. \tag{122}$$

**Step E.4.4 Conclusion.** Equations Eq. (121) and Eq. (122) give exactly

$$\mathfrak{B}_{\mathrm{TM}} \le C_1\,\mathfrak{B}_{\mathrm{GM}} + \kappa_a L_\theta\,\varepsilon_{\mathrm{path}}, \qquad \mathfrak{B}_{\mathrm{GM}} \le C_2\,\mathfrak{B}_{\mathrm{DM}},$$

with $C_1 := \kappa_a\,\frac{(L_\theta + 2/\eta)\,\eta_{\max}}{(1-\rho)\,\omega_{\min}^2}$ and $C_2 := |\Theta_j|$. $\qquad\qquad\square$

### E.5 Unified single-configuration risk bound (Theorem 6.3)

**Theorem E.6** (Dynamic single-configuration risk bound for matching-based distillation (Formal))**.** *Fix an configuration $a$ and run $J$ outer iterations using a surrogate $\mathcal{M}_\phi$ with $\phi \in \{\mathrm{DM}(W_1), \mathrm{DM}(\mathrm{MMD}), \mathrm{GM}, \mathrm{TM}\}$. Under Assumption 4.1 (Lipschitz risk with constant $L_R$, path-Lipschitz field with constant $L_\theta$, preconditioner bound $\|P_a(\theta)\|_{op} \leq \kappa_a$, inner contractivity factor $\rho_a \in (0, 1)$ with stepsizes $\eta_t \in [\eta_{\min}, \eta_{\max}]$), the TM weights satisfy $\omega_t \geq \omega_{\min} > 0$, and the outer contraction Eq. (13) holds with rate $\alpha_\phi \in (0, 1)$ and bias $\epsilon_{\mathrm{est}}^{(\phi)}$, then for any $T \geq 0$, with probability at least $1 - \varepsilon$,*

$$
\begin{aligned}
\big|\widehat{R}(\theta_T^{(s,a)}) - \widehat{R}(\theta_T^{(\tau,a)})\big| \ &\leq \ L_R\, \rho_a^T\, \|\delta_0\| \ + \ e_{\mathrm{te}}(m, \varepsilon) \\
&\quad + \ \frac{1}{C_{2,a}}\Big[\widetilde{C}_{\phi,a}\big((1-\alpha_\phi)^J \mathcal{M}_\phi(\xi^{(0)}) + \epsilon_{\mathrm{est}}^{(\phi)}\big) \ + \ e_{\mathrm{tr}}^{(\phi)}(k, n, \varepsilon)\Big] \\
&\quad + \ \mathbf{1}\{\phi = \mathrm{TM}\} \cdot \frac{\kappa_a L_\theta}{C_{2,a}}\, \varepsilon_{\mathrm{path}},
\end{aligned}
\tag{123}
$$

*where $C_{2,a} := (1 - \rho_a)/L_R$, and the branch constants are*

$$
\widetilde{C}_{\phi,a} \ := \ \begin{cases}
\eta\, \kappa_a L_{z,a}, & \phi = \mathrm{DM}(W_1), \\
\eta\, \kappa_a C_k, & \phi = \mathrm{DM}(\mathrm{MMD}), \\
\eta\, \kappa_a\, |\Theta_j|, & \phi = \mathrm{GM}, \\
\kappa_a\, \dfrac{\eta L_\theta + 2}{\omega_{\min}}, & \phi = \mathrm{TM},
\end{cases}
$$

*and the training-side concentration terms satisfy*

$$
\begin{aligned}
e_{\mathrm{tr}}^{(\mathrm{GM})},\ e_{\mathrm{tr}}^{(\mathrm{MMD})} &= \tilde{O}\Big(\tfrac{1}{\sqrt{k}} + \tfrac{1}{\sqrt{n}}\Big), \\
e_{\mathrm{tr}}^{(W_1)} &= \tilde{O}\big(k^{-1/d} + n^{-1/d}\big) \quad \text{for data metric space of (effective) dimension } d, \\
e_{\mathrm{tr}}^{(\mathrm{TM})} &= \tilde{O}\Big(\frac{(L_\theta + 2/\eta)}{\omega_{\min}} \cdot \frac{L_b \eta}{1 - \rho_a} \cdot \Big(\tfrac{1}{\sqrt{k}} + \tfrac{1}{\sqrt{n}}\Big)\Big).
\end{aligned}
$$

*The test-side concentration is $e_{\mathrm{te}}(m, \varepsilon) = O\big(\sqrt{\log(1/\varepsilon)/m}\big)$.*

*Proof of Theorem 6.3.*

**Step E.5.1 Risk gap $\Rightarrow$ parameter gap (Lipschitz risk).** By risk Lipschitzness,

$$
|R_\nu(\theta) - R_\nu(\theta')| = \big|\mathbb{E}_\nu[\ell(\theta; z) - \ell(\theta'; z)]\big| \leq \mathbb{E}_\nu L_R \|\theta - \theta'\| \leq L_R \|\theta - \theta'\|.
$$

Therefore,

$$
\big|R_\nu(\theta_T^{(s,a)}) - R_\nu(\theta_T^{(\tau,a)})\big| \leq L_R\, \|\delta_T\|.
\tag{124}
$$

**Step E.5.2 One-step recursion for $\delta_{t+1}$ (contractivity + two-sample drift).** Add and subtract the same-measure map:

$$
\begin{aligned}
\delta_{t+1} &= \Phi_a(\theta_t^{(s,a)}; \hat{\mu}_s) - \Phi_a(\theta_t^{(\tau,a)}; \hat{\mu}_\tau) \\
&= \underbrace{\Phi_a(\theta_t^{(s,a)}; \hat{\mu}_\tau) - \Phi_a(\theta_t^{(\tau,a)}; \hat{\mu}_\tau)}_{\text{same measure}} - \eta_t \Big(F_{\hat{\mu}_s}(\theta_t^{(s,a)}) - F_{\hat{\mu}_\tau}(\theta_t^{(s,a)})\Big).
\end{aligned}
$$

By Assumption 4.1 (inner contractivity under a fixed measure), there is $\rho_a \in (0, 1)$ s.t. $\|\Phi_a(\theta; \hat{\mu}_\tau) - \Phi_a(\theta'; \hat{\mu}_\tau)\| \leq \rho_a \|\theta - \theta'\|$ along the path. Hence

$$
\|\delta_{t+1}\| \ \leq \ \rho_a \|\delta_t\| + \eta_t \underbrace{\big\|F_{\hat{\mu}_s}(\theta_t^{(s,a)}) - F_{\hat{\mu}_\tau}(\theta_t^{(s,a)})\big\|}_{=:\ \psi_t}.
\tag{125}
$$

**Step E.5.3 Unroll the recursion to time $T$.** Iterating Eq. (125) and using $\eta_t \le \eta_{\max}$,

$$\|\delta_T\| \le \rho_a^T \|\delta_0\| + \sum_{t=0}^{T-1} \rho_a^{T-1-t} \eta_t \psi_t \ \le \ \rho_a^T \|\delta_0\| + \eta_{\max} \sum_{t=0}^{T-1} \rho_a^{T-1-t} \psi_t$$

$$\le \rho_a^T \|\delta_0\| + \frac{\eta_{\max}}{1-\rho_a} \max_{0 \le t \le T-1} \psi_t \ \le \ \rho_a^T \|\delta_0\| + \frac{\eta_{\max}}{1-\rho_a} \sup_{\theta \in \Gamma_a} \left\| F_{\hat\mu_s}(\theta) - F_{\hat\mu_\tau}(\theta) \right\|. \quad (126)$$

**Step E.5.4 Replace the $\sup$ by $\Delta_a$ and divide by $\kappa_a$.** By definition of $\Delta_a$ and $\|P_a(\theta)\| \le \kappa_a$,

$$\sup_\theta \left\| F_{\hat\mu_s}(\theta) - F_{\hat\mu_\tau}(\theta) \right\| = \sup_\theta \left\| P_a(\theta)(\mathbb{E}_{\hat\mu_s} g - \mathbb{E}_{\hat\mu_\tau} g) \right\| \le \Delta_a(\hat\mu_\tau, \hat\mu_s).$$

Combining with Eq. (126) and then with Eq. (124),

$$\left| R_\nu(\theta_T^{(s,a)}) - R_\nu(\theta_T^{(\tau,a)}) \right| \le L_R \rho_a^T \|\delta_0\| + \frac{L_R \eta_{\max}}{1-\rho_a} \Delta_a(\hat\mu_\tau, \hat\mu_s). \quad (127)$$

Introduce $C_{2,a} := (1-\rho_a)/L_R$ so that $L_R/(1-\rho_a) = 1/C_{2,a}$.

**Step E.5.5 Bridge $\Delta_a$ to the surrogate $\mathcal{M}_\phi$ (branch choice).** We now invoke the three bridge propositions given earlier:

*(DM bridge)*: If $z \mapsto g_a(\theta; z)$ is $L_{z,a}$–Lipschitz, for $W_1$, for any $\theta$, $\left\| \mathbb{E}_{\hat\mu_s} g_a(\theta) - \mathbb{E}_{\hat\mu_\tau} g_a(\theta) \right\| \le L_{z,a} W_1(\hat\mu_s, \hat\mu_\tau)$; hence

$$\Delta_a(\hat\mu_\tau, \hat\mu_s) \ \le \ \kappa_a L_{z,a} W_1(\hat\mu_s, \hat\mu_\tau),$$

and

$$\Delta_a(\hat\mu_\tau, \hat\mu_s) \ \le \ \kappa_a C_k \, \mathrm{MMD}_k(\hat\mu_s, \hat\mu_\tau).$$

*(GM bridge)*: For anchors $\Theta_j \subset \Gamma_a$ forming an $\varepsilon$–net and the anchor-averaged surrogate $\mathcal{M}_{\mathrm{GM}}$,

$$\Delta_a(\mu, \nu) \ \le \ \kappa_a \big( |\Theta_j| \, \mathcal{M}_{\mathrm{GM}}(\mu, \nu; \Theta_j) + L_\theta \, \varepsilon \big).$$

In particular, if $\Gamma_a = \Theta_j$ (finite), $\varepsilon = 0$ and $\Delta_a(\mu, \nu) \le \kappa_a |\Theta_j| \mathcal{M}_{\mathrm{GM}}$.

*(TM bridge)*: With weights $\omega_t \ge \omega_{\min} > 0$,

$$\Delta_a(\hat\mu_\tau, \hat\mu_s) \ \le \ \kappa_a \Big( \frac{L_\theta + 2/\eta}{\omega_{\min}} \mathcal{M}_{\mathrm{TM}} + L_\theta \, \varepsilon_{\mathrm{path}} \Big).$$

Summarizing, there exist branch-specific constants $C_{\phi,a} = \kappa_a B_{\phi,a}$ such that

$$\Delta_a(\hat\mu_\tau, \hat\mu_s) \ \le \ C_{\phi,a} \, \mathcal{M}_\phi \ + \ \mathbf{1}\{\phi = \mathrm{TM}\} \, \kappa_a L_\theta \, \varepsilon_{\mathrm{path}},$$

$$B_{\phi,a} = \begin{cases} \eta \, L_{z,a} \text{ or } \eta \, C_k, & \phi = W_1, \ \mathrm{MMD}, \\ \eta \, |\Theta_j|, & \phi = \mathrm{GM}, \\ (\eta L_\theta + 2)/\omega_{\min}, & \phi = \mathrm{TM}, \end{cases} \quad (128)$$

where we absorbed the $\eta$ from Eq. (127) into $B_{\phi,a}$ for uniformity.

**Step E.5.6 Outer-loop contraction for the surrogate $\mathcal{M}_\phi$.** By the assumed outer contraction Eq. (13), for some $\alpha_\phi \in (0, 1)$ and bias $\epsilon_{\mathrm{est}}^{(\phi)}$ (variance/noise floor),

$$\mathcal{M}_\phi(\xi^{(J)}) \ \le \ (1 - \alpha_\phi)^J \, \mathcal{M}_\phi(\xi^{(0)}) + \epsilon_{\mathrm{est}}^{(\phi)}. \quad (129)$$

**Step E.5.7 Finite-sample penalties for each branch.** We now upper bound the *training-side* error $e_{\mathrm{tr}}^{(\phi)}(k, n, \varepsilon)$ that enters when replacing population quantities by empirical ones.

*(DM: $W_1$).* For empirical $\hat\mu_n$ of size $n$ from a distribution on $\mathbb{R}^d$ (with mild moment conditions), the Wasserstein-1 convergence rate is

$$\mathbb{E} W_1(\mu, \hat\mu_n) \ = \ \begin{cases} O(n^{-1/2}), & d = 1, \\ O(n^{-1/2} \log n), & d = 2, \\ O(n^{-1/d}), & d \ge 3, \end{cases}$$

with high-probability analogues. Hence, by the triangle inequality, $W_1(\hat{\mu}_s, \hat{\mu}_\tau) \leq W_1(\hat{\mu}_s, \mu) + W_1(\mu, \hat{\mu}_\tau)$ yields

$$e_{\text{tr}}^{(W_1)}(k, n, \varepsilon) = \tilde{O}\big(k^{-1/d} + n^{-1/d}\big)$$

(or faster under low-dimensional/covering assumptions).

*(DM: MMD).* For bounded kernels, the empirical MMD concentrates at a CLT rate: $\big|\text{MMD}_k(\hat{\mu}, \hat{\nu}) - \text{MMD}_k(\mu, \nu)\big| = O_\P(1/\sqrt{m_\mu} + 1/\sqrt{m_\nu})$. Thus $e_{\text{tr}}^{(\text{MMD})} = \tilde{O}(1/\sqrt{k} + 1/\sqrt{n})$.

*(GM).* Define the scalar class $\mathcal{F} := \{ z \mapsto \langle v, g_a(\theta; z) \rangle : \theta \in \Gamma_a, \|v\|_2 \leq 1 \}$. Then $\sup_\theta \|\mathbb{E} g_a(\theta) - \mathbb{E}_{\hat{\mu}} g_a(\theta)\| = \sup_{f \in \mathcal{F}} (\mathbb{E} f - \mathbb{E}_{\hat{\mu}} f)$. By symmetrization and Rademacher complexity plus Ledoux–Talagrand contraction, under boundedness/Lipschitz envelopes we get

$$\sup_\theta \big\|\mathbb{E} g_a(\theta) - \mathbb{E}_{\hat{\mu}} g_a(\theta)\big\| = \tilde{O}\big(1/\sqrt{m}\big).$$

Applying this to both synthetic and real samples and averaging over anchors gives $e_{\text{tr}}^{(\text{GM})} = \tilde{O}(1/\sqrt{k} + 1/\sqrt{n})$.

*(TM).* The TM bridge (Proposition "TM bridge") plus the one-step recursion yields

$$\mathcal{M}_{\text{TM}} \leq \frac{\kappa_a}{(1 - \rho_a)\, \omega_{\min}} \sum_{t=0}^{L_b - 1} \eta_t \big\|\mathbb{E}_{\hat{\mu}_s} g_a(\theta_t^{(s)}) - \mathbb{E}_{\hat{\mu}_\tau} g_a(\theta_t^{(s)})\big\| + \frac{L_\theta}{1 - \rho_a}\, \varepsilon_{\text{path}}.$$

The same Rademacher argument as for GM applied at each $\theta_t^{(s)}$ gives a CLT rate scaled by the schedule factor $S_a := \frac{(L_\theta + 2/\bar{\eta})}{\omega_{\min}} \cdot \frac{\sum_t \eta_t}{1 - \rho_a}$:

$$e_{\text{tr}}^{(\text{TM})}(k, n, \varepsilon) = \tilde{O}\Big(S_a\Big(\frac{1}{\sqrt{k}} + \frac{1}{\sqrt{n}}\Big)\Big).$$

**Step E.5.8 Test-side concentration.** For the empirical test risk over $m$ samples, $e_{\text{te}}(m, \varepsilon) = O\big(\sqrt{\log(1/\varepsilon)/m}\big)$.

**Step E.5.9: Assemble all pieces.** Insert Eq. (128) and Eq. (129) into Eq. (127), and add the training-side and test-side penalties from Steps E.5.7–E.5.8:

$$\big|R_\nu(\theta_T^{(s,a)}) - R_\nu(\theta_T^{(\tau,a)})\big| \leq L_R \rho_a^T \|\delta_0\| + e_{\text{te}}(m, \varepsilon) + \mathbf{1}\{\phi = \text{TM}\} \frac{\kappa_a L_\theta}{C_{2,a}}\, \varepsilon_{\text{path}}$$

$$+ \frac{1}{C_{2,a}} \Big[ C_{\phi,a}\big((1 - \alpha_\phi)^J \mathcal{M}_\phi(\xi^{(0)}) + \epsilon_{\text{est}}^{(\phi)}\big) + e_{\text{tr}}^{(\phi)}(k, n, \varepsilon) \Big]. \quad (130)$$

$\square$

## E.6 Coverage-aware bound with dynamic outer progress (Theorem 6.5)

**Theorem E.7** (Coverage-aware bound with dynamic outer progress (formal))**.** *Let $(\mathcal{C}, D_{\mathcal{A}})$ be an configuration space. Fix a radius $\rho > 0$ and let $\mathcal{A} = \{a_1, \ldots, a_N\} \subset \mathcal{C}$ be a $\rho$-net (i.e., a $\rho$-packing that also $\rho$-covers $\mathcal{C}$; if only a $\rho$-packing is given, replace $\rho$ by $2\rho$ via the standard packing→covering conversion). For each center $a_j \in \mathcal{A}$, run $J$ outer steps with branch $\phi \in \{\text{DM}(W_1), \text{DM}(\text{MMD}), \text{GM}, \text{TM}\}$ and surrogate $\mathcal{M}_\phi$, under Assumption 4.1 uniformly over $a \in \mathcal{C}$ and the cross-configuration Lipschitz transfer Assumption 5.1:*

$$\|F_{\mu,a}(\theta) - F_{\mu,a'}(\theta)\| \leq L_{\text{conf}}\, d_{\mathcal{A}}(a, a') \qquad (\forall\, \theta,\, \mu \in \{\hat{\mu}_s, \hat{\mu}_\tau\}).$$

*Define the uniform constants*

$$C_{2,\min} := \inf_{a \in \mathcal{C}} \frac{1 - \rho_a}{L_R}, \quad C_{\phi,\max} := \sup_{a \in \mathcal{C}} \kappa_a\, B_{\phi,a}, \quad \alpha_{\phi,\min} := \inf_{a \in \mathcal{C}} \alpha_\phi(a),$$

*where $B_{\phi,a}$ are as in Theorem 6.3:*

$$B_{\phi,a} = \begin{cases} \eta\, L_{z,a}\ \text{or}\ \eta\, C_k, & \phi = \text{DM}(W_1),\ \text{DM}(\text{MMD}), \\ \eta\, |\Theta_j|, & \phi = \text{GM}, \\ (\eta L_{\theta,a} + 2)/\omega_{\min}, & \phi = \text{TM}. \end{cases}$$

*Then, with probability at least $1 - \varepsilon$,*

$$\sup_{a \in \mathcal{C}} \left| R_\nu(\theta_T^{(s,a)}) - R_\nu(\theta_T^{(\tau,a)}) \right| \leq \underbrace{\sup_{a \in \mathcal{C}} L_R(a) \rho_a^T \|\delta_0\|}_{\text{inner residual}} + \underbrace{e_{\text{te}}(m, \varepsilon)}_{\text{test}}$$

$$+ \underbrace{\frac{C_{\text{trans}} \rho}{C_{2,\min}}}_{\text{coverage transfer}} + \mathbf{1}\{\phi = \text{TM}\} \underbrace{\frac{\kappa_{\max} L_{\theta,\max}}{C_{2,\min}} \varepsilon_{\text{path}}}_{\text{TM discretization}}$$

$$+ \frac{1}{C_{2,\min}} \left[ C_{\phi,\max} \left( (1 - \alpha_{\phi,\min})^J \mathcal{M}_{\phi,\max}^{(0)} + \epsilon_{\text{est,max}}^{(\phi)} \right) + \widetilde{e}_{\text{tr}}^{(\phi)}(k, n, \varepsilon, N) \right], \quad (131)$$

*where*

$$\mathcal{M}_{\phi,\max}^{(0)} := \max_{1 \leq j \leq N} \mathcal{M}_\phi(\xi^{(0)}; a_j),$$

$$\kappa_{\max} := \sup_a \kappa_a, \quad L_{\theta,\max} := \sup_a L_{\theta,a},$$

$$C_{\text{trans}} := \kappa_{\max} L_{\text{conf}},$$

*and*

$$\widetilde{e}_{\text{tr}}^{(\phi)}(k, n, \varepsilon, N) = \begin{cases} \tilde{O}\left(k^{-1/d} + n^{-1/d}\right), & \phi = \text{DM}(W_1) \\ \tilde{O}\left(\left(\frac{1}{\sqrt{k}} + \frac{1}{\sqrt{n}}\right)\sqrt{\log N + \log(1/\varepsilon)}\right), & \phi = \text{DM}(\text{MMD}), \text{GM}, \text{TM}, \end{cases}$$

*and $N := |\mathcal{A}|$ is the covering number at scale $\rho$. The test-side term satisfies $e_{\text{te}}(m, \varepsilon) = O\left(\sqrt{\log(1/\varepsilon)/m}\right)$.*

*Proof.*

**Step E.5.1 Alignment transfer from $a$ to its center $a_j$.** Fix $a \in \mathcal{C}$ and take $a_j \in \mathcal{A}$ with $d_{\mathcal{A}}(a, a_j) \leq \rho$. By the triangle inequality, for any $\theta$,

$$\|F_{\hat{\mu}_s, a}(\theta) - F_{\hat{\mu}_\tau, a}(\theta)\| \leq \|F_{\hat{\mu}_s, a} - F_{\hat{\mu}_s, a_j}\| + \|F_{\hat{\mu}_s, a_j} - F_{\hat{\mu}_\tau, a_j}\| + \|F_{\hat{\mu}_\tau, a_j} - F_{\hat{\mu}_\tau, a}\|$$
$$\leq 2 L_{\text{conf}} d_{\mathcal{A}}(a, a_j) + \|F_{\hat{\mu}_s, a_j}(\theta) - F_{\hat{\mu}_\tau, a_j}(\theta)\|. \quad (132)$$

Taking the supremum over $\theta \in \Gamma_a$ yields

$$\Delta_a(\hat{\mu}_\tau, \hat{\mu}_s) \leq \Delta_{a_j}(\hat{\mu}_\tau, \hat{\mu}_s) + 2 L_{\text{conf}} \rho. \quad (133)$$

If the preconditioner norms $\|P_a(\theta)\|$ vary across configurations, we upper bound them by $\kappa_{\max} := \sup_{a,\theta} \|P_a(\theta)\|$ and absorb the variation into $C_{\text{trans}} := \kappa_{\max} L_{\text{conf}}$, giving the stated transfer constant.

**Step E.5.2 Risk–alignment reduction at $a$.** For any fixed configuration $a$, the single-configuration reduction gives

$$\left| R_\nu(\theta_T^{(s,a)}) - R_\nu(\theta_T^{(\tau,a)}) \right| \leq L_R(a) \rho_a^T \|\delta_0\| + \frac{1}{C_{2,a}} \Delta_a(\hat{\mu}_\tau, \hat{\mu}_s) + e_{\text{te}}(m, \varepsilon), \quad (134)$$

where $C_{2,a} = (1 - \rho_a)/L_R(a)$.

**Step E.5.3 Plug the transfer bound into the risk inequality.** Combine Eq. (133) and Eq. (134):

$$\left| R_\nu(\theta_T^{(s,a)}) - R_\nu(\theta_T^{(\tau,a)}) \right| \leq L_R(a) \rho_a^T \|\delta_0\| + \frac{1}{C_{2,a}} \left[ \Delta_{a_j}(\hat{\mu}_\tau, \hat{\mu}_s) + 2 C_{\text{trans}} \rho \right] + e_{\text{te}}(m, \varepsilon). \quad (135)$$

**Step E.5.4 Bridge and outer contraction at the trained centers.** By construction, we *train* only at the centers $a_j$. At each $a_j$, the branch-specific bridge (DM/GM/TM) and the outer contraction yield :

$$\Delta_{a_j}(\hat{\mu}_\tau, \hat{\mu}_s) \leq C_{\phi,a_j} \mathcal{M}_\phi(\xi^{(J)}; a_j) + \mathbf{1}\{\phi = \text{TM}\} \kappa_{a_j} L_{\theta,a_j} \varepsilon_{\text{path}}$$
$$\leq C_{\phi,a_j} \left( (1 - \alpha_\phi(a_j))^J \mathcal{M}_\phi(\xi^{(0)}; a_j) + \epsilon_{\text{est}}^{(\phi)}(a_j) \right) + \mathbf{1}\{\phi = \text{TM}\} \kappa_{a_j} L_{\theta,a_j} \varepsilon_{\text{path}}. \quad (136)$$

Recall $C_{\phi,a_j} = \kappa_{a_j} B_{\phi,a_j}$ with $B_{\phi,a_j} = \eta L_{z,a_j}$ or $\eta C_k$ (DM), $\eta|\Theta_j|$ (GM), $(\eta L_{\theta,a_j} + 2)/\omega_{\min}$ (TM).

**Step E.5.5 Uniformize constants and take sup over $a \in \mathcal{C}$.** Define the worst/best constants over $\mathcal{C}$:

$$C_{2,\min} := \inf_{a \in \mathcal{C}} \frac{1 - \rho_a}{L_R}, \quad C_{\phi,\max} := \sup_{a \in \mathcal{C}} C_{\phi,a},$$

$$\alpha_{\phi,\min} := \inf_{a \in \mathcal{C}} \alpha_\phi(a), \quad \kappa_{\max} := \sup_a \kappa_a, \quad L_{\theta,\max} := \sup_a L_{\theta,a}.$$

Let $\mathcal{M}^{(0)}_{\phi,\max} := \max_{1 \le j \le N} \mathcal{M}_\phi(\xi^{(0)}; a_j)$ and $\epsilon^{(\phi)}_{\text{est},\max} := \max_j \epsilon^{(\phi)}_{\text{est}}(a_j)$. Using $1/C_{2,a} \le 1/C_{2,\min}$ and Eq. (136) in Eq. (135) gives

$$\left| R_\nu(\theta_T^{(s,a)}) - R_\nu(\theta_T^{(\tau,a)}) \right| \le \underbrace{L_R(a)\rho_a^T \|\delta_0\|}_{\text{inner residual}} + e_{\text{te}}(m, \varepsilon) + \frac{2 C_{\text{trans}} \rho}{C_{2,\min}}$$

$$+ \frac{1}{C_{2,\min}} \left[ C_{\phi,\max} \left( (1 - \alpha_{\phi,\min})^J \mathcal{M}^{(0)}_{\phi,\max} + \epsilon^{(\phi)}_{\text{est},\max} \right) \right]$$

$$+ \mathbf{1}\{\phi = \text{TM}\} \frac{\kappa_{\max} L_{\theta,\max}}{C_{2,\min}} \varepsilon_{\text{path}}. \tag{137}$$

Taking the supremum in $a \in \mathcal{C}$ replaces $L_R(a)\rho_a^T$ by $\sup_a L_R(a)\rho_a^T$ on the first term, leaving the rest unchanged.

**Step E.5.6 Coverage-aware training-side concentration over $N$ centers.** We now upgrade the training-side surrogate estimation to hold *uniformly* over the $N$ trained centers. This produces the coverage-aware term $\widetilde{e}^{(\phi)}_{\text{tr}}(k, n, \varepsilon, N)$ stated in the theorem.

*DM($W_1$).* For empirical measures in $\mathbb{R}^d$, nonasymptotic bounds give $W_1(\mu, \hat{\mu}_m) = O_\P(m^{-1/d})$ for $d \ge 3$, $O_\P(m^{-1/2} \log m)$ for $d = 2$, and $O_\P(m^{-1/2})$ for $d = 1$. A union bound over $N$ centers multiplies failure probability by $N$; in the $\tilde{O}(\cdot)$ notation (suppressing polylog factors), we retain the geometric-rate term:

$$\widetilde{e}^{(W_1)}_{\text{tr}}(k, n, \varepsilon, N) = \tilde{O}\big(k^{-1/d} + n^{-1/d}\big).$$

*DM(MMD) & GM.* For bounded kernels, $\text{MMD}_k$ concentrates at CLT rate; for GM, let $\mathcal{F} := \{z \mapsto \langle v, g_a(\theta; z) \rangle : \|v\| \le 1, \theta \in \Gamma\}$ and apply symmetrization + Rademacher complexity with Ledoux–Talagrand's contraction to obtain $O_\P(1/\sqrt{m})$. A union bound over $N$ centers contributes a $\sqrt{\log N + \log(1/\varepsilon)}$ factor:

$$\widetilde{e}^{(\text{MMD})}_{\text{tr}}, \widetilde{e}^{(\text{GM})}_{\text{tr}} = \tilde{O}\Big(\big(\tfrac{1}{\sqrt{k}} + \tfrac{1}{\sqrt{n}}\big) \sqrt{\log N + \log(1/\varepsilon)}\Big).$$

*TM.* The TM bridge plus the one-step recursion shows that the TM surrogate aggregates $L_b$ CLT-scale deviations, scaled by the schedule factor

$$S_a = \frac{(L_{\theta,a} + 2/\bar{\eta}_a)}{\omega_{\min,a}} \cdot \frac{\sum_t \eta_{t,a}}{1 - \rho_a}.$$

Uniformizing over $a \in \mathcal{C}$ (and thus over centers) and applying the same union bound gives

$$\widetilde{e}^{(\text{TM})}_{\text{tr}} = \tilde{O}\Big(S_{\max}\big(\tfrac{1}{\sqrt{k}} + \tfrac{1}{\sqrt{n}}\big) \sqrt{\log N + \log(1/\varepsilon)}\Big), \qquad S_{\max} := \sup_{a \in \mathcal{C}} S_a.$$

**Step E.5.7 Assemble and rename constants.** Collect the inner residual into $\sup_{a \in \mathcal{C}} L_R(a)\rho_a^T \|\delta_0\|$, keep $e_{\text{te}}(m, \varepsilon)$ unchanged, and define $C_{\text{trans}} := \kappa_{\max} L_{\text{conf}}$.

$$\mathcal{M}^{(0)}_{\phi,\max} := \max_j \mathcal{M}_\phi(\xi^{(0)}; a_j), \quad \epsilon^{(\phi)}_{\text{est},\max} := \max_j \epsilon^{(\phi)}_{\text{est}}(a_j), \quad N := |\mathcal{A}|.$$

---

**Algorithm 1:** Single-configuration evaluation

---

**Input:** dataset $\mathcal{D}$, distilled sets $\{\mathcal{S}_k\}$ for budgets $k$, source configuration $a_0$
**Output:** points $\{(k, \Delta_{a_0}(k))\}$ and linear fit of $\Delta$ vs. $1/\sqrt{k}$
**foreach** $k$ **do**
    load distilled set $\mathcal{S}_k$
    **for** *repeat* $r = 1..R$ **do**
        initialize student $\theta \sim a_0$
        train $\theta$ on $\mathcal{S}_k$ using the student protocol of $a_0$
        evaluate accuracy $\mathrm{Acc}_{\mathrm{syn}}(k, r)$ on the test split of $\mathcal{D}$
    train a real-data baseline once under $a_0$ to obtain $\mathrm{Acc}_{\mathrm{real}}$
    set $\Delta_{a_0}(k) = \mathrm{Acc}_{\mathrm{real}} - \mathrm{mean}_r \, \mathrm{Acc}_{\mathrm{syn}}(k, r)$
Fit a line $y = ax + b$ with $x = 1/\sqrt{k}$ and $y = \Delta_{a_0}(k)$; report slope/intercept/$R^2$.

---

**Algorithm 2:** Configuration coverage: from per-configuration curves to the coverage law

---

**Input:** dataset $\mathcal{D}$, distilled sets $\{\mathcal{S}_k\}$, configuration family $\mathcal{C}$
**Output:** coverage points $\{(X, Y)\}$ with $X = \sqrt{\log m}/\sqrt{k}$ and $Y = \Delta(k, m)$; global fit
**foreach** $k$ **do**
    **foreach** *configuration* $a \in \mathcal{C}$ **do**
        train a student $\theta_a$ on $\mathcal{S}_k$ under $a$ and record $\mathrm{Acc}_{\mathrm{syn}}(k, a)$
        obtain once-per-configuration real baseline $\mathrm{Acc}_{\mathrm{real}}(a)$
        set $\Delta(k, a) = \mathrm{Acc}_{\mathrm{real}}(a) - \mathrm{Acc}_{\mathrm{syn}}(k, a)$
Let $A$ be the ordered list of configurations used.
**for** $m = 1, \cdots, |A|$ **do**
    choose a size-$m$ subset of configurations (prefix or random) and denote it $A_m$
    **foreach** $k$ **do**
        set $Y = \Delta(k, m) = \frac{1}{m} \sum_{a \in A_m} \Delta(k, a)$, set $X = \sqrt{\log m}/\sqrt{k}$; append $(X, Y)$ to
        the coverage set
Fit a single line $Y = aX + b$ over all coverage points; report slope/intercept/$R^2$.

---

Plugging the center-wise bridge+contraction Eq. (136) and the coverage-aware training terms into Eq. (137) yields

$$\sup_{a \in \mathcal{C}} \left| R_\nu(\theta_T^{(s,a)}) - R_\nu(\theta_T^{(\tau,a)}) \right| \leq \underbrace{\sup_{a \in \mathcal{C}} L_R(a) \rho_a^T \|\delta_0\| + e_{\mathrm{te}}(m, \varepsilon)}_{\text{configuration/branch/}k\text{-independent floor}} \tag{138}$$

$$+ \frac{2 C_{\mathrm{trans}} \rho}{C_{2,\min}} + \mathbf{1}\{\phi = \mathrm{TM}\} \frac{\kappa_{\max} L_{\theta,\max}}{C_{2,\min}} \varepsilon_{\mathrm{path}} \tag{139}$$

$$+ \frac{1}{C_{2,\min}} \left[ C_{\phi,\max} \left( (1 - \alpha_{\phi,\min})^J \mathcal{M}_{\phi,\max}^{(0)} + \epsilon_{\mathrm{est,max}}^{(\phi)} \right) + \widetilde{e}_{\mathrm{tr}}^{(\phi)}(k, n, \varepsilon, N) \right]. \tag{140}$$

$\square$

# F   Experiments

## F.1   Detailed Experimental Setup

**Datasets.** We evaluate on **MNIST**, **CIFAR-10/100** (official train/test splits), and **ImageNette** (a 10-class ImageNet subset). For all datasets we follow the preprocessing prescribed by the respective baseline implementations (e.g., normalization, ZCA whitening when enabled), ensuring strict comparability.

**Distillation methods.** We study three established *matching-based* families—*Gradient Matching* (GM: DC/DSA), *Distribution Matching* (DM), and *Trajectory Matching* (TM: MTT)—and further include a recent *diffusion-based* pipeline (MGD$^3$) on ImageNette to test whether our theory extends beyond the matching paradigm. For each method we rely exclusively on the authors' open-source repositories with their default hyperparameters; the only controlled variable is the distilled budget $k$ (via IPC). Each distillation run is executed to completion under the default schedules of the respective methods.

**Training configurations.** A configuration $a$ is defined as a triplet *(architecture $\times$ optimizer $\times$ augmentation)*. Architectures include `ConvNet`, `LeNet`, `ResNet-18`, and `AlexNet`; for coverage experiments on CIFAR-10/100 we additionally use `MLP` and `VGG11`. Optimizers are `SGD` (momentum 0.9, weight decay $5 \times 10^{-4}$) and `Adam` (default betas). Augmentation is either `none` or `DSA` when enabled by the baseline. Distillation is always performed in a fixed *source configuration* (`ConvNet+SGD`, with DSA on when applicable), and target configurations are evaluated across diverse *target configurations* sampled from $\mathcal{C}$.

**Distillation budget.** We sweep images-per-class values IPC $\in$ $\{1, 2, 4, 6, 8, 12, 18, 28, 51, 100, 200\}$ (up to 100 on CIFAR-100), yielding a total distilled size of $k = \text{IPC} \times \#\text{classes}$. All methods are evaluated on the same IPC grid.

**Evaluation metric.** We report the generalization error

$$\Delta = \big| \hat{R}(\theta_T^{(\hat{\mu}_s, a)}) - \hat{R}(\theta_T^{(\hat{\mu}_\tau, a)}) \big|,$$

the accuracy gap between training on distilled and real data within the same configuration $a$. Each result is averaged over 5 independent repeats; for single-configuration runs we regress $\Delta$ against $1/\sqrt{k}$ and report slope, intercept, and $R^2$.

**Target configuration training protocol.** Students are trained from scratch with batch size 256, strictly following the DC/DSA evaluation protocol. When DSA is enabled, students are trained for 1000 epochs; otherwise, for 300 epochs. SGD uses an initial learning rate of 0.01 with momentum 0.9 and weight decay $5 \times 10^{-4}$, decayed $\times 0.1$ midway through training. Adam uses its default settings. Architectures (`ConvNet`, `LeNet`, `ResNet-18`, `AlexNet`) all follow this protocol.

**Coverage-law construction.** To test the predicted scaling with coverage complexity, we aggregate multiple configurations. For each subset of size $m$, and each IPC $k$, we compute the averaged gap

$$Y = \Delta(k, m), \quad X = \frac{\sqrt{\log m}}{\sqrt{k}}.$$

We then regress $Y$ against $X$ to test the coverage law $Y \propto X$. Two subset strategies are used: `prefix` (deterministic) and `random` (averaged over $T=5$ trials). For each configuration included, the *real-data baseline* is trained once on the full dataset with the identical optimizer, epochs, and augmentation as in the distilled run; this baseline is reused across $k$.

**Hardware and software.** Experiments are conducted on servers with AMD EPYC 7642 CPUs (96 vCPUs), CUDA 12.4, and up to 4$\times$NVIDIA RTX 4090 GPUs.

F.2 ALGORITHMS

For clarity, we briefly summarize the two evaluation protocols. In the *single-configuration evaluation* (Algorithm 1), we fix a source configuration and train students on distilled datasets of varying budget $k$. Each student is evaluated against its real-data counterpart in the same configuration, and the resulting accuracy gaps $\Delta(k)$ are regressed against $1/\sqrt{k}$ to reveal the single-configuration scaling law.

In the *coverage-law evaluation* (Algorithm 2), we extend the analysis across multiple target configurations. For each subset of size $m$ drawn from the configuration family $\mathcal{C}$, we average the gaps over the selected configurations to obtain $\overline{\Delta}(k, m)$. Plotting $\overline{\Delta}$ against $\sqrt{\log m}/\sqrt{k}$ tests the predicted coverage law, and the slope of the regression quantifies the penalty induced by ecological diversity.

## G  Limitation and Future Work

Despite providing a unified Configuration-dynamics-error framework, our study still has several limitations that we explicitly acknowledge.

**Assumptions on optimization dynamics.** Our bounds rely on Polyak–Łojasiewicz (PL) contraction and Lipschitz continuity of update fields (Assumption 4.1). These assumptions hold for SGD variants under moderate learning rates, but may fail in regimes such as large-batch training, adaptive optimizers (e.g., AdamW), or architectures with non-smooth objectives. Extending our results to weaker conditions such as one-point convexity or uniform stability is an important open direction. Concretely, one next step is to verify whether the single-configuration scaling law (theorem 4.2) still exhibits linear $1/\sqrt{k}$ behavior when training with AdamW or adaptive schedulers, and to adapt the proof techniques accordingly.

**Coverage complexity estimation.** Our coverage law (Theorem 5.1, Corollary 5.3) shows that risk scales as $\Delta(k, m) \propto \sqrt{\mathcal{H}_{\mathrm{cov}}}/\sqrt{k}$, with $\mathcal{H}_{\mathrm{cov}}(r)$ the coverage complexity under configuration-distance $d_{\mathcal{A}}$. In experiments, we approximate $\mathcal{H}_{\mathrm{cov}}$ by $\sqrt{\log m}$ with $m$ configurations, which can underestimate heterogeneity when optimizers or architectures differ sharply in $d_{\mathcal{A}}$. This limits the direct deployment of our bounds. A concrete next step is to develop empirical estimators of $\mathcal{H}_{\mathrm{cov}}$, for instance by clustering configurations in the $d_{\mathcal{A}}$ metric space and allocating distilled prototypes adaptively with respect to cluster counts, rather than raw $m$.

**Ecological scope beyond algorithmic variation.** In this work we define configurations by algorithmic choices (optimizer, architecture, augmentation). This abstraction omits distributional or semantic shifts, such as cross-domain transfer, class imbalance, or multimodal inputs. Consequently, our current coverage law captures algorithmic but not data-level diversity. A concrete next step is to extend $d_{\mathcal{A}}$ to include a distributional term (e.g., Wasserstein or MMD distance between domains) and evaluate whether the scaling $\Delta \propto \sqrt{\mathcal{H}_{\mathrm{cov}}}/\sqrt{k}$ continues to hold in cross-domain settings (e.g., CIFAR-10 $\rightarrow$ STL-10, ImageNet $\rightarrow$ DomainNet).

In summary, these limitations highlight well-scoped extensions: relaxing optimization assumptions, sharpening coverage estimation, and expanding configuration definitions to data-level shifts. We view these as promising future directions rather than weaknesses of the current framework.

## H  Usage of LLM

In preparing this work, we made limited use of ChatGPT (OpenAI) as a supportive tool. Specifically, it was consulted in two ways:

- **Coding support**: ChatGPT-4o was occasionally used during debugging to suggest possible corrections for coding errors. All implementations were written, tested, and verified independently by the authors.
- **Language polishing**: At the final stage of manuscript preparation, ChatGPT-5 was used to polish the English expression of the appendix. The suggestions were carefully reviewed and adapted by the authors to ensure accuracy and consistency with the original technical content.

No AI tool was involved in generating research ideas, conducting experiments, or drawing conclusions. All scientific contributions are the authors' own.

