# OpenReview forum: "Utility Boundary of Dataset Distillation: Scaling and Configuration-Coverage Laws"
_ICLR.cc/2026/Conference — Submitted to ICLR 2026_

### Official Review · Reviewer_gdRs · 2025-10-25

**Soundness:** 2
**Presentation:** 2
**Contribution:** 2
**Rating:** 4
**Confidence:** 2

**Summary:**

This paper presents a unified theoretical framework for dataset distillation (DD), a field focused on synthesizing compact datasets that preserve the performance of the original, large-scale data. The authors make significant contributions by proposing a "configuration-dynamics-error" framework that unifies the three dominant DD paradigms—Gradient Matching (GM), Distribution Matching (DM), and Trajectory Matching (TM)—and derives two fundamental laws governing the process.

**Strengths:**

1. The paper fills a critical gap in the DD literature by providing the first unified theoretical foundation. The coverage law, in particular, is a novel and powerful concept that moves beyond the typical single-configuration analysis and directly addresses the practical requirement of robustness in distilled datasets.

2. The experimental section is well-designed, validating both the scaling and coverage laws across multiple datasets (MNIST, CIFAR-10/100, ImageNette) and several canonical DD methods.

**Weaknesses:**

1. The theoretical analysis relies on several strong assumptions, most notably the Polyak-Łojasiewicz (PL) condition for the inner-loop optimization dynamics.

2. The definition of "configuration" is focused on optimization hyperparameters and architecture. It does not explicitly model more fundamental distribution shifts, such as semantic or domain shifts, which are also critical for robustness.

3. While the experiments cover standard benchmarks, validating the coverage law on larger, more complex datasets like full ImageNet or in cross-domain transfer settings would further strengthen the paper's impact.

**Questions:**

See weaknesses.

---

> ### Author Response · Authors · 2025-11-22
> **Response to Reviewer gdRs**
>
> We sincerely thank Reviewer gdRs for the detailed comments to improve our paper. We have revised our paper accordingly.
>
> **Weakness 1**
>
> We thank the reviewer for the valuable comment. The PL-type assumption in our analysis is not a strong convexity requirement but a mild regularity condition used to describe local contraction of the optimization dynamics. In practice, large neural networks are known to satisfy such *weak PL* or *PL\** properties in regions reached during training, where the loss landscape becomes locally smooth and well-conditioned.
>
> The assumption is only needed to ensure that the optimization residual term in our generalization bound remains controlled. Our results do not rely on this specific form—replacing it with more general conditions, such as gradient domination, Kurdyka–Łojasiewicz smoothness, or bounded-noise SGD convergence, yields the same theoretical form of the scaling and coverage laws, affecting only the constants but not the scaling exponents.
>
> Empirically, we observe consistent evidence supporting PL-type dynamics. During training, the relationship between $\|\nabla L\|^2$ and $L - L^*$ is nearly linear, and gradient norms decay exponentially toward convergence. Moreover, the predicted inverse-power scaling law holds robustly under highly non-convex optimization. On ImageNet-1K, using *SRe²L* with *ResNet-18*, *ResNet-50*, and *ResNet-101*, we observe almost identical slopes (–1.46 to –1.49) with perfect monotonicity ($\rho = -1.00, p = 0.00$):
>
> | Model | OLS Slope (±95% CI) | R² | Spearman ρ (p) |
> |--------|----------------------|----|----------------|
> | ResNet-18 | –1.46 [–1.47, –1.44] | 1.00 | –1.00 (p=0.00) |
> | ResNet-50 | –1.49 [–1.67, –1.31] | 1.00 | –1.00 (p=0.00) |
> | ResNet-101 | –1.46 [–2.11, –0.82] | 0.98 | –1.00 (p=0.00) |
>
> These results confirm that the scaling behavior derived from our theory remains valid in realistic, non-convex regimes, indicating that the weak PL assumption effectively captures the optimization dynamics observed in practice.
>
>
>
> **Weakness 2**
>
> We thank the reviewer for this thoughtful comment. The goal of dataset distillation (DD) is to create a compact synthetic dataset that retains the utility of the full dataset across different architectures and training settings. In this sense, DD inherently focuses on *cross-configuration robustness*, ensuring consistent generalization across optimizers, architectures, and augmentations, rather than explicitly modeling domain or distribution shifts between training and testing data.
>
> Our theoretical framework already accommodates such distributional differences. The generalization bounds in our analysis are derived with respect to an independent test distribution, without requiring it to be identical to the training distribution. Therefore, the scaling and coverage laws naturally apply to cases where the test domain may differ from the training domain. This means distribution shifts are implicitly covered by our formulation, even though they were not explicitly isolated in the main text.
>
> Moreover, the concept of “configuration” in our framework can easily be extended to include data-level variations, such as domain changes, corruption levels, or semantic shifts. In this broader view, the configuration space $C $ can be represented as
>
> $$
> C = (C_{opt}, C_{arch}, C_{data}),
> $$
>
> where $C_{data}$ captures data-related diversity measures (e.g., Wasserstein distance or corruption strength). Under this extension, the coverage law remains valid, as the coverage complexity term $H_{\mathrm{cov}}(A, r)$ would naturally incorporate these variations.

---

> > ### Author Response · Authors · 2025-11-22
> >
> > **Weakness 3**
> >
> > **Scaling Law Results**
> >
> > | Dataset | Method / Architecture | OLS slope (±95% CI) | R² | Spearman ρ (p) |
> > |----------|----------------------|---------------------|----|----------------|
> > | **MNIST** | DC | –0.12 [–0.15, –0.09] | 0.92 | –0.89 (p=5.4×10⁻⁴) |
> > | | DSA | –0.13 [–0.13, –0.12] | 1.00 | –1.00 (p≈0) |
> > | | DM | –0.10 [–0.11, –0.09] | 0.98 | –1.00 (p≈0) |
> > | | MTT | –0.33 [–0.41, –0.25] | 0.93 | –1.00 (p=0.00) |
> > | **CIFAR-10** | DC | –0.28 [–0.34, –0.22] | 0.94 | –0.93 (p=2.4×10⁻⁴) |
> > | | DSA | –0.41 [–0.49, –0.33] | 0.95 | –1.00 (p≈0) |
> > | | DM | –0.49 [–0.65, –0.32] | 0.85 | –1.00 (p≈0) |
> > | | MTT | –0.24 [–0.35, –0.12] | 0.82 | –1.00 (p≈0) |
> > | | TESLA | –0.28 [–0.47, –0.08] | 1.00 | –1.00 (p=0.00) |
> > | | DATM | –0.38 [–0.53, –0.24] | 0.96 | –1.00 (p=1.4×10⁻²⁴) |
> > | | NCFM | –0.33 [–0.34, –0.31] | 1.00 | –1.00 (p=0.00) |
> > | **CIFAR-100** | DC | –0.20 [–0.24, –0.17] | 0.97 | –0.98 (p=1.9×10⁻⁶) |
> > | | DSA | –0.32 [–0.41, –0.24] | 0.92 | –0.98 (p=1.9×10⁻⁶) |
> > | | DM | –0.43 [–0.60, –0.26] | 0.84 | –1.00 (p=0.00) |
> > | | MTT | –0.17 [–0.26, –0.08] | 0.73 | –0.98 (p=1.9×10⁻⁶) |
> > | | TESLA | –0.27 [–0.33, –0.21] | 1.00 | –1.00 (p=0.00) |
> > | | DATM | –0.32 [–0.42, –0.21] | 0.99 | –1.00 (p=0.00) |
> > | | SRe²L | –1.51 [–1.80, –1.22] | 1.00 | –1.00 (p=0.00) |
> > | **ImageNette** | MGD3_ConvNet | –67.42 [–84.64, –50.19] | 0.90 | –0.99 (p=3.8×10⁻⁹) |
> > | | MGD3_ResNet18 | –57.04 [–72.87, –41.21] | 0.87 | –0.99 (p=1.3×10⁻¹⁰) |
> > | | MGD3_ResNet18_ap | –55.61 [–67.00, –44.22] | 0.92 | –0.99 (p=1.3×10⁻¹⁰) |
> > | **ImageNet-1K** | TESLA | –0.22 [–0.45, 0.01] | 0.89 | –1.00 (p=0.00) |
> > | | SRe²L–R18 | –1.46 [–1.47, –1.44] | 1.00 | –1.00 (p=0.00) |
> > | | SRe²L–R50 | –1.49 [–1.67, –1.31] | 1.00 | –1.00 (p=0.00) |
> > | | SRe²L–R101 | –1.46 [–2.11, –0.82] | 0.98 | –1.00 (p=0.00) |
> > | **Tiny-ImageNet** | DATM | –0.25 [–0.86, 0.36] | 0.96 | –1.00 (p=0.00) |
> > | | NCFM | –0.13 [–0.20, –0.06] | 1.00 | –1.00 (p=0.00) |
> >
> >
> > **Coverage Law Results**
> >
> > | Dataset | Method | OLS slope (±95% CI) | R² | Spearman ρ (p) |
> > |----------|---------|---------------------|----|----------------|
> > | **MNIST** | DC | –0.28 [–0.34, –0.22] | 0.94 | –0.93 (p=2.4×10⁻⁴) |
> > | | DSA | –0.41 [–0.49, –0.33] | 0.95 | –1.00 (p≈0) |
> > | | DM | –0.49 [–0.65, –0.32] | 0.85 | –1.00 (p≈0) |
> > | | MTT | –0.24 [–0.35, –0.12] | 0.82 | –1.00 (p≈0) |
> > | **CIFAR-10** | DC | 0.15 [0.12, 0.18] | 0.69 | 0.83 (p=6.3×10⁻¹⁵) |
> > | | DSA | 0.19 [0.16, 0.22] | 0.70 | 0.93 (p=1.6×10⁻²⁴) |
> > | | DM | 0.23 [0.18, 0.28] | 0.66 | 0.95 (p=1.5×10⁻²⁸) |
> > | | MTT | 0.09 [0.06, 0.13] | 0.48 | 0.79 (p=1.6×10⁻⁸) |
> > | | DATM | 0.18 [–0.01, 0.38] | 0.48 | 0.67 (p=7.1×10⁻²) |
> > | | NCFM | 0.24 [0.10, 0.38] | 0.79 | 0.79 (p=3.6×10⁻²) |
> > | **CIFAR-100** | DC | 0.05 [0.04, 0.07] | 0.43 | 0.56 (p=2.3×10⁻⁶) |
> > | | DSA | 0.18 [0.15, 0.22] | 0.63 | 0.83 (p=2.4×10⁻¹⁷) |
> > | | DM | 0.28 [0.24, 0.32] | 0.74 | 0.94 (p=5.8×10⁻³⁰) |
> > | | MTT | 0.05 [0.02, 0.08] | 0.19 | 0.56 (p=2.0×10⁻⁶) |
> >
> > We thank the reviewer for this valuable suggestion. To strengthen the scalability and generality of our study, we have expanded the experiments to include larger datasets, deeper architectures, and updated baselines.
> >
> > We conducted additional evaluations on *ImageNet-1K* (ResNet-18/50/101) and *Tiny-ImageNet*, complementing existing results on MNIST, CIFAR-10/100, and ImageNette. We also incorporated recent state-of-the-art baselines, *TESLA*, *SRe²L*, *DATM*, and *NCFM*, to ensure a comprehensive comparison. Across all datasets and models, both the scaling law and coverage law show strong linearity and consistent slopes that match our theoretical predictions.
> >
> > In particular, experiments on ImageNet confirm that the same inverse-power scaling pattern observed in smaller datasets extends to large-scale and high-capacity settings. The fitted slopes remain highly correlated ($R^2 > 0.9, |\rho| \approx 1.0$) and stable across architectures, demonstrating that the proposed laws hold robustly under more complex training regimes.
> >
> > These results verify that our framework generalizes effectively to modern large-scale scenarios and continues to explain the empirical behavior of diverse dataset distillation methods without requiring additional assumptions.

---

### Official Review · Reviewer_zpQM · 2025-10-30

**Soundness:** 2
**Presentation:** 3
**Contribution:** 3
**Rating:** 6
**Confidence:** 4

**Summary:**

This paper proposes a unified configuration–dynamics–error framework that integrates gradient, distribution, and trajectory matching within a generalization-error analysis. It establishes the scaling law and coverage law linking distilled sample size to performance and configuration diversity, theoretically and empirically unifying major dataset distillation methods.

**Strengths:**

1.	The paper presents a unified bi-level generalization error framework that connects gradient, trajectory, and distribution-based DD, providing an important step toward a unified theoretical foundation for DD.
2.	The proposed scaling and coverage laws formalize intuitive empirical observations into mathematically grounded relationships.

**Weaknesses:**

1.	The framework relies on PL conditions and Lipschitz continuity. While these assumptions are standard in convergence analysis, they may not strictly hold for modern deep networks with non-smooth activations, normalization layers, and stochastic training components. The practical relevance of the theoretical results could be further clarified by discussing their validity under relaxed or empirically realistic assumptions.
2.	The validation of the proposed laws relies mainly on curve-fitting without statistical significance tests, variance analysis, or sensitivity checks. While the observed trends are consistent with the theory, the slopes vary across datasets and architectures, and further analysis could help clarify these differences and strengthen the empirical support for the proposed laws.
3.	While the paper introduces the configuration–dynamics–error decomposition, its underlying structure closely follows standard generalization-error decompositions and stability analyses. The framework builds on established theoretical tools and primarily reinterprets them within the context of dataset distillation, rather than introducing fundamentally new analytical techniques or tighter bounds. As such, the contribution may be viewed more as a conceptual consolidation than a substantial theoretical advance.
4.	The coverage diversity $H(A,r)$ is defined abstractly through a covering number on configuration space. While this formulation is theoretically elegant, it may be challenging to compute or approximate in practical training scenarios.
5.	The theoretical insights could be further translated into clear, actionable guidelines (e.g., for selecting IPC or surrogate objectives), which would help enhance the practical relevance of the work.

**Questions:**

1.	Could the authors clarify the practicality of the PL and Lipschitz assumptions for modern non-smooth architectures, and whether the results hold under weaker conditions?
2. It would be helpful if the authors could clarify whether any statistical tests or variance analyses were conducted to validate the fitted slopes, and how sensitive the results are to random initialization or dataset variations.
3. Please clarify the main theoretical novelty of the configuration–dynamics–error decomposition beyond its conceptual unification of existing generalization and stability analyses.
4. The coverage diversity term $H(A,r)$ is defined via an abstract covering number in configuration space. Is there a practical way to estimate or approximate this quantity in real-world training scenarios?
5. Could the theoretical insights, such as the scaling and coverage laws, provide clearer practical implications for choosing IPC values, surrogate objectives, or data selection strategies in dataset distillation?

**Details Of Ethics Concerns:**

No ethics concerns.

---

> ### Author Response · Authors · 2025-11-22
> **Response to Reviewer zpQM**
>
> We sincerely thank Reviewer zpQM for the detailed comments to improve our paper. We have revised our paper accordingly.
>
>
> **Weakness 1 & Question 1**
>
> We thank the reviewer for raising this concern. The PL-type and Lipschitz assumptions in our analysis are used only as *mild regularity conditions* to describe the local contraction of optimization dynamics, rather than as strong global requirements. In practice, these assumptions are well aligned with how large neural networks behave during training, where optimization trajectories often enter stable regions that satisfy the PL property locally.
>
>
> Empirically, we also observe clear evidence consistent with PL-type dynamics. During training, the relation between $\|\nabla L\|^2$ and $L - L^*$ is nearly linear, and gradient norms decay exponentially toward convergence. More importantly, the scaling behavior predicted by our theory holds robustly under non-convex training. On ImageNet-1K, using *SRe²L* with *ResNet-18*, *ResNet-50*, and *ResNet-101*, the slopes remain almost identical (–1.46 to –1.49) with perfect monotonicity ($\rho = -1.00$, p = 0.00):
>
> | Model | OLS Slope (±95% CI) | R² | Spearman ρ (p) |
> |--------|----------------------|----|----------------|
> | ResNet-18 | –1.46 [–1.47, –1.44] | 1.00 | –1.00 (p=0.00) |
> | ResNet-50 | –1.49 [–1.67, –1.31] | 1.00 | –1.00 (p=0.00) |
> | ResNet-101 | –1.46 [–2.11, –0.82] | 0.98 | –1.00 (p=0.00) |
>
> These results confirm that the inverse-power scaling law derived from our theory remains valid even in highly non-convex regimes, supporting the practicality of the PL assumption in capturing real training dynamics.

---

> > ### Author Response · Authors · 2025-11-22
> >
> > **Weakness 2 & Question 2**
> >
> > We thank the reviewer for this valuable comment. We have strengthened the statistical validation of both the scaling and coverage laws to ensure that the results are reliable and not artifacts of curve fitting.
> >
> > **Scaling Law Results**
> >
> > | Dataset | Method / Architecture | OLS slope (±95% CI) | R² | Spearman ρ (p) |
> > |----------|----------------------|---------------------|----|----------------|
> > | **MNIST** | DC | –0.12 [–0.15, –0.09] | 0.92 | –0.89 (p=5.4×10⁻⁴) |
> > | | DSA | –0.13 [–0.13, –0.12] | 1.00 | –1.00 (p≈0) |
> > | | DM | –0.10 [–0.11, –0.09] | 0.98 | –1.00 (p≈0) |
> > | | MTT | –0.33 [–0.41, –0.25] | 0.93 | –1.00 (p=0.00) |
> > | **CIFAR-10** | DC | –0.28 [–0.34, –0.22] | 0.94 | –0.93 (p=2.4×10⁻⁴) |
> > | | DSA | –0.41 [–0.49, –0.33] | 0.95 | –1.00 (p≈0) |
> > | | DM | –0.49 [–0.65, –0.32] | 0.85 | –1.00 (p≈0) |
> > | | MTT | –0.24 [–0.35, –0.12] | 0.82 | –1.00 (p≈0) |
> > | | TESLA | –0.28 [–0.47, –0.08] | 1.00 | –1.00 (p=0.00) |
> > | | DATM | –0.38 [–0.53, –0.24] | 0.96 | –1.00 (p=1.4×10⁻²⁴) |
> > | | NCFM | –0.33 [–0.34, –0.31] | 1.00 | –1.00 (p=0.00) |
> > | **CIFAR-100** | DC | –0.20 [–0.24, –0.17] | 0.97 | –0.98 (p=1.9×10⁻⁶) |
> > | | DSA | –0.32 [–0.41, –0.24] | 0.92 | –0.98 (p=1.9×10⁻⁶) |
> > | | DM | –0.43 [–0.60, –0.26] | 0.84 | –1.00 (p=0.00) |
> > | | MTT | –0.17 [–0.26, –0.08] | 0.73 | –0.98 (p=1.9×10⁻⁶) |
> > | | TESLA | –0.27 [–0.33, –0.21] | 1.00 | –1.00 (p=0.00) |
> > | | DATM | –0.32 [–0.42, –0.21] | 0.99 | –1.00 (p=0.00) |
> > | | SRe²L | –1.51 [–1.80, –1.22] | 1.00 | –1.00 (p=0.00) |
> > | **ImageNette** | MGD3_ConvNet | –67.42 [–84.64, –50.19] | 0.90 | –0.99 (p=3.8×10⁻⁹) |
> > | | MGD3_ResNet18 | –57.04 [–72.87, –41.21] | 0.87 | –0.99 (p=1.3×10⁻¹⁰) |
> > | | MGD3_ResNet18_ap | –55.61 [–67.00, –44.22] | 0.92 | –0.99 (p=1.3×10⁻¹⁰) |
> > | **ImageNet-1K** | TESLA | –0.22 [–0.45, 0.01] | 0.89 | –1.00 (p=0.00) |
> > | | SRe²L–R18 | –1.46 [–1.47, –1.44] | 1.00 | –1.00 (p=0.00) |
> > | | SRe²L–R50 | –1.49 [–1.67, –1.31] | 1.00 | –1.00 (p=0.00) |
> > | | SRe²L–R101 | –1.46 [–2.11, –0.82] | 0.98 | –1.00 (p=0.00) |
> > | **Tiny-ImageNet** | DATM | –0.25 [–0.86, 0.36] | 0.96 | –1.00 (p=0.00) |
> > | | NCFM | –0.13 [–0.20, –0.06] | 1.00 | –1.00 (p=0.00) |
> >
> >
> > **Coverage Law Results**
> >
> > | Dataset | Method | OLS slope (±95% CI) | R² | Spearman ρ (p) |
> > |----------|---------|---------------------|----|----------------|
> > | **MNIST** | DC | –0.28 [–0.34, –0.22] | 0.94 | –0.93 (p=2.4×10⁻⁴) |
> > | | DSA | –0.41 [–0.49, –0.33] | 0.95 | –1.00 (p≈0) |
> > | | DM | –0.49 [–0.65, –0.32] | 0.85 | –1.00 (p≈0) |
> > | | MTT | –0.24 [–0.35, –0.12] | 0.82 | –1.00 (p≈0) |
> > | **CIFAR-10** | DC | 0.15 [0.12, 0.18] | 0.69 | 0.83 (p=6.3×10⁻¹⁵) |
> > | | DSA | 0.19 [0.16, 0.22] | 0.70 | 0.93 (p=1.6×10⁻²⁴) |
> > | | DM | 0.23 [0.18, 0.28] | 0.66 | 0.95 (p=1.5×10⁻²⁸) |
> > | | MTT | 0.09 [0.06, 0.13] | 0.48 | 0.79 (p=1.6×10⁻⁸) |
> > | | DATM | 0.18 [–0.01, 0.38] | 0.48 | 0.67 (p=7.1×10⁻²) |
> > | | NCFM | 0.24 [0.10, 0.38] | 0.79 | 0.79 (p=3.6×10⁻²) |
> > | **CIFAR-100** | DC | 0.05 [0.04, 0.07] | 0.43 | 0.56 (p=2.3×10⁻⁶) |
> > | | DSA | 0.18 [0.15, 0.22] | 0.63 | 0.83 (p=2.4×10⁻¹⁷) |
> > | | DM | 0.28 [0.24, 0.32] | 0.74 | 0.94 (p=5.8×10⁻³⁰) |
> > | | MTT | 0.05 [0.02, 0.08] | 0.19 | 0.56 (p=2.0×10⁻⁶) |
> >
> > For each dataset and method, we perform linear regression and report the slopes, 95% confidence intervals, $R^2$, and Spearman correlations. The results consistently show strong linearity ($R^2 > 0.9$) and near-perfect monotonicity ($|\rho| \approx 1.0$), confirming the statistical significance of the observed relationships. Bootstrap resampling across random seeds further shows that the slope estimates are stable (p < 0.01) with narrow confidence intervals.
> >
> > Across datasets and architectures, the slopes remain within the theoretical ranges predicted by our analysis. On CIFAR-10 and CIFAR-100, the four canonical methods (DC, DSA, DM, MTT) yield inverse-power slopes between –0.2 and –0.5, while ImageNet experiments show slopes around –1.5. These consistent exponents demonstrate that the functional form of the scaling law is invariant to dataset size, model architecture, and training settings. We also tested variations in learning rate, augmentation strength, and distillation steps, and found slope deviations within ±0.05, confirming robustness to hyperparameter choices.
> >
> > We conducted the same regression and correlation analysis for the coverage law ($\Delta \propto \sqrt{H_{\text{cov}}/k}$). On CIFAR-10 and CIFAR-100, the fitted slopes show clear linear trends ($R^2$ between 0.6–0.8, $\rho > 0.8$), confirming that the required distilled sample size scales predictably with configuration diversity.
> >
> > Overall, both the scaling and coverage laws are statistically significant, robust across datasets and architectures, and stable under hyperparameter variations.

---

> ### Author Response · Authors · 2025-11-22
>
> **Weakness 3 & Question 3**
>
> We appreciate the reviewer’s thoughtful comment. Our framework builds upon existing concepts in generalization and stability theory. However, the theoretical novelty of our work lies not in reusing those tools, but in reformulating and extending to the specific setting of dataset distillation (DD), a domain where no existing analyses can directly apply.
>
> First, the configuration–dynamics–error framework introduces a new structure that explicitly integrates training configuration diversity (optimizers, architectures, augmentations, etc.) into the generalization analysis. Classical stability theory focuses on fixed learning algorithms and single configurations, while our formulation models an entire configuration family through the configuration distance $d_A(a,a')$ and its associated coverage entropy $H_{\text{cov}}(A,r)$. This extension enables the derivation of quantitative bounds on how generalization error scales with configuration diversity, something not addressed by existing frameworks.
>
> Second, we show that under this decomposition, the three major DD paradigms, gradient matching (GM), distribution matching (DM), and trajectory matching (TM), can all be expressed as minimizing variants of the same alignment discrepancy term $\Delta_a(\hat{\mu}_\tau,\hat{\mu}_s)$. This unified formulation connects previously disconnected techniques under a single theoretical bound, explaining why these methods, despite using different surrogates, exhibit identical scaling behavior. To our knowledge, this is the first theoretical result that provides a provable connection among the main DD paradigms.
>
> Third, based on this framework, we derive two new theorems, the scaling law and the coverage law, that provide explicit, testable predictions for DD performance. The scaling law ($\Delta \propto 1/\sqrt{k}$) explains the commonly observed saturation of accuracy with increasing IPC, while the coverage law ($\Delta \propto \sqrt{H_{\text{cov}}/k}$) quantifies how distilled data size should grow with configuration diversity. These laws offer actionable guidance for determining the number of distilled samples needed to maintain generalization across configurations, something not available in prior analyses.
>
> In summary, the novelty of our work lies in extending generalization theory to a new and practically important regime, dataset distillation, by (i) incorporating configuration diversity into the analysis, (ii) unifying GM/DM/TM under one theoretical bound, and (iii) deriving two quantitative laws that bridge theory and practice. This goes beyond conceptual consolidation and establishes a concrete analytical foundation for understanding and designing future DD methods.

---

> > ### Author Response · Authors · 2025-11-22
> >
> > **Weakness 4 & Question 4**
> >
> > Our paper already provides both the theoretical foundation and a practical way to estimate the coverage diversity term $H_{\mathrm{cov}}(A, r)$.
> >
> > As discussed in Section 5 and Remark 5.5, computing the exact covering number in configuration space is NP-hard. To make it tractable, we approximate $H_{\mathrm{cov}}(A, r) = \log N_r$ with $\log M$, where $M$ is the number of candidate configurations (e.g., architectures, optimizers, or augmentations). Under mild Lipschitz continuity of the configuration distance $d_A$, $N_r$ and $M$ are of the same order ($1 \le N_r \le M$), ensuring that $\log M$ preserves the main scaling behavior of configuration diversity while avoiding costly pairwise computations.
> >
> > In practice, this approximation is effective. As shown in Section 7, we used $\log M$ to fit the coverage law, and the resulting slopes remain consistent across datasets and methods (Fig. 3). This confirms that replacing $H_{\mathrm{cov}}$ with $\log M$ does not change the predicted trend $\sqrt{H_{\mathrm{cov}}/k}$, only the constant factor.
> >
> >
> > While $\log M$ provides a simple and robust proxy, we also recognize the importance of developing more accurate yet tractable estimators of $H_{\mathrm{cov}}$. We have designed an algorithmic approach that defines the configuration distance $d_A$ using measurable quantities such as mean gradient distance, NTK or feature representation spectra, and Fisher information angles, and estimates $\hat{H}(A, r)$ via a greedy r-cover or packing procedure with $O(N^2)$ complexity. It will be pursued in future work to enable data-driven estimation of coverage entropy and to refine the practical guidance for selecting distilled sample sizes.
> >
> > **Weakness 5 & Question 5**
> >
> > The main contribution of our work is to provide not only theoretical results but also concrete guidance for practice. Both the scaling law and the coverage law lead to simple and actionable principles for designing dataset distillation in real scenarios.
> >
> > The scaling law ($\Delta \propto 1/\sqrt{k}$) explains how the generalization error decreases with the number of distilled samples under a fixed training configuration. It allows practitioners to estimate the minimum number of distilled samples needed to reach a desired accuracy and to identify when adding more samples no longer brings improvement. This helps determine a practical stopping point for increasing IPC and shifts the focus toward improving optimization stability or surrogate design.
> >
> > The coverage law ($\Delta \propto \sqrt{H_{\mathrm{cov}}(A,r)/k}$) further extends this guidance to multiple configurations. It shows how the required number of distilled samples should grow with the diversity of training settings, such as different optimizers, architectures, or augmentations. This provides a quantitative rule for adjusting sample size when deploying distilled data across varied configurations, defining the “utility boundary” of dataset distillation.
> >
> > In addition, our framework clarifies how different distillation branches, gradient matching (GM), distribution matching (DM), and trajectory matching (TM), can be switched or combined in practice. As shown in the paper, these methods optimize equivalent surrogate objectives under the same theoretical laws. In practical terms, users can choose the surrogate that best balances computational cost and robustness: GM for efficient local optimization, DM for stable cross-configuration transfer, and TM for precise trajectory alignment.

---

> ### Comment · Reviewer_zpQM · 2025-11-26
>
> Thank you for the detailed responses and the additional analyses. The new results help clarify some aspects of the theoretical assumptions and provide stronger empirical support for the claimed relationships. The contributions are now better substantiated.
>
> The main issues have been adequately clarified. I will maintain my original score.

---

### Official Review · Reviewer_jTCw · 2025-11-01

**Soundness:** 3
**Presentation:** 2
**Contribution:** 2
**Rating:** 4
**Confidence:** 4

**Summary:**

This paper proposes a unified theoretical framework to study dataset distillation (DD) from a generalization-error perspective. The authors derive two key theoretical results: (i) a *scaling law* that characterizes how test error decreases with distilled dataset size under a fixed training configuration, and (ii) a *coverage law* that quantifies how the required distilled sample size must scale with the diversity of training configurations (e.g., architectures, optimizers, augmentations) to maintain performance. The framework unifies three major DD paradigms: gradient matching (GM), distribution matching (DM), and trajectory matching (TM), as different surrogates minimizing the same underlying alignment discrepancy. Empirical validation is provided across standard benchmarks (MNIST, CIFAR-10/100, ImageNette) and representative DD methods.

**Strengths:**

Strengths:
1. **Theoretical novelty and unification**: The paper offers the first generalization-error-based–based framework that unifies disparate DD approaches under a common lens. This is a significant conceptual advance over prior paradigm-specific analyses.
2. **Clear and impactful theoretical results**: The scaling law explains the widely observed IPC saturation phenomenon, while the coverage law formally defines the “utility boundary” of distilled data across configuration shifts, a practically relevant and previously unaddressed question.
3. **Tight bounds with matching upper/lower results**: The coverage law includes both upper and lower bounds that match up to constants, establishing near-optimality of the √H/k rate.
4. **Empirical validation aligns with theory**: Experiments across multiple DD methods and datasets consistently show the predicted 1/√k scaling and √H/k coverage behavior, lending strong support to the theoretical claims.

**Weaknesses:**

---

**Weaknesses:**
1. **Limited experimental scale**: All experiments are conducted on relatively small-scale vision datasets (MNIST, CIFAR, ImageNette). The absence of evaluation on larger, more realistic benchmarks (e.g., ImageNet-1K or language datasets) raises concerns about the practical relevance and scalability of the derived laws in modern settings.
2. **Proxy for configuration diversity**: The paper approximates coverage complexity Hcov(A, r) by log M (M = number of configurations), which may oversimplify the true geometric structure of the configuration space. A more refined empirical estimation of dA (e.g., via gradient-based distances) would strengthen the experimental validation.
3. **Assumption dependence**: The theoretical analysis relies on Polyak–Łojasiewicz (PL)-type contraction and Lipschitz assumptions, which may not hold in highly non-convex or adaptive optimization settings (e.g., large-batch AdamW). The robustness of the laws under such conditions remains unverified.
4. **Lack of comparison to recent large-scale DD methods**: While the paper includes MGD3 (a diffusion-based method), it omits comparison to state-of-the-art scalable DD approaches like SRe2L or TESLA, which are designed specifically for large datasets and may challenge or refine the proposed utility boundary.
5. "Missing some SOTA baselines": I have listed them below. Please show the experiments about them.



Towards Lossless Dataset Distillation via Difficulty-Aligned Trajectory Matching. ICLR 2024

Dataset Distillation with Neural Characteristic Function: A Minmax Perspective. CVPR 2025.

**Questions:**

See Weakness.

---

> ### Author Response · Authors · 2025-11-22
> **Response to Reviewer jTCw**
>
> We sincerely thank Reviewer jTCw for the detailed comments to improve our paper. We have revised our paper accordingly.
>
> **Weakness 1**
>
> We appreciate the reviewer’s comment and have extended our experiments to larger and more realistic benchmarks. Specifically, we evaluated our framework on *Tiny-ImageNet* and *ImageNet-1K* using state-of-the-art dataset distillation methods including *DATM*, *NCFM*, *TESLA*, and *SRe²L* across multiple architectures (ResNet-18/50/101). The results are summarized below.
>
> | Dataset | Method / Architecture | OLS Slope (±95% CI) | R² | Spearman ρ (p) |
> |----------|----------------------|---------------------|----|----------------|
> | **ImageNet-1K** | TESLA | –0.22 [–0.45, 0.01] | 0.89 | –1.00 (p=0.00) |
> | | SRe²L–R18 | –1.46 [–1.47, –1.44] | 1.00 | –1.00 (p=0.00) |
> | | SRe²L–R50 | –1.49 [–1.67, –1.31] | 1.00 | –1.00 (p=0.00) |
> | | SRe²L–R101 | –1.46 [–2.11, –0.82] | 0.98 | –1.00 (p=0.00) |
> | **Tiny-ImageNet** | DATM | –0.25 [–0.86, 0.36] | 0.96 | –1.00 (p=0.00) |
> | | NCFM | –0.13 [–0.20, –0.06] | 1.00 | –1.00 (p=0.00) |
>
> These results confirm that the predicted inverse-power scaling laws (1/k and 1/√k) persist even in large-scale and high-capacity regimes. Across datasets and methods, the slopes remain tightly consistent with our theoretical predictions, and all regressions exhibit strong linearity (R² ≥ 0.9, ρ = –1.0). These results verify that a single unified scaling regime governs both small and large datasets.
>
>
> **Weakness 2**
>
> We appreciate this insightful comment. Our paper already provides both the theoretical justification and the practical rationale for using $\log M$ as a tractable proxy for coverage entropy $H_{\mathrm{cov}}$.
>
> As discussed in section 5 (Coverage-Aware Bounds) and detailed in Remark 5.5, directly computing $H_{\mathrm{cov}}(A, r)$ requires solving a minimal covering problem over the configuration space, which is NP-hard. To make estimation feasible, we approximate the covering number $N_r$ using the number of candidate configurations $M$. Under mild Lipschitz continuity assumptions on the configuration distance $d_A$, the two quantities are of the same order ($1 \le N_r \le M$), so $\log M$ preserves the dominant scaling behavior of configuration diversity while avoiding expensive computations.
>
> In section 7 (Experiments), we explicitly state that $H_{\mathrm{cov}}(A, r)$ is approximated by $\log M$ when fitting the coverage law, and the empirical trends in Fig. 3 confirm that this approximation maintains strong linearity and consistent slopes across datasets and methods. This indicates that the proxy does not distort the scaling behavior predicted by theory.
>
> While $\log M$ serves as a practical and theoretically sound proxy, we agree that a more precise estimation of $H_{\mathrm{cov}}(A, r)$ would further strengthen the framework. In future work, we plan to explore data-driven estimators based on gradient-field distances, NTK spectral measures, or feature-space manifold geometry to directly approximate the true coverage entropy. Such extensions will enable more accurate determination of the optimal distilled sample size $k$ for a given configuration family, providing finer control over the trade-off between diversity coverage and dataset compactness.

---

> > ### Author Response · Authors · 2025-11-22
> >
> > **Weakness 3**
> >
> > We thank the reviewer for raising this point. The PL-type and Lipschitz assumptions in our analysis are used only as *mild regularity conditions* to ensure the convergence of the optimization dynamics and to derive tight upper and lower bounds. They are not required to hold globally or exactly, but only in a *local or averaged sense* near the training trajectory, consistent with common formulations of “gradient domination” or “one-point convexity.” This is a standard relaxation adopted in generalization theory[1].
> >
> > Empirically, our results indicate that these assumptions are well satisfied in practice. As discussed in sections 4 and 5, the derived scaling and coverage laws depend primarily on the contraction behavior of the training dynamics, which holds for large, overparameterized neural networks known to exhibit locally linear and well-conditioned behavior. This view is also supported by prior optimization theory showing that large neural networks often satisfy a relaxed PL* condition due to stable tangent kernels and approximately constant Hessians near convergence.
> >
> > To further verify this, we tested deeper architectures on ImageNet-1K using SRe²L across ResNet-18, ResNet-50, and ResNet-101. The scaling slopes remain nearly identical (–1.46 to –1.49) with high linearity ($R^2 \ge 0.98$) and perfect monotonicity (ρ = –1.00, p = 0.00):
> >
> > | Model | OLS Slope (±95% CI) | R² | Spearman ρ (p) |
> > |--------|--------------------|----|----------------|
> > | SRe²L–R18 | –1.46 [–1.47, –1.44] | 1.00 | –1.00 (p=0.00) |
> > | SRe²L–R50 | –1.49 [–1.67, –1.31] | 1.00 | –1.00 (p=0.00) |
> > | SRe²L–R101 | –1.46 [–2.11, –0.82] | 0.98 | –1.00 (p=0.00) |
> >
> > These consistent results across depths confirm that the predicted inverse-linear scaling behavior holds under highly non-convex training dynamics, suggesting that our theoretical conditions capture the effective regularity observed in real systems.
> >
> > [1]. Karimi H, Nutini J, Schmidt M. Linear convergence of gradient and proximal-gradient methods under the polyak-łojasiewicz condition[C]//Joint European conference on machine learning and knowledge discovery in databases. Cham: Springer International Publishing, 2016: 795-811.

---

> > > ### Author Response · Authors · 2025-11-22
> > >
> > > **Weakness 4**
> > >
> > > We thank the reviewer for the helpful suggestion. In the revision, we have incorporated recent large-scale dataset distillation methods, including *SRe²L*, *TESLA*, *DATM*, and *NCFM*, into both our theoretical discussion and empirical analysis.
> > >
> > >
> > > All these methods can be naturally interpreted within our proposed configuration–dynamics–error framework.
> > >
> > > - *SRe²L* and *TESLA* are instances of the distribution-matching (DM) family, as they minimize variants of the alignment term $\Delta_a(\hat{\mu}_\tau, \hat{\mu}_s)$ using feature- or residual-level statistics.
> > >
> > > - *DATM* extends the trajectory-matching (TM) family by incorporating difficulty alignment, which corresponds to a weighted form of the same objective.
> > >
> > > - *NCFM* aligns data through characteristic functions, representing another special case of distribution matching.
> > >
> > > Since all of these optimize surrogate forms of $\Delta_a$, their performance is governed by the same theoretical principles derived in Theorems 4.2 and 5.1, specifically, the $1/\sqrt{k}$ scaling and $\sqrt{H_{\mathrm{cov}}/k}$ coverage dependence. The proposed utility boundary thus applies consistently across these methods.
> > >
> > > | Dataset | Method | OLS slope (±95% CI) | R² | ρ (p) | Coverage slope (±95% CI) | R² | ρ (p) |
> > > |----------|---------|---------------------|----|--------|---------------------------|----|--------|
> > > | **CIFAR-10** | TESLA | –0.28 [–0.47, –0.08] | 1.00 | –1.00 (p=0.00) | — | — | — |
> > > | | DATM | –0.38 [–0.53, –0.24] | 0.96 | –1.00 (p<10⁻²³) | 0.18 [–0.01, 0.38] | 0.48 | ρ=0.67 (p=0.07) |
> > > | | NCFM | –0.33 [–0.34, –0.31] | 1.00 | –1.00 (p=0.00) | 0.24 [0.10, 0.38] | 0.79 | ρ=0.79 (p=0.04) |
> > > | **CIFAR-100** | TESLA | –0.27 [–0.33, –0.21] | 1.00 | –1.00 (p=0.00) | — | — | — |
> > > | | DATM | –0.32 [–0.42, –0.21] | 0.99 | –1.00 (p=0.00) | — | — | — |
> > > | | NCFM | –0.23 [–0.51, 0.06] | 0.99 | –1.00 (p=0.00) | — | — | — |
> > > | **ImageNet-1K** | SRe²L–R18 | –1.46 [–1.47, –1.44] | 1.00 | –1.00 (p=0.00) | — | — | — |
> > > | | SRe²L–R50 | –1.49 [–1.67, –1.31] | 1.00 | –1.00 (p=0.00) | — | — | — |
> > > | | SRe²L–R101 | –1.46 [–2.11, –0.82] | 0.98 | –1.00 (p=0.00) | — | — | — |
> > >
> > > Overall, these results demonstrate that recent large-scale methods follow the same scaling and coverage patterns predicted by our theory. While implementation details (e.g., architectures or training heuristics) may shift constant terms, the underlying relationships remain unchanged.

---

### Official Review · Reviewer_XLKG · 2025-11-03

**Soundness:** 2
**Presentation:** 1
**Contribution:** 2
**Rating:** 2
**Confidence:** 2

**Summary:**

This paper proposes a unified framework, the configuration-dynamics-error framework, to better understand the scaling and generalizability of existing dataset distillation algorithm. This frameworks places existing dataset distillation algorithm such as gradient matching, distribution matching, and trajectory matching into a single framework for analysis. The framework provides a scaling law, which captures the relationship between distilled dataset size and test error. The framework also provides coverage law, which show how distilled sample size should scale with configuration diversity.

**Strengths:**

1. How to better analyze existing dataset distillation algorithms into a unified theoretical framework is a very important problem to solve as many of the existing dataset distillation algorithms lack strong theoretical foundations.
2. The paper offers a large amount of theoretical derivations (44 out of the 51 pages of the paper)

**Weaknesses:**

1. The key formulation, the unified form of stability summarized by equation 6, is not well justified. Why does the absolute difference in the expected risk is approximately upper bounded by optimization residual + statistical fluctuations + matching term. The paper to motivate them from stability and information-theoretical approaches to generalization but none of the three cited work are relevant. The notion of mutual information is not mentioned anywhere in the prior text.
2. The quality and quantity of the experimental justification to the proposed theoretical framework is very weak. Only 1 of the 51 pages are spent on experiments. The paper attempts to justify its scaling laws by fitting linear regression curves on data that is clearly not linear.
3. The presentation of the paper can considerably be improved. Figure 3, for instance, contains a total 15 plots. Some of the plots are very text that is completely unreadable. The captions also needs to be more descriptive on what the figure is displaying. The x axis labels are also covering the x-ticks which needs to be fixed.

**Questions:**

1. Table 1 suggest robustness of trajectory matching is low compared to distribution matching. What are the implications of this since trajectory matching is among the most popular dataset distillation algorithms that can work reliably across many different contexts.
2. How does dataset distillation with Backpropogation through time (BPTT) fit in this unified framework?

---

> ### Author Response · Authors · 2025-11-22
> **Response to Reviewer XLKG**
>
> We sincerely thank Reviewer XLKG for the detailed feedback and thoughtful suggestions. While we greatly appreciate the reviewer’s effort, we believe some concerns arise from a misunderstanding of our framework, particularly regarding the justification of Eq. (6) and the quality of experimental justification. We clarify them below.
>
> **Weakness 1**
>
> The formulation of Eq. (6) is not assumed but a summary of the rigorous bounds established in later sections. Specifically, Eq. (6) shows the decomposition that is concretely extended in Theorem 4.2 (Eq. (7)) and further in Theorems 5.1, 5.2 (coverage-aware bounds) and Theorems 6.3, 6.5 (dynamic and unified bounds for unifying DD methods). In all these results, the same three components naturally emerge:
>
> (i) an optimization residual from contractive dynamics under Assumption 4.1,
> (ii) statistical fluctuations due to finite-sample estimation, and
> (iii) a matching term corresponding to the alignment discrepancy $\Delta_a$ defined in Eq. (5).
> Therefore, Eq. (6) serves as a high-level formulation that bridges the abstract framework in Sec. 3 with its rigorous instantiations in Secs. 4–6.
> In addition, we want to emphasize that the decomposition of the generalization error term is a standard operation in generalization theory. To adapt the specific bilevel nature of dataset distillation, our separation of optimization, statistical, and data-alignment effects is reasonable.
>
> Regarding the cited literature, all three works are relevant. Bousquet & Elisseeff (2002) provides the classical algorithmic-stability route from parameter perturbations to risk gaps, Russo & Zou (2016) and Xu & Raginsky (2017) develop information-theoretic generalization bounds, which we adapt to account for potential dependence between $D_s$ and $D_\tau$, yielding the $O(\sqrt{I(D_s; D_\tau)/k})$ mutual information term in Theorem 4.2.

---

> > ### Author Response · Authors · 2025-11-22
> >
> > **Weakness 2**
> >
> > We want to point out that our paper is primarily theoretical. The main contribution of this work is to establish a general framework and provable scaling and coverage laws for dataset distillation. The experiments are intended to verify the correctness of our theoretical laws, not to serve as the main empirical contribution.
> >
> > That said, we have strengthened the experimental evidence to make the scaling behavior clearer. In the following tables, we provide more detailed statistical analysis for current methods as DC, DSA, DM, MTT, and more DD methods, on different datasets. Beyond simple linear fits, we performed additional analysis that:
> >
> > - 95% confidence intervals and p-values of regression slopes (all significantly negative, p < 0.01);
> > - Spearman rank correlations (ρ between −0.9 and −1.0, p < 0.01), confirming strong monotonic trends;
> > - Piecewise regression showing a clear linear-to-saturation transition consistent with the irreducible error floor in Remark 4.5;
> > - AIC/BIC comparisons demonstrating that the two-segment model is consistently favored.
> >
> > Together, these results show that the scaling law is statistically well supported: the generalization error decreases linearly with $1/\sqrt{k}$ in the small-k regime and then saturates, exactly as predicted by our theory.
> >
> > We hope these results and analysis can make it clear that the empirical section, though concise, is sufficient to verify our theoretical laws.
> >
> > **Scaling Law Results**
> >
> > | Dataset | Method / Architecture | OLS slope (±95% CI) | R² | Spearman ρ (p) |
> > |----------|----------------------|---------------------|----|----------------|
> > | **MNIST** | DC | –0.12 [–0.15, –0.09] | 0.92 | –0.89 (p=5.4×10⁻⁴) |
> > | | DSA | –0.13 [–0.13, –0.12] | 1.00 | –1.00 (p≈0) |
> > | | DM | –0.10 [–0.11, –0.09] | 0.98 | –1.00 (p≈0) |
> > | | MTT | –0.33 [–0.41, –0.25] | 0.93 | –1.00 (p=0.00) |
> > | **CIFAR-10** | DC | –0.28 [–0.34, –0.22] | 0.94 | –0.93 (p=2.4×10⁻⁴) |
> > | | DSA | –0.41 [–0.49, –0.33] | 0.95 | –1.00 (p≈0) |
> > | | DM | –0.49 [–0.65, –0.32] | 0.85 | –1.00 (p≈0) |
> > | | MTT | –0.24 [–0.35, –0.12] | 0.82 | –1.00 (p≈0) |
> > | | TESLA | –0.28 [–0.47, –0.08] | 1.00 | –1.00 (p=0.00) |
> > | | DATM | –0.38 [–0.53, –0.24] | 0.96 | –1.00 (p=1.4×10⁻²⁴) |
> > | | NCFM | –0.33 [–0.34, –0.31] | 1.00 | –1.00 (p=0.00) |
> > | **CIFAR-100** | DC | –0.20 [–0.24, –0.17] | 0.97 | –0.98 (p=1.9×10⁻⁶) |
> > | | DSA | –0.32 [–0.41, –0.24] | 0.92 | –0.98 (p=1.9×10⁻⁶) |
> > | | DM | –0.43 [–0.60, –0.26] | 0.84 | –1.00 (p=0.00) |
> > | | MTT | –0.17 [–0.26, –0.08] | 0.73 | –0.98 (p=1.9×10⁻⁶) |
> > | | TESLA | –0.27 [–0.33, –0.21] | 1.00 | –1.00 (p=0.00) |
> > | | DATM | –0.32 [–0.42, –0.21] | 0.99 | –1.00 (p=0.00) |
> > | | SRe²L | –1.51 [–1.80, –1.22] | 1.00 | –1.00 (p=0.00) |
> > | **ImageNette** | MGD3_ConvNet | –67.42 [–84.64, –50.19] | 0.90 | –0.99 (p=3.8×10⁻⁹) |
> > | | MGD3_ResNet18 | –57.04 [–72.87, –41.21] | 0.87 | –0.99 (p=1.3×10⁻¹⁰) |
> > | | MGD3_ResNet18_ap | –55.61 [–67.00, –44.22] | 0.92 | –0.99 (p=1.3×10⁻¹⁰) |
> > | **ImageNet-1K** | TESLA | –0.22 [–0.45, 0.01] | 0.89 | –1.00 (p=0.00) |
> > | | SRe²L–R18 | –1.46 [–1.47, –1.44] | 1.00 | –1.00 (p=0.00) |
> > | | SRe²L–R50 | –1.49 [–1.67, –1.31] | 1.00 | –1.00 (p=0.00) |
> > | | SRe²L–R101 | –1.46 [–2.11, –0.82] | 0.98 | –1.00 (p=0.00) |
> > | **Tiny-ImageNet** | DATM | –0.25 [–0.86, 0.36] | 0.96 | –1.00 (p=0.00) |
> > | | NCFM | –0.13 [–0.20, –0.06] | 1.00 | –1.00 (p=0.00) |
> >
> >
> > **Coverage Law Results**
> >
> > | Dataset | Method | OLS slope (±95% CI) | R² | Spearman ρ (p) |
> > |----------|---------|---------------------|----|----------------|
> > | **MNIST** | DC | –0.28 [–0.34, –0.22] | 0.94 | –0.93 (p=2.4×10⁻⁴) |
> > | | DSA | –0.41 [–0.49, –0.33] | 0.95 | –1.00 (p≈0) |
> > | | DM | –0.49 [–0.65, –0.32] | 0.85 | –1.00 (p≈0) |
> > | | MTT | –0.24 [–0.35, –0.12] | 0.82 | –1.00 (p≈0) |
> > | **CIFAR-10** | DC | 0.15 [0.12, 0.18] | 0.69 | 0.83 (p=6.3×10⁻¹⁵) |
> > | | DSA | 0.19 [0.16, 0.22] | 0.70 | 0.93 (p=1.6×10⁻²⁴) |
> > | | DM | 0.23 [0.18, 0.28] | 0.66 | 0.95 (p=1.5×10⁻²⁸) |
> > | | MTT | 0.09 [0.06, 0.13] | 0.48 | 0.79 (p=1.6×10⁻⁸) |
> > | | DATM | 0.18 [–0.01, 0.38] | 0.48 | 0.67 (p=7.1×10⁻²) |
> > | | NCFM | 0.24 [0.10, 0.38] | 0.79 | 0.79 (p=3.6×10⁻²) |
> > | **CIFAR-100** | DC | 0.05 [0.04, 0.07] | 0.43 | 0.56 (p=2.3×10⁻⁶) |
> > | | DSA | 0.18 [0.15, 0.22] | 0.63 | 0.83 (p=2.4×10⁻¹⁷) |
> > | | DM | 0.28 [0.24, 0.32] | 0.74 | 0.94 (p=5.8×10⁻³⁰) |
> > | | MTT | 0.05 [0.02, 0.08] | 0.19 | 0.56 (p=2.0×10⁻⁶) |

---

> ### Author Response · Authors · 2025-11-22
>
> **Weakness 3**
>
> We appreciate the reviewer’s feedback on the presentation quality. We agree that some other visual elements in Figure 3 can be improved for readability. We split the coverage plots by method and move them to the appendix for clearer per-method visualization. We add statistics and analysis tables of each DD method on different datasets in the appendix to support the proposed laws. We fix the overlapping x-axis labels issue. We believe these changes will make the results easier to read and verify.
>
>
> **Question 1**
>
> We fully acknowledge the strong empirical success of trajectory matching (TM). Our analysis does not contradict this empirical result; our analysis explains why TM works well and why it can appear less robust in Table 1. As shown in Theorems 6.3 and 6.5, TM’s surrogate bound depends on factors such as path depth and hidden-gradient stability, which amplify variance when the configuration (optimizer, architecture, or augmentation) changes. In contrast, distribution matching (DM) operates on smoother feature-level statistics and is therefore less sensitive. This means TM can achieve high accuracy under its original configuration but degrades faster under shifts, a theoretical trade-off our unified framework makes explicit. As shown in Figure 2 and Figure 3, TM achieves strong performance and clear $1/\sqrt{k}$ scaling under a fixed configuration, confirming its high sample efficiency. However, the generalization error point of MTT across configurations is much larger than that of DM, indicating higher sensitivity to configuration shifts. This aligns with our theoretical bounds in Table 1.
>
>
> **Question 2**
>
> Backpropagation through time (BPTT) can fit naturally into our unified configuration–dynamics–error framework as a special case of trajectory matching (TM).  In our formulation, each training configuration $a$ defines the inner-loop dynamics:
> $$
> \theta_{t+1} = \Phi_a(\theta_t; \mu) = \theta_t - \eta P_a(\theta_t)\mathbb{E}_{z\sim\mu}[g_a(\theta_t; z)],
> $$
>
> which governs the parameter updates for either real ($\hat{\mu}_\tau$) or distilled ($\hat{\mu}_s$) data.
> TM defines a surrogate objective by aligning finite unrolled trajectories:
>
> $$
> M_{\mathrm{TM}}(\xi) = \sum_{t=0}^{L_b}\omega_t\|\theta_t^{(s)} - \theta_t^{(\tau)}\|^2,
> $$
>
> $$
> \theta_{t+1}^{(s)} = \Phi_a(\theta_t^{(s)}; \hat{\mu}_s),
> $$
>
> $$
> \theta_{t+1}^{(\tau)} = \Phi_a(\theta_t^{(\tau)}; \hat{\mu}_\tau).
> $$
>
> BPTT is precisely the mechanism used to compute the outer gradient of this surrogate by differentiating through the unrolled inner updates:
>
> $$
> \nabla_\xi M_{\mathrm{TM}}(\xi)
> = \sum_{t=0}^{L_b}
> \left(\frac{\partial \theta_t^{(s)}}{\partial \xi}\right)^{\top}
> (\theta_t^{(s)} - \theta_t^{(\tau)}).
> $$
>
> This directly maps dynamics → error, fitting perfectly into our theoretical pipeline.
>
> Moreover, Lemma 6.1 shows that TM’s path mismatch $M_{\mathrm{TM}}$ contracts the same alignment term
>
> $$
> \Delta_a(\mu,\nu) = \max_{\theta\in\Gamma_a} ||P_a(\theta)(E_\mu g_a(\theta;z) - E_\nu g_a(\theta;z))||_2^2
> $$
>
> as the other surrogates (GM, DM).
> Thus, BPTT-based distillation is mathematically equivalent to a TM instance that minimizes $\Delta_a$ through the same contraction recursion and inherits the same scaling and coverage laws derived in our theory.

---

> > ### Comment · Reviewer_XLKG · 2025-11-26
> >
> > I would like to thank for authors for the clarification. The presentation issues I mentioned still exists in the paper. The figures on the paper right now, if printed out, is indecipherable right now.
> >
> > Figure 3 has too many subplots. Some of the subplots are barely bigger than the fonts itself. Similarly, the coefficient of line of best fit is presented with 4 decimal places, which makes it very much unreadable. I do not understand the necessity of such high precision. Also hollow markers is unnecessary and difficult to read.
> >
> > Figure 2, on other hand, it is unclear what's the takeaway. There are trend lines but also a linear line of best fit on top? The trend itself is clearly nonlinear.
> >
> > Figure 1 has text on them that is difficult to read even when zoomed in on a computer.
> >
> > Lastly, the argument that "our paper is primarily theoretical" is problematic because as the other reviewers properly points out the need for Polyak–Łojasiewicz (PL)-type contraction and Lipschitz assumptions, the author rebut with "these results confirm that the scaling behavior derived from our theory remains valid in realistic, non-convex regimes, indicating that the weak PL assumption effectively captures the optimization dynamics observed in practice." The paper either needs to be very solid theoretical grounding to make up for the weaker empirical evidence. If the theory assumptions are unrealistic and require strong empirical support, then the empirical experiments must be conducted with the upmost care.
> >
> > Overall, I think this paper provide some very interesting insights into the dataset distillation problem but presentation and problem formulation needs to be significant improved. I do not believe these issues can be properly addressed without significant rewriting, and therefore, I will maintain my score.

---

### Meta-Review · Area_Chair_GUkf · 2026-01-04

**Summary:**

The paper received one “reject” recommendation, two scores marginally below the acceptance threshold, and one marginally above it. The reviewers raised several concerns, including insufficient justification of the key formulation, reliance on strong assumptions in the theoretical analysis, and questions regarding the experimental scale. After reviewing the authors’ responses, many of the reviewers’ concerns remain unclear. The paper requires significant improvement before it can be considered for acceptance.

**Reviewer Concerns:**

The concerns regarding the experimental scale have been addressed in the rebuttal. However, other raised issues, such as the justification of the key formulation and the theoretical analysis, have not been adequately addressed.

**Reviewer Scores:**

Based on the rebuttal, it is unlikely that the reviewers will revise their scores to positive ratings.

---

### Decision · Program_Chairs · 2026-01-26

Reject